# T1: One-to-One Channel-Head Binding for Multivariate Time-Series Imputation

**Dongik Park, Hyunwoo Ryu, Suahn Bae, Keondo Park, and Hyung-Sin Kim**†
Graduate School of Data Science, Seoul National University
{breadsora, yahhallong, snubbb, gundo0102, hyungkim}@snu.ac.kr

## Abstract

Imputing missing values in multivariate time series remains challenging, especially under diverse missing patterns and heavy missingness. Existing methods suffer from suboptimal performance as corrupted temporal features hinder effective cross-variable information transfer, amplifying reconstruction errors. Robust imputation requires both extracting temporal patterns from sparse observations within each variable and selectively transferring information across variables—yet current approaches excel at one while compromising the other. We introduce **T1** (**T**ime series imputation with **1**-to-1 channel-head binding), a CNN-Transformer hybrid architecture that achieves robust imputation through *Channel-Head Binding*—a mechanism creating one-to-one correspondence between CNN channels and attention heads. This design enables selective information transfer: attention pathways adapt based on observable patterns, down-weighting corrupted channels while maintaining reliable cross-variable connections. Experiments on 11 benchmark datasets demonstrate that T1 achieves state-of-the-art performance, reducing MSE by 46% on average compared to the second-best baseline, with particularly strong gains under extreme sparsity (70% missing ratio). The model generalizes to unseen missing patterns without retraining and uses a consistent hyperparameter configuration across all datasets. The code is available at https://github.com/Oppenheimerdinger/T1.

## 1 Introduction

Multivariate time-series data underpin decision making in healthcare (Ghassemi et al., 2015; Lee & Hauskrecht, 2021; Lee et al., 2025), finance (Niu et al., 2020), climate (Nketiah et al., 2023; Chen & Dong, 2025), and industrial monitoring (Sharma et al., 2022). Yet measurements are routinely incomplete: sensors fail, transmissions drop, sampling is irregular, and entire windows go missing (Little & Rubin, 2019; Silva et al., 2012; Yi et al., 2016). Before any downstream task—forecasting, anomaly detection, classification—can succeed, we must *impute* these gaps with high fidelity. Formally, given $X \in \mathbb{R}^{M \times T}$ ($M$ variables with sequence length $T$) and an observation mask $\Omega \subseteq [M] \times [T]$, the goal is to impute $X$ on $\Omega^c$ using only the observed entries $X|_\Omega$. This is challenging because imputation must simultaneously (1) reconstruct temporal structure from sparse, irregularly-sampled observations within each variable and (2) transfer complementary information across variables without importing noise. When temporal features are corrupted by missingness, cross-variable information transfer amplifies errors; when this transfer is naïve, it ignores which variables are informative under the current mask.

Current methods for time-series imputation leave a gap for robust, efficient processing under heavy missingness. As illustrated in (i)-(iv) of Figure 1a , existing approaches make architectural compromises that limit their effectiveness. (i) Time-axis tokenization approaches (Wu et al., 2021) suffer from fundamental limitations: Vanilla Transformers (Vaswani et al., 2017) mix all variables at each timestep token where missing values directly corrupt the representation, allowing corrupted features to contaminate all computations. While methods using diagonally-masked attention (Du et al., 2023) improve temporal modeling, they inherit the same tokenization problem—missing values degrade token representations that propagate through attention layers. (ii) Variable-axis tokenization (Liu

---

† Corresponding author.

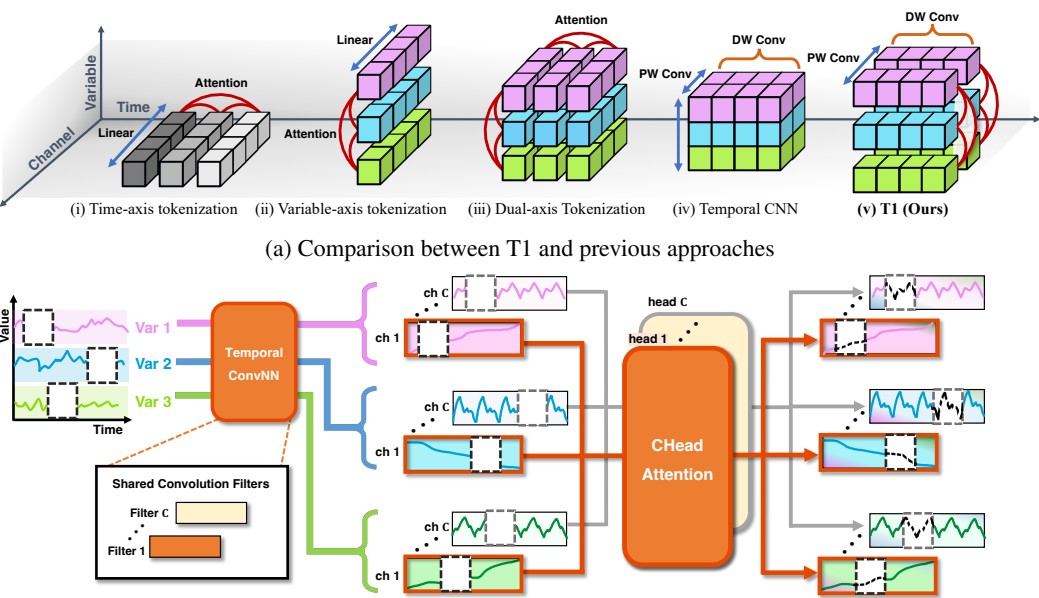

(a) Comparison between T1 and previous approaches

(b) *CHead Attention*: Tight binding between CNN channel and Attention head

Figure 1: T1 introduces CNN-Transformer hybrid architecture that effectively processes information by strategically assigning CNN or attention to the temporal, feature, and variable dimensions using depthwise (DW) and pointwise (PW) convolutions. In our novel mechanism, *CHead Attention*, each channel encoded by shared CNN is directly aligned with a single attention head. It facilitates cross-variable information exchange, ensuring that interactions occur only between semantically similar temporal features.

et al., 2024) addresses this but fuses all temporal patterns through a single representation, losing feature-level selectivity. (iii) Dual-axis tokenization methods (Nie et al., 2024) employ attention on both temporal and variable axes, but struggle to transfer information across both dimensions when missing values block intermediate pathways. (iv) Temporal Convolutional Neural Network (CNN) approaches (Wu et al., 2023; Luo & Wang, 2024) efficiently extract multi-scale temporal features but provide limited cross-variable information transfer.

We show that robust imputation benefits from task-aligned architecture—specialized temporal and cross-variable components whose information transfer accounts for their interdependencies. We propose **T1** (**T**ime series imputation with **1**-to-1 channel-head binding), a hybrid architecture where CNNs extract temporal features from incomplete observations within variables and attention performs selective cross-variable information transfer ((v) in Figure 1a). T1 employs modernized temporal convolutions (Luo & Wang, 2024), leveraging the inherent property of CNNs where each channel learns to capture distinct temporal patterns from the observed data. This process effectively encodes the input into a set of diverse feature maps, yielding variable tokens that directly parameterize query, key, and value representations for cross-variable attention. This design leverages each architecture's strengths for imputation: the convolutional modules excel at building robust temporal representations from sparse observations, while variable-wise attention dynamically identifies informative variables based on their observed patterns. However, a naïve combination of these modules is insufficient. When missingness corrupts specific temporal features, treating each variable as a single token forces all its channels to mix, preventing isolation of corrupted features from reliable ones during information transfer. This necessitates an architectural refinement for feature-specific control.

Our key mechanism, *Channel-Head Binding* (CHead Attention, Figure 1b), seamlessly integrates CNNs and inter-variable attention, by creating a one-to-one correspondence between CNN channels and attention heads. Each CNN channel captures a distinct temporal feature while each attention head processes only its corresponding channel across variables, enabling fine-grained, feature-level information transfer pathways. This feature-level binding enables robust imputation: when missingness prevents a channel from observing its specialized pattern, the feature it extracts becomes less informative. Consequently, a corresponding attention head can temper its reliance on that chan-

nel during information transfer, while feature-level selectivity prevents these localized uncertainties from contaminating other channels.

In our extensive experiments across 11 benchmark datasets, T1 achieves state-of-the-art performance, demonstrating its effectiveness in diverse scenarios including point, block, and naturally occurring missingness. Furthermore, a model trained with a single missing ratio maintains performance when tested on both higher and lower ratios, a crucial property for real-world applications. These results are achieved using a consistent hyperparameter configuration across all datasets, suggesting robustness to hyperparameter choices.

Our main contributions are summarized as follows:

- We introduce *T1*, a CNN-Transformer hybrid architecture that tackles imputation through complementary specialization: CNNs for robust temporal feature extraction under missingness, and Transformers for selective information transfer across informative variables.

- We propose *Channel-Head Binding* (CHead Attention), an architectural mechanism that creates a one-to-one correspondence between CNN channels and attention heads, enabling robust imputation by isolating feature-specific information transfer pathways that adapt to varying missingness patterns.

- We demonstrate that T1 achieves state-of-the-art performance across 11 datasets, reducing MSE by 46% on average and maintaining this advantage under extreme missingness (70% missing ratio), while generalizing to unseen missing patterns without retraining.

## 2  RELATED WORK

**Time-series Imputation.** Time-series imputation has evolved from statistical methods (Dempster et al., 1977; Van Buuren & Groothuis-Oudshoorn, 2011) to deep learning approaches. RNN-based methods like BRITS (Cao et al., 2018) and M-RNN (Yoon et al., 2019) model bidirectional temporal dependencies. Transformer-based approaches including SAITS (Du et al., 2023) and ImputeFormer (Nie et al., 2024) leverage self-attention mechanisms with masked training objectives to capture long-range dependencies. Generative models, particularly diffusion-based CSDI (Tashiro et al., 2021), SSSD (Alcaraz & Strodthoff, 2023), and PriSTI (Liu et al., 2023a), achieve high quality through iterative refinement but with prohibitive inference latency. Graph methods like GRIN (Cini et al., 2022) and SPIN (Marisca et al., 2022) model inter-variable relationships via message passing but rely on static graphs that cannot adapt to instance-specific missingness.

**Temporal and Cross-variable Modeling.** Effective imputation requires both robust temporal extraction and selective cross-variable fusion, yet existing methods excel at one while compromising the other. For temporal modeling, linear models (DLinear, NLinear) decompose via projections (Zeng et al., 2023). Vanilla Transformers (Vaswani et al., 2017) tokenize all variables at each timestep, while extended versions like PatchTST (Nie et al., 2023), Autoformer (Wu et al., 2021), and FEDformer (Zhou et al., 2022) apply temporal attention with decomposition strategies. CNN-based methods—TCN (Bai et al., 2018), TimesNet (Wu et al., 2023), and notably ModernTCN (Luo & Wang, 2024)—extract multi-scale features through dilated or large-kernel depthwise convolutions. While powerful for temporal patterns, these methods *lack dynamic cross-variable relationships*. For cross-variable modeling, Crossformer (Zhang & Yan, 2023) attempts across temporal and variable dimensions but still entangles representations. iTransformer (Liu et al., 2024) achieves pure variable-axis attention by inverting dimensions, treating each variable's sequence as a single token for clean cross-variable fusion. However, these *compress or entangle temporal information*. Meanwhile, convolutional approaches like ModernTCN effectively capture temporal patterns but rely on static cross-variable mixing that cannot adapt to missing patterns. T1 combines these strengths through shared depthwise convolutions and variable-axis attention for cross-variable fusion. The shared convolutions ensure each channel extracts the same pattern type across all variables, while mask-aware embeddings and CHead Attention enable dynamic, validity-based information transfer.

## 3  THE T1 ARCHITECTURE FOR TIME SERIES IMPUTATION

We address the problem of time series imputation. Let a multivariate time series be represented by $X = \{x^{(1)}, ..., x^{(M)}\} \in \mathbb{R}^{M \times T}$ where $M$ denotes the number of variables and $T$ is the sequence

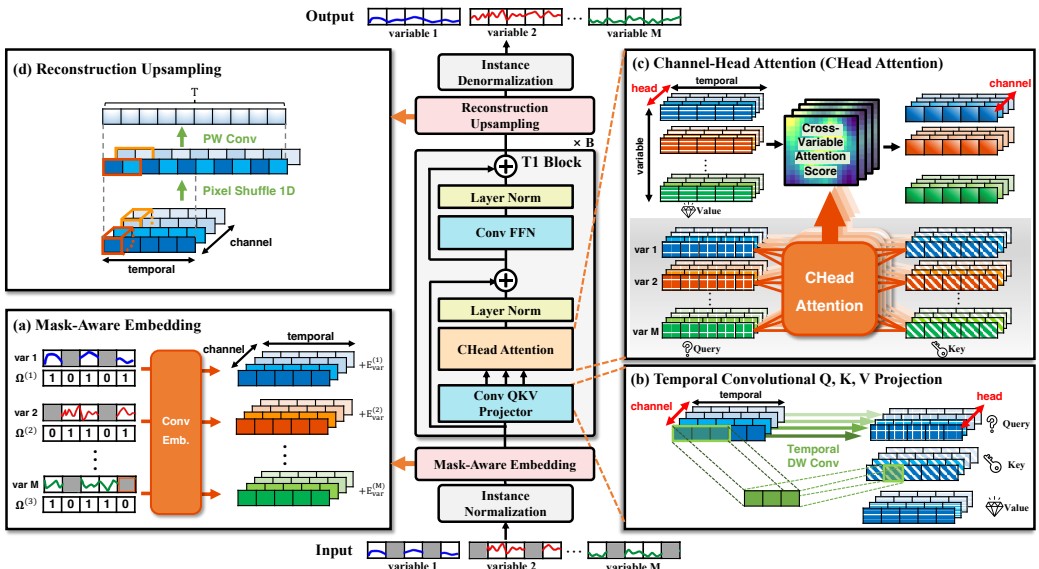

Figure 2: An overview of the T1 architecture. **(a)** The Mask-Aware Embedding module encodes the input series and its observation mask into a latent representation using 1D convolutions. **(b)** The Temporal Convolutional QKV Projection block employs Depthwise Convolutions to extract consistent temporal patterns for each channel. The kernel weights are shared across variables, resulting in semantically-aligned Query, Key, and Value embedding. **(c)** Our proposed Channel-Head Attention (CHead Attention) is applied across the variable axis to selectively transfer information. Each head is bound to a single channel, enabling feature-specific fusion between semantically-aligned patterns. **(d)** The Reconstruction Upsampler restores the original temporal resolution of the series via a parameter-free 1D PixelShuffle operation followed by a final pointwise convolution.

length. The accompanying observation mask $\Omega \in \{0,1\}^{M \times T}$ indicates whether a value is observed ($\Omega_{m,t} = 1$) or missing ($\Omega_{m,t} = 0$). The objective is to impute the missing values by leveraging each variable's unique temporal patterns and inter-variable correlations.

## 3.1 OVERALL ARCHITECTURE

As presented in Figure 2, our novel architecture, T1, comprises three main components: Mask-Aware Embedding, T1 blocks and Reconstruction Upsampler.

**Mask-Aware Embedding.** As an initial step, instance normalization is applied to each input series $x^{(m)}$, computing the normalized series as $x_{\text{norm}}^{(m)} = (x^{(m)} - \mu^{(m)})/\sigma^{(m)}$. To properly handle missing data in imputation tasks, the per-instance mean $\mu^{(m)}$ and standard deviation $\sigma^{(m)}$ are computed solely from observed values (where $\Omega_{m,t} = 1$) and stored for the final denormalization.

To explicitly encode missing value locations, the normalized series and its observation mask are stacked into a two-channel input (as presented in Figure 2a). The resulting tensor ($\in \mathbb{R}^{2 \times T}$) is processed by a strided 1D convolution with $C$ filters and augmented with a learnable variable-wise encoding, producing the final embedding $z^{(m)} \in \mathbb{R}^{C \times L}$ where $L$ is the latent temporal dimension:

$$z^{(m)} = \text{Conv1D}\left(\begin{bmatrix} x_{\text{norm}}^{(m)} \\ \Omega^{(m)} \end{bmatrix}\right) + E_{\text{var}}^{(m)} \tag{1}$$

Here $E_{\text{var}}^{(m)} \in \mathbb{R}^{C \times L}$ is a learnable variable-specific encoding (analogous to positional encoding for tokens).

**T1 Blocks.** The aggregated embedding $Z = [z^{(1)}, z^{(2)}, ..., z^{(M)}] \in \mathbb{R}^{M \times C \times L}$ is processed through stacked T1 blocks that implement a CNN-Transformer hybrid design. Each variable maintains independent temporal CNN feature spaces while Channel-Head Attention models inter-variable relationships. Optionally, downsampling can be applied between blocks to reduce the temporal resolution for subsequent layers. The details of T1 block design are presented in Section 3.2.

**Reconstruction Upsampler.** The final representation from the T1 blocks, denoted as $Z_{\text{out}} \in \mathbb{R}^{M \times C \times L}$, is passed to the reconstruction upsampler to generate the final imputed output, as presented in Figure 2d. For the upsampling stage, we employ a 1D variant of PixelShuffle (Shi et al., 2016), a parameter-free operation that rearranges the channel dimension into the temporal dimension. This process reshapes the input from $\mathbb{R}^{M \times C \times L}$ to $\mathbb{R}^{M \times (C/r) \times (L \cdot r)}$, where $r = T/L$ is the upsampling ratio. Using PixelShuffle1D avoids the checkerboard artifacts common in transposed convolutions while maintaining efficiency. A subsequent pointwise convolution (PWConv) projects to the target dimension:

$$\hat{x}_{\text{norm}} = \text{PWConv}(\text{PixelShuffle1D}(Z_{\text{out}})) \in \mathbb{R}^{M \times 1 \times T} \tag{2}$$

Final imputation $\hat{x}^{(m)} = \hat{x}_{\text{norm}}^{(m)} \cdot \sigma^{(m)} + \mu^{(m)}$ is obtained through denormalization using the stored statistics.

## 3.2 T1 Block

The T1 block addresses multivariate imputation through three specialized components: Temporal Convolutional Q, K, V Projection for multi-scale temporal feature extraction, CHead Attention for cross-variable information transfer, and Convolutional Feed-Forward Network (FFN) for channel-wise feature refinement. The shared depthwise convolutions ensure that features are extracted consistently across variables, while the 1-to-1 channel-head binding mechanism allows the attention to selectively transfer information at the feature level.

**Temporal Convolutional Q, K, V Projection.** To generate the Query, Key, and Value embeddings, we use a projection block based on depthwise convolutions (DWConv) (as illustrated in Figure 2b), a technique effectively utilized for time-series analysis in ModernTCN (Luo & Wang, 2024). This design choice leverages the inherent property of CNNs where each channel naturally specializes in capturing distinct patterns.

In our architecture, the weights of the DWConv operators are shared across all variables. This straightforward design choice allows each channel to learn a consistent feature type from every variable, producing the semantically aligned representations required for the subsequent Channel-Head Attention. Moreover, we employ parallel kernels of different sizes for multi-scale analysis. The projections are formally defined as:

$$\begin{aligned} Q_{m,c} &= \text{DWConv}_{\text{large},Q}(Z_{m,c}) + \text{DWConv}_{\text{small},Q}(Z_{m,c}), \\ K_{m,c} &= \text{DWConv}_{\text{large},K}(Z_{m,c}) + \text{DWConv}_{\text{small},K}(Z_{m,c}), \quad \forall m \in \{1, ..., M\}, c \in \{1, ..., C\} \\ V_{m,c} &= \text{DWConv}_{\text{large},V}(Z_{m,c}) + \text{DWConv}_{\text{small},V}(Z_{m,c}) \end{aligned} \tag{3}$$

where each DWConv operator acts on $Z_{m,c} \in \mathbb{R}^{1 \times L}$ for variable $m$ and channel $c$.

**CHead Attention for Cross-Variable Information Transfer.** As shown in Figure 2c, our Channel-Head Attention creates a one-to-one correspondence between CNN channels and attention heads ($n_h = C$), ensuring each head processes a single channel across all variables. This design prevents indiscriminate fusion—instead enabling selective information transfer where each channel independently identifies and transfers relevant patterns across variables.

For each channel $c \in \{1, ..., C\}$, the attention operation is:

$$O_c = \text{Softmax}\left(\frac{Q_c K_c^T}{\sqrt{L}}\right) V_c \tag{4}$$

where $Q_c, K_c, V_c \in \mathbb{R}^{M \times L}$ represent channel $c$'s features across all variables.

The output tensor $O \in \mathbb{R}^{M \times C \times L}$ is constructed by concatenating the individual channel outputs $\{O_1, ..., O_C\}$ along the channel dimension. The refined embedding $Z_{\text{attn}}$, is obtained by applying a pointwise convolution to $O$, followed by layer normalization and residual skip-connection:

$$Z_{\text{attn}} = Z + \text{LayerNorm}(\text{PWConv}(O)) \tag{5}$$

**Convolutional Feed-Forward Network.** Following Channel-Head Attention, we apply a convolutional feed-forward network for channel-wise feature refinement:

$$Z_{\text{out}} = Z_{\text{attn}} + \text{LayerNorm}(\text{PWConv}_2(\text{GeLU}(\text{PWConv}_1(Z_{\text{attn}})))) \tag{6}$$

We use pointwise convolutions rather than linear transformations to preserve the temporal structure inherent in time series data. This design ensures that each temporal position is processed independently while enabling non-linear interactions across channels. The network follows a inverted bottleneck architecture where $PWConv_1$ projects to an intermediate dimension and $PWConv_2$ maps back to the original channel dimension $C$. Through stacked T1 blocks, the FFN-mixed features form new channel representations for subsequent layers, enabling progressive feature combination while CHead Attention maintains feature-level selectivity.

## 4 EXPERIMENTS

In this section, we comprehensively evaluate T1 across various missing data scenarios and benchmark datasets. We conduct three main experiments to demonstrate the effectiveness of our approach: (1) point missing scenario with varying missing ratios, (2) block missing scenario simulating sensor failures, (3) evaluation on naturally occurring missing data. Additionally, we provide detailed representation analysis and ablation studies to better understand the contribution of each component.

### 4.1 EXPERIMENTAL SETUP

**Datasets.** We evaluate on 9 widely-used time series benchmark datasets: ETTh1, ETTh2, ETTm1, ETTm2 (Zhou et al., 2021), Electricity (Trindade, 2015), Weather (Wetterstation), Illness (CDC), Exchange (Lai et al., 2018), and PEMS03 (Chen et al., 2001). Additionally, we use two naturally missing datasets: PhysioNet Challenge 2012 (Silva et al., 2012) and AQI36 (Yi et al., 2016).

**Baselines.** We compare against 11 state-of-the-art methods spanning two categories: (1) *General time series and forecasting models*: TimeMixer++ (Wang et al., 2024), ModernTCN (Luo & Wang, 2024), iTransformer (Liu et al., 2024), TimesNet (Wu et al., 2023), PatchTST (Nie et al., 2023), and DLinear (Zeng et al., 2023); (2) *Specialized imputation models*: ImputeFormer (Nie et al., 2024), SAITS (Du et al., 2023), CSDI (Tashiro et al., 2021), BRITS (Cao et al., 2018), and PSW-I (Wang et al., 2025a). Architecturally, these methods span time-axis tokenization (PatchTST, SAITS), variable-axis tokenization (iTransformer), dual-axis tokenization (ImputeFormer, CSDI), temporal CNN (ModernTCN, TimesNet), RNN-based (BRITS), MLP-based (DLinear), hybrid (TimeMixer++), and optimal transport (PSW-I).

**Implementation Details.** We set the sequence length to 96 for all experiments. During training, we employ self-supervised learning where 40% of observed values are randomly masked and used as reconstruction targets, minimizing MSE loss between predictions and ground truth. For fair comparison, general time series models are trained under identical conditions to T1, while specialized imputation methods retain their original training protocols; all models are evaluated with the same data splits and random seeds. Performance is evaluated using mean absolute error (MAE) and mean squared error (MSE) following previous studies (Liu et al., 2024; Wang et al., 2025a). Full training details and loss formulation are provided in Appendix A.2, and experimental results including standard deviations are in Appendix F.

### 4.2 MAIN RESULTS

#### 4.2.1 POINT MISSING SCENARIO

**Setup.** We test on four different missing ratios (0.1, 0.3, 0.5, 0.7) to assess the robustness of each method under various missing conditions.

Table 1: Imputation performance on nine benchmark datasets under point missing scenario. Results are averaged across four missing ratios (0.1, 0.3, 0.5, 0.7). Best results are marked in **bold** and second best in underlined.

| Dataset | T1 (Ours) | | TimeMixer++ | | ModernTCN | | iTransformer | | TimesNet | | PatchTST | | DLinear | | ImputeFormer | | SAITS | | CSDI | | BRITS | | PSW-I | |
|---|---|---|---|---|---|---|---|---|---|---|---|---|---|---|---|---|---|---|---|---|---|---|---|---|
| | MSE | MAE | MSE | MAE | MSE | MAE | MSE | MAE | MSE | MAE | MSE | MAE | MSE | MAE | MSE | MAE | MSE | MAE | MSE | MAE | MSE | MAE | MSE | MAE |
| ETTh1 | **0.049** | **0.138** | 0.132 | 0.232 | 0.083 | 0.189 | 0.129 | 0.236 | 0.130 | 0.237 | 0.082 | 0.185 | 0.180 | 0.273 | 0.223 | 0.266 | 0.092 | 0.178 | 0.083 | 0.178 | 0.121 | 0.223 | 0.126 | 0.231 |
| ETTh2 | **0.036** | **0.113** | 0.068 | 0.161 | 0.051 | 0.145 | 0.064 | 0.165 | 0.065 | 0.169 | 0.049 | 0.142 | 0.073 | 0.178 | 0.429 | 0.354 | 0.275 | 0.342 | 0.075 | 0.144 | 0.226 | 0.327 | 0.046 | 0.142 |
| ETTm1 | **0.022** | **0.091** | 0.052 | 0.136 | 0.040 | 0.124 | 0.063 | 0.159 | 0.045 | 0.130 | 0.038 | 0.119 | 0.132 | 0.225 | 0.086 | 0.155 | 0.051 | 0.127 | 0.034 | 0.114 | 0.070 | 0.166 | 0.047 | 0.131 |
| ETTm2 | **0.017** | **0.070** | 0.030 | 0.099 | 0.026 | 0.098 | 0.032 | 0.111 | 0.027 | 0.100 | 0.024 | 0.089 | 0.040 | 0.128 | 0.151 | 0.183 | 0.103 | 0.201 | 0.035 | 0.087 | 0.245 | 0.314 | 0.021 | 0.094 |
| Weather | **0.029** | 0.045 | 0.034 | 0.055 | 0.038 | 0.072 | 0.090 | 0.138 | 0.040 | 0.079 | 0.037 | 0.069 | 0.044 | 0.084 | 0.042 | 0.053 | 0.034 | 0.045 | 0.084 | **0.042** | 0.112 | 0.117 | 0.107 | 0.072 |
| PEMS03 | **0.021** | **0.093** | 0.044 | 0.143 | 0.056 | 0.166 | 0.048 | 0.147 | 0.059 | 0.171 | 0.038 | 0.133 | 0.094 | 0.220 | 0.080 | 0.175 | 0.060 | 0.154 | 0.082 | 0.155 | 0.076 | 0.176 | 0.049 | 0.149 |
| Exchange | **0.002** | **0.018** | 0.002 | 0.023 | 0.009 | 0.062 | 0.004 | 0.034 | 0.003 | 0.032 | 0.003 | 0.027 | 0.005 | 0.044 | 0.031 | 0.070 | 0.007 | 0.054 | | | 0.115 | 0.249 | 0.031 | 0.026 |
| Illness | **0.038** | **0.102** | 0.238 | 0.291 | 0.260 | 0.350 | 0.205 | 0.283 | 0.583 | 0.458 | 0.130 | 0.223 | 0.345 | 0.392 | 0.636 | 0.505 | 0.180 | 0.344 | 586.936 | 9.057 | 0.426 | 0.399 | 0.067 | 0.122 |
| Electricity | **0.043** | **0.131** | 0.071 | 0.172 | 0.121 | 0.253 | 0.090 | 0.199 | 0.105 | 0.225 | 0.089 | 0.208 | 0.191 | 0.331 | 0.076 | 0.177 | 0.152 | 0.277 | 0.144 | 0.235 | 0.168 | 0.298 | 0.106 | 0.208 |
| Avg | **0.027** | **0.084** | 0.075 | 0.142 | 0.070 | 0.151 | 0.079 | 0.159 | 0.119 | 0.172 | 0.050 | 0.123 | 0.114 | 0.193 | 0.210 | 0.220 | 0.176 | 0.236 | 73.417 | 1.229 | 0.174 | 0.247 | 0.062 | 0.121 |

Table 2: Performance comparison under varying test-time missing ratios averaged across all datasets. Models are trained with 0.4 missing ratio and evaluated on different missing intensities.

| Missing Ratio | T1 (Ours) | | TimeMixer++ | | ModernTCN | | iTransformer | | TimesNet | | PatchTST | | DLinear | | ImputeFormer | | SAITS | | CSDI | | BRITS | | PSW-I | |
|---|---|---|---|---|---|---|---|---|---|---|---|---|---|---|---|---|---|---|---|---|---|---|---|---|
| | MSE | MAE | MSE | MAE | MSE | MAE | MSE | MAE | MSE | MAE | MSE | MAE | MSE | MAE | MSE | MAE | MSE | MAE | MSE | MAE | MSE | MAE | MSE | MAE |
| 0.1 | **0.017** | **0.070** | 0.055 | 0.129 | 0.063 | 0.153 | 0.057 | 0.141 | 0.089 | 0.158 | 0.040 | 0.116 | 0.138 | 0.233 | 0.098 | 0.150 | 0.104 | 0.189 | 124.217 | 1.452 | 0.080 | 0.165 | 0.048 | 0.111 |
| 0.3 | **0.021** | **0.077** | 0.056 | 0.129 | 0.048 | 0.132 | 0.061 | 0.144 | 0.095 | 0.157 | 0.038 | 0.113 | 0.068 | 0.157 | 0.122 | 0.168 | 0.125 | 0.208 | 75.365 | 1.286 | 0.109 | 0.200 | 0.058 | 0.122 |
| 0.5 | **0.027** | **0.089** | 0.069 | 0.141 | 0.059 | 0.144 | 0.076 | 0.160 | 0.113 | 0.172 | 0.048 | 0.126 | 0.088 | 0.174 | 0.176 | 0.209 | 0.167 | 0.240 | 40.385 | 0.991 | 0.168 | 0.260 | 0.068 | 0.133 |
| 0.7 | **0.049** | **0.121** | 0.118 | 0.184 | 0.135 | 0.220 | 0.128 | 0.210 | 0.173 | 0.225 | 0.092 | 0.176 | 0.198 | 0.270 | 0.384 | 0.335 | 0.299 | 0.324 | 21.136 | 0.745 | 0.336 | 0.384 | 0.093 | 0.157 |

**Results.** As shown in Table 1, T1 demonstrates superior performance across all datasets. On average, T1 achieves a 46% MSE reduction compared to the next best PatchTST baseline and a 56% reduction against the specialized imputer PSW-I. Table 2 further highlights T1's robustness against increasing data sparsity. At the highest missing ratio of 0.7, where many baselines struggle, T1's MSE is nearly half that of the next best methods, PatchTST (0.049 vs. 0.092), underscoring its resilience in scenarios with severe data loss.

### 4.2.2 BLOCK MISSING SCENARIO

**Setup.** To simulate realistic sensor failure scenarios, we introduce two types of missing patterns at test time: (1) 5% probability of point missing for random measurement noise, and (2) 0.15% probability of consecutive block missing with random lengths between 24 to 96 time steps for temporary sensor failures or communication interruptions.

Table 3: Imputation performance under block missing scenario simulating realistic sensor failures. Test patterns combine 5% point missing and 0.15% block missing (24-96 consecutive timesteps).

| Dataset | T1 (Ours) | | TimeMixer++ | | ModernTCN | | iTransformer | | TimesNet | | PatchTST | | DLinear | | ImputeFormer | | SAITS | | CSDI | | BRITS | |
|---|---|---|---|---|---|---|---|---|---|---|---|---|---|---|---|---|---|---|---|---|---|---|---|
| | MSE | MAE | MSE | MAE | MSE | MAE | MSE | MAE | MSE | MAE | MSE | MAE | MSE | MAE | MSE | MAE | MSE | MAE | MSE | MAE | MSE | MAE |
| ETTh1 | 0.030 | **0.107** | 0.105 | 0.210 | 0.066 | 0.172 | 0.094 | 0.205 | 0.104 | 0.217 | 0.050 | 0.151 | 0.192 | 0.299 | 0.063 | 0.156 | **0.028** | 0.109 | 0.037 | 0.127 | 0.056 | 0.145 |
| ETTh2 | **0.027** | **0.092** | 0.062 | 0.153 | 0.048 | 0.138 | 0.060 | 0.152 | 0.055 | 0.156 | 0.039 | 0.125 | 0.078 | 0.184 | 0.179 | 0.228 | 0.145 | 0.260 | 0.074 | 0.112 | 0.133 | 0.250 |
| ETTm1 | 0.030 | 0.082 | 0.062 | 0.131 | 0.044 | 0.115 | 0.070 | 0.145 | 0.043 | 0.118 | 0.037 | 0.103 | 0.202 | 0.285 | 0.036 | 0.111 | 0.022 | 0.087 | 0.023 | 0.092 | 0.026 | 0.099 |
| ETTm2 | **0.016** | **0.059** | 0.029 | 0.094 | 0.024 | 0.090 | 0.028 | 0.099 | 0.028 | 0.099 | 0.024 | 0.081 | 0.047 | 0.141 | 0.118 | 0.144 | 0.075 | 0.164 | 0.048 | 0.070 | 0.082 | 0.181 |
| Weather | **0.026** | 0.039 | 0.032 | 0.054 | 0.040 | 0.085 | 0.092 | 0.140 | 0.040 | 0.086 | 0.035 | 0.068 | 0.050 | 0.106 | 0.040 | 0.048 | **0.026** | 0.030 | 0.086 | 0.036 | 0.035 | 0.039 |
| PEMS03 | **0.022** | **0.084** | 0.050 | 0.144 | 0.065 | 0.180 | 0.053 | 0.152 | 0.061 | 0.174 | 0.044 | 0.132 | 0.166 | 0.307 | 0.031 | 0.103 | 0.049 | 0.131 | 0.178 | 0.143 | 0.048 | 0.127 |
| Exchange | 0.003 | 0.017 | 0.002 | 0.021 | 0.006 | 0.047 | 0.004 | 0.031 | 0.003 | 0.031 | 0.004 | 0.026 | 0.009 | 0.056 | 0.034 | 0.059 | 0.105 | 0.279 | 0.037 | 0.064 | 0.041 | 0.120 |
| Illness | **0.037** | **0.089** | 0.230 | 0.280 | 0.263 | 0.397 | 0.158 | 0.237 | 0.418 | 0.384 | 0.125 | 0.224 | 0.518 | 0.533 | 0.468 | 0.433 | 0.389 | 0.401 | 1182. | 11.52 | 0.236 | 0.292 |
| Electricity | **0.038** | **0.118** | 0.088 | 0.180 | 0.146 | 0.283 | 0.080 | 0.190 | 0.099 | 0.212 | 0.090 | 0.208 | 0.302 | 0.444 | 0.061 | 0.152 | 0.135 | 0.262 | 0.133 | 0.220 | 0.117 | 0.244 |
| Avg | **0.026** | **0.076** | 0.073 | 0.141 | 0.078 | 0.167 | 0.071 | 0.150 | 0.094 | 0.164 | 0.050 | 0.124 | 0.174 | 0.262 | 0.114 | 0.159 | 0.108 | 0.191 | 131.4 | 1.376 | 0.086 | 0.166 |

**Results.** T1's strong performance continues in the more challenging block missing scenario. As shown in Table 3, T1 outperforms the next best method, PatchTST, with a 48% reduction in average MSE. This result underscores the effectiveness of T1's cross-variable information transfer when long segments of temporal information are unavailable.

### 4.2.3 NATURAL MISSING DATASET

**Setup.** We evaluate on two datasets with naturally occurring missing values using different protocols:

- **PhysioNet Challenge 2012** contains multivariate clinical time series from 4,000 ICU patients with 37 physiological variables and approximately 80% inherent missing values. We add artificial missing patterns (0.1, 0.3, 0.5, 0.7) on top of existing missing values, creating compound missing scenarios with up to 94% total missing rate.

- **AQI36** consists of air quality measurements from 36 monitoring stations with 15-30% natural missing values due to sensor malfunctions. We evaluate directly on the test set's natural missing patterns without additional masking.

**Results.** Under real-world conditions with naturally occurring missing data, T1 proves its practical applicability. On the PhysioNet2012 dataset, T1 demonstrates remarkable stability and achieves a 23% performance improvement in average MSE over the next best method, DLinear (Table 4).

This robustness is also demonstrated on the AQI36 dataset, where T1 outperforms the next best method, PatchTST, with a 13% reduction in MSE. These results confirm the robustness of our architecture across diverse and critically sparse data regimes.

Table 4: Performance on naturally missing datasets. PhysioNet2012: compound missing with 80% inherent + additional masking. AQI36: evaluation on natural test set missing patterns (15-30%).

| | PhysioNet2012 - Natural ( 80% ) + Additional Missing | | | | | | | | | | | | | | | | | | |
|---|---|---|---|---|---|---|---|---|---|---|---|---|---|---|---|---|---|---|---|
| Additional | **T1 (Ours)** | | TimeMixer++ | | ModernTCN | | iTransformer | | TimesNet | | PatchTST | | DLinear | | ImputeFormer | | SAITS | |
| Missing Ratio | MSE | MAE | MSE | MAE | MSE | MAE | MSE | MAE | MSE | MAE | MSE | MAE | MSE | MAE | MSE | MAE | MSE | MAE |
| 0.1 (Total: 82%) | **0.049** | **0.067** | 0.091 | 0.107 | 0.092 | 0.118 | 0.107 | 0.128 | 0.082 | 0.103 | 0.099 | 0.116 | 0.075 | 0.100 | 0.078 | **0.067** | 0.089 | 0.070 |
| 0.3 (Total: 86%) | **0.064** | 0.077 | 0.372 | 0.111 | 0.103 | 0.121 | 0.122 | 0.129 | 0.096 | 0.108 | 0.106 | 0.120 | 0.093 | 0.108 | 0.090 | **0.075** | 0.088 | 0.078 |
| 0.5 (Total: 90%) | **0.081** | 0.090 | 0.130 | 0.117 | 0.110 | 0.126 | 0.120 | 0.131 | 0.101 | 0.116 | 0.113 | 0.125 | 0.104 | 0.117 | 0.114 | 0.091 | 0.107 | **0.089** |
| 0.7 (Total: 94%) | **0.106** | 0.110 | 0.236 | 0.126 | 0.124 | 0.134 | 0.129 | 0.135 | 0.114 | 0.127 | 0.124 | 0.132 | 0.118 | 0.127 | 0.151 | 0.114 | 0.129 | **0.106** |
| Avg | **0.075** | **0.086** | 0.207 | 0.115 | 0.107 | 0.125 | 0.119 | 0.131 | 0.098 | 0.114 | 0.110 | 0.123 | 0.097 | 0.113 | 0.108 | 0.087 | 0.103 | 0.086 |
| | AQI36 - Natural Missing Only (15-30%) | | | | | | | | | | | | | | | | | |
| Test Set | **0.226** | **0.226** | 0.274 | 0.318 | 0.281 | 0.311 | 0.314 | 0.331 | 0.337 | 0.337 | 0.262 | 0.303 | 0.338 | 0.343 | 0.447 | 0.411 | 0.469 | 0.400 |

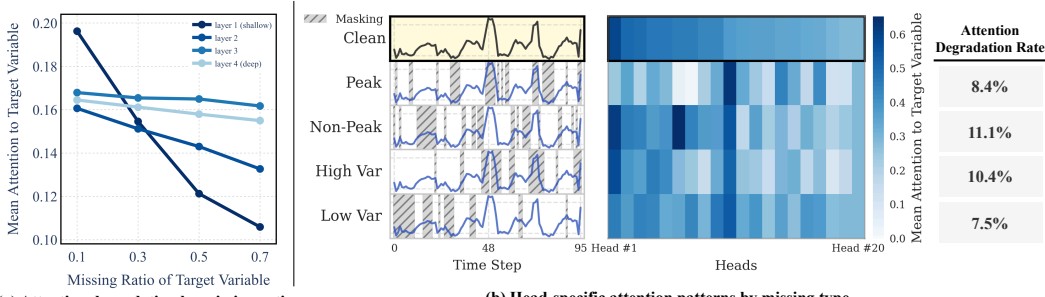

(a) Attention degradation by missing ratio  (b) Head-specific attention patterns by missing type

Figure 3: Representation analysis of T1's attention mechanism. (a) Layer-wise attention weights from other variables to target variable under varying missing ratios (entire ETTh1 test set). Attention weights decrease with increasing missing ratio, with shallow layers showing more pronounced degradation. (b) Head-specific attention patterns of clean signal and under various missing patterns (peak vs non-peak and high vs low variance, 30% each), showing top-20 heads sorted by clean attention weights.

### 4.2.4 REPRESENTATION ANALYSIS

We conduct two controlled experiments on ETTh1 to qualitatively analyze the effectiveness of CHead Attention.

**Missing Response Across Layers.** Using the entire ETTh1 test set, we select one variable as target and vary its missing ratio from 0.1 to 0.7 while keeping the missing ratio of all other variables at 0.4. Figure 3a shows attention weights assigned to the target variable decrease with increasing missing ratio. This trend is most noticeable in the shallow layer while deeper layers exhibit reduced sensitivity to missingness. Attention weights in the first layer exhibit sharp drop of 46% (0.195→0.105) while weights in the last layer drop by only 6% (0.165→0.155). This suggests partial reconstruction in early layers improves information availability for subsequent layers.

**Observable Pattern Dependence.** Using a single ETTh1 test sample, we mask 30% of the target variable in regions with different characteristics: peak regions (far from center) versus non-peak regions (near center), and regions with top 30% versus bottom 30% local variance. As shown in left panel of Figure 3b, these masks leave fundamentally different temporal patterns in the observed portion of the target variable. The middle panel of Figure 3b visualizes the corresponding attention responses for the top-20 heads, sorted by clean attention weights. Clearly, this visualization reveals distinct response patterns for each masking scenario. Quantitatively, removing high-variance regions reduces attention by 10.4% while removing low-variance regions reduces it by 7.5%. This indicates that attention modulation depends on which temporal patterns remain observable, not solely on missing ratio. CHead Attention enables each channel to assess whether its corresponding temporal features can be extracted from the observed data.

These results demonstrate that T1 learns to adaptively down-weight unreliable information pathways based on both observation density and the extractability of temporal patterns. The layer-wise stabilization and channel-specific responses support our architectural design combining CNN feature extraction with channel-bound attention, contributing to the performance gains observed under structured missingness (Table 3).

### 4.2.5 ABLATION STUDY

We conduct comprehensive ablation studies to analyze the contribution of each component in T1. All experiments are performed on six datasets (ETTh1, ETTh2, ETTm1, ETTm2, Weather, Electricity) with 40% training mask ratio and evaluated across four test missing ratios (0.1, 0.3, 0.5, 0.7). Table 5 reports averaged results when replacing only the specified component while keeping all others at their default configuration.

**Cross-variable Mechanism.** Replacing attention with pointwise convolution degrades performance by 12.91%, demonstrating that adaptive information transfer outperforms fixed patterns. Removing cross-variable modeling entirely results in 56.16% degradation, confirming that cross-variable information is essential for imputation.

**Channel-Head Binding.** We evaluate the impact of channel-head grouping by varying the number of channels per attention head: 8, 16, and 32 channels per head (compared to our default one-to-one correspondence with 128 channels). Performance degrades by 7.45%, 16.86%, and 14.57% respectively, with 16 channels per head showing the worst degradation. These results confirm that fine-grained, one-to-one channel-head correspondence is crucial for maintaining feature-specific information pathways and preventing the mixing of corrupted and reliable temporal patterns during cross-variable transfer.

**Mask-Aware Embedding.** Removing the explicit mask channel from input embedding causes 3.64% degradation. This indicates that providing missing patterns directly to the model improves its ability to distinguish between observed and missing values during feature extraction.

**Reconstruction Method.** PixelShuffle outperforms linear upsampling by 3.19%, validating our choice for artifact-free temporal reconstruction.

The substantial gap between convolution (12.91%) and no cross-variable modeling (56.16%) reveals an important finding: while cross-variable information is crucial, the method of information transfer matters significantly. Our attention mechanism better identifies which variables contain reliable information for imputation compared to fixed convolutional patterns.

Table 5: Comprehensive ablation study on model components (MSE). Each row shows the performance when replacing only the specified component from our full model. The last column shows the percentage increase in error relative to our full model.

| Component | Alternative | ETTh1 | ETTh2 | ETTm1 | ETTm2 | Weather | ECL | Avg | Δ (%)↓ |
|---|---|---|---|---|---|---|---|---|---|
| **T1 (Ours)** | | **0.049** | **0.036** | **0.022** | **0.017** | **0.029** | **0.043** | **0.033** | - |
| Cross-variable | Conv | 0.056 | 0.040 | 0.024 | 0.020 | 0.029 | 0.052 | 0.037 | + 12.91 |
| Component | w/o | 0.095 | 0.064 | 0.040 | 0.029 | 0.031 | 0.048 | 0.051 | + 56.16 |
| Channel-Head | 32 Chns | 0.061 | 0.040 | 0.030 | 0.020 | 0.030 | 0.044 | 0.037 | + 14.57 |
| Binding | 16 Chns | 0.066 | 0.041 | 0.028 | 0.020 | 0.030 | 0.045 | 0.038 | + 16.86 |
| | 8 Chns | 0.055 | 0.038 | 0.025 | 0.019 | 0.030 | 0.044 | 0.035 | + 7.45 |
| Embedding | w/o mask | 0.052 | 0.037 | 0.023 | 0.018 | 0.029 | 0.044 | 0.034 | + 3.64 |
| Reconstruction | Linear | 0.050 | 0.036 | 0.022 | 0.018 | 0.030 | 0.046 | 0.034 | + 3.19 |

## 5 CONCLUSION AND FUTURE WORK

In this paper, we presented T1, a CNN-Transformer hybrid architecture for multivariate time series imputation. By strategically assigning CNNs for temporal feature extraction and attention for cross-variable information transfer, T1 addresses the fundamental challenge of imputation under heavy missingness. Our key innovation, Channel-Head Binding, creates one-to-one correspondences between CNN channels and attention heads, enabling feature-specific information pathways that adapt to varying missingness patterns. Extensive experiments demonstrate that T1 maintains computational efficiency while achieving state-of-the-art performance across diverse datasets and missing scenarios. The architecture's robustness under extreme missing conditions and its stable performance with a consistent hyperparameter configuration highlight its practical applicability. Looking forward, we will explore extensions to online streaming environments for real-time imputation and active sensing strategies that can guide optimal sensor selection under resource constraints.

ACKNOWLEDGMENTS

This work was supported in part by Institute of Information communications Technology Planning Evaluation (IITP) grant funded by the Korea government (MSIT) (RS-2025-25463302), in part by the National Research Foundation (NRF) of Korea grant funded by the Korea government (MSIT) (No. RS-2023-00222663), and in part by AI-Bio Research Grant through Seoul National University.

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

# A  IMPLEMENTATION DETAILS

## A.1  DATASET DETAILS

### A.1.1  DATASET DESCRIPTIONS

We conduct experiments on 11 multivariate time series datasets spanning diverse domains including energy, transportation, climate, healthcare, and economics. All experiments use a sequence length of 96 timesteps, except for PhysioNet2012 which uses 48 timesteps due to its clinical nature and irregular sampling patterns. The datasets are categorized into two groups: complete datasets for artificial missing experiments and naturally missing datasets for realistic evaluation scenarios. Table 6 summarizes the key statistics.

**Complete Datasets** ETT (Zhou et al., 2021) comprise electricity transformer measurements including with hourly (ETTh1, ETTh2) and 15-minute (ETTm1, ETTm2) sampling frequencies. Electricity (Trindade, 2015)tracks consumer power consumption. Weather (Wetterstation) contains meteorological indicators from the Max Planck Institute weather station. Illness (CDC) records CDC influenza surveillance data across US states. Exchange (Lai et al., 2018)covers international currency rates from 1990-2016. PEMS03 (Chen et al., 2001) represents highway traffic sensor measurements from California transportation networks.

**Naturally Missing Datasets** PhysioNet Challenge 2012 (Silva et al., 2012) contains ICU patient physiological measurements with 80% inherent missingness due to irregular clinical sampling protocols. Experiments are conducted on the 20% observed portions. AQI36 (Yi et al., 2016) includes air quality monitoring data with 13.3% real missingness from sensor failures: general missing (8.2%) from random transmission errors, spatial block missing (2.2%) from regional power/network outages, and temporal block missing (3.5%, 11 timesteps average block length ) from maintenance periods. These datasets span diverse domains and temporal scales, providing comprehensive evaluation under varying missingness scenarios from dense sensor networks to sparse clinical measurements.

Table 6: Dataset descriptions.

| Type | Dataset | Variables | Train | Valid | Test | Frequency | Missing Ratio |
|------|---------|-----------|-------|-------|------|-----------|---------------|
| Complete | ETTh1,ETTh2 | 7 | 8,545 | 2,785 | 2,785 | Hourly | - |
| | ETTm1,ETTm2 | 7 | 34,465 | 11,425 | 11,425 | 15min | - |
| | Electricity | 321 | 18,346 | 2,621 | 5,424 | Hourly | - |
| | Weather | 21 | 36,820 | 5,260 | 10,521 | 10min | - |
| | Illness | 7 | 609 | 87 | 175 | Weekly | - |
| | Exchange | 8 | 5,245 | 749 | 1,499 | Daily | - |
| | PEMS03 | 358 | 18,279 | 2,611 | 5,223 | 5min | - |
| Naturally Missing | PhysioNet2012 | 37 | 2,557 | 640 | 800 | Irregular | 80.0% |
| | AQI36 | 36 | 4,422 | 649 | 2,548 | Hourly | 13.3% |

## A.2  EXPERIMENT DETAILS

### A.2.1  T1 CONFIGURATION DETAILS

We maintain consistent architectural design across different datasets, with deterministic sequence-length scaling for kernel sizes when sequence lengths differ from the standard 96 timesteps. Importantly, we use the same model configuration (channel count, layer depth, FFN ratio) regardless of the number of variables in each dataset, demonstrating the model's robustness across varying data dimensions.

**Standard Configuration.** For datasets with sequence length 96, Conv1D embedding with kernel size 2 and stride 1 projects input to 128 channels. The architecture consists of four T1 blocks arranged in two hierarchical groups. The first group contains two T1 blocks employing dual-scale depthwise convolutions with kernel sizes 71 and 5, followed by downsampling with kernel size 2 and stride 2. The second group contains two T1 blocks with adjusted kernel sizes 31 and 5, operating on downsampled features. This hierarchical design allows the model to capture multi-scale temporal

patterns at different resolutions. FFN expansion ratio is set to 1.0. This configuration remains fixed across all datasets, from 7-variable datasets (ETT series) to 358-variable datasets (PEMS03).

**Sequence Length Adaptation.** For datasets with different sequence lengths (e.g., PhysioNet with 48 timesteps), we apply deterministic scaling to adjust kernel sizes:

$$k_{\text{adjusted}} = \lfloor (T/96) \times k_{\text{default}} \rfloor \tag{7}$$

where $T$ is the sequence length. This yields $71 \rightarrow 35$ and $31 \rightarrow 15$ for the large kernels, while small kernels (size 5) remain unchanged. This systematic rule preserves proportional receptive field coverage without dataset-specific tuning. All other parameters remain identical to the standard configuration.

### A.2.2 EXPERIMENT DESIGN

All experiments use five random seeds (102, 202, 302, 402, 502) with mean and standard deviation reported. Experiments were performed on NVIDIA H100 80GB GPUs.

We evaluate models across three missing scenarios to assess generalization capability. Point missing applies independent probability masking at each timestep with varying ratios (10%, 30%, 50%, 70%). Block missing simulates realistic sensor failures by combining 5% point missing with 0.15% probability of initiating consecutive missing blocks spanning 24-96 timesteps. The key experimental principle is training with specific missing ratios and evaluating across multiple missing scenarios.

**Complete Datasets** T1 uses 0.4 point-wise random masking for training in both point missing and block missing experiments. This single trained model is evaluated across multiple test scenarios: point missing experiments test on ratios of 0.1, 0.3, 0.5, and 0.7 with point-wise patterns, while block missing experiments test on the complex block patterns described above. This design directly tests whether models trained on simple point patterns can generalize to more complex structured missing without specific training.

**Naturally Missing Datasets** We apply additional artificial missing on top of inherent missing patterns for imputation training and evaluation. PhysioNet2012 models train with 0.2 point-wise random masking applied to non-missing values, then test on various missing ratios (0.1, 0.3, 0.5, 0.7) applied to non-missing regions. AQI36 models train using real-pattern based artificial missing augmented with additional random point-wise masking ratios (0.2, 0.5, 0.8) sampled per batch, while testing uses exclusively the dataset's provided real-pattern based artificial missing patterns.

### A.2.3 EVALUATION METRICS

We employ Mean Squared Error (MSE) and Mean Absolute Error (MAE) as primary evaluation metrics for imputation performance:

$$\text{MSE} = \frac{1}{|\mathcal{M}|} \sum_{(m,t)\in\mathcal{M}} (\hat{x}_t^{(m)} - y_t^{(m)})^2, \quad \text{MAE} = \frac{1}{|\mathcal{M}|} \sum_{(m,t)\in\mathcal{M}} |\hat{x}_t^{(m)} - y_t^{(m)}| \tag{7}$$

where $\mathcal{M}$ denotes the set of artificially masked positions during evaluation, $y_t^{(m)}$ represents ground truth values, and $\hat{x}_t^{(m)}$ represents imputed values. Metrics are computed only on artificially masked positions, not on originally missing values, ensuring consistent evaluation across all methods.

### A.2.4 TRAINING IMPLEMENTATION

We employ a self-supervised training strategy where observed values are artificially masked during training and the loss is computed only on these masked positions. We distinguish between the original observation mask $\Omega \in \{0,1\}^{M \times T}$ where 1 indicates observed values and 0 indicates missing values, and the training mask $\Psi \in \{0,1\}^{M \times T}$ where 0 indicates artificially masked positions for training. The model minimizes Mean Squared Error between predictions $\hat{x}_t^{(m)}$ and ground truth $y_t^{(m)}$ at artificially masked locations:

$$\mathcal{L}_{\text{MSE}} = \frac{1}{\sum_{m,t} I(\Psi_t^{(m)} = 0)} \sum_{\Psi_t^{(m)}=0} (\hat{x}_t^{(m)} - y_t^{(m)})^2 \tag{8}$$

This approach ensures the model learns to reconstruct values from partial observations without using originally missing data as supervision. We use the Adam optimizer with $\beta_1 = 0.9$ and $\beta_2 = 0.999$, learning rate of 0.001 (0.0001 for Weather due to rapid convergence), batch size of 16, and maximum 300 epochs with early stopping patience of 30.

### A.3 BASELINE IMPLEMENTATION DETAILS

We evaluate two categories of baseline models with distinct configuration strategies to ensure fair and comprehensive comparison. All baseline implementations are based on established frameworks including Time-Series Library, PyPOTS (Du et al., 2025), and Awesome-Imputation (Du et al., 2024) repositories to ensure reproducibility and fair comparison.

**General and Forecasting Time Series Models** TimeMixer++ (Wang et al., 2024), ModernTCN (Luo & Wang, 2024), iTransformer (Liu et al., 2024), TimesNet (Wu et al., 2023), PatchTST (Nie et al., 2023), and DLinear (Zeng et al., 2023) adopt identical training protocols to T1, using 0.4 point-wise random masking during training. MSE loss computed only on masked positions, Adam optimizer with learning rate 0.001, batch size 16, and maximum 300 epochs with early stopping (patience=30). This standardization isolates architectural differences from training strategies. Model architectures follow hierarchical selection priority: official imputation configurations for specific datasets when available, configurations from similar variable count imputation tasks, long-term forecasting configurations for the same dataset, or forecasting configurations from datasets with similar variable counts.

**Specialized Imputation Models** ImputeFormer (Nie et al., 2024), SAITS (Du et al., 2023), CSDI (Tashiro et al., 2021), and BRITS (Cao et al., 2018) retain their published training protocols to leverage model-specific capabilities. These models employ original loss functions (such as CSDI's diffusion loss and BRITS's consistency loss), published optimization schedules, model-specific missing pattern strategies, and architecture-specific parameters from official implementations. When exact configurations were unavailable, the same hierarchical priority was applied while preserving each model's unique training methodology. Both model categories adapt to natural missing experiments with PhysioNet2012 training using 0.2 point-wise masking on non-missing values, while AQI36 follows the T1 protocol with real-pattern based missing augmentation.

## B EFFICIENCY ANALYSIS

**Comparison with Baseline Methods.** Table 7 presents computational efficiency and performance metrics across T1, DLinear (Zeng et al., 2023), ModernTCN (Luo & Wang, 2024), iTransformer (Liu et al., 2024), TimesNet (Wu et al., 2023), PatchTST (Nie et al., 2023), TimeMixer++ (Wang et al., 2024), SAITS (Du et al., 2023), ImputeFormer (Nie et al., 2024), CSDI (Tashiro et al., 2021). T1 achieves the best imputation performance on both ETTh1 and Weather datasets while maintaining reasonable computational requirements. The comparison reveals significant variations in resource consumption across models, with methods like CSDI and TimeMixer++ requiring substantially higher computational complexity, while lightweight approaches like DLinear sacrifice accuracy for speed. T1 demonstrates an effective balance between performance quality and computational efficiency, making it suitable for practical deployment scenarios where both accuracy and resource constraints are important considerations.

---

https://github.com/thuml/Time-Series-Library

Table 7: Computational efficiency and performance comparison on ETTh1 and Weather datasets. Params (M): parameters in millions; Memory : inference memory; GFLOPs: computational complexity; Train Speed: ms per iteration; MSE: Mean Squared Error (lower is better).

| Dataset | Model | Params (M) | Memory | GFLOPs | Train Speed (ms/iter) | MSE |
|---|---|---|---|---|---|---|
| ETTh1 | **T1 (Ours)** | **0.543** | **356.45** | **0.156** | **29.84** | **0.049** |
| | DLinear | 0.024 | 22.36 | 0.003 | 10.04 | 0.18 |
| | ModernTCN | 1.716 | 120.99 | 0.039 | 13.7 | 0.083 |
| | iTransformer | 0.223 | 22.71 | 0.003 | 13.95 | 0.129 |
| | TimesNet | 0.588 | 157.78 | 0.176 | 39.18 | 0.13 |
| | PatchTST | 2.185 | 2571.3 | 10.042 | 89.46 | 0.082 |
| | TimeMixer++ | 2.357 | 437.84 | 6.235 | 158.13 | 0.132 |
| | SAITS | 5.273 | 294.49 | 0.506 | 37.01 | 0.092 |
| | ImputeFormer | 1.368 | 1060.11 | 0.645 | 34.49 | 0.223 |
| | CSDI | 1.195 | 777.71 | 19.045 | 154.45 | 0.083 |
| Weather | **T1 (Ours)** | **0.715** | **793.49** | **0.467** | **34.37** | **0.029** |
| | DLinear | 0.051 | 29.07 | 0.008 | 7.4 | 0.044 |
| | ModernTCN | 2.598 | 316.17 | 0.125 | 11.8 | 0.038 |
| | iTransformer | 4.827 | 119.79 | 0.203 | 13.97 | 0.09 |
| | TimesNet | 4.698 | 224.82 | 1.35 | 34.59 | 0.04 |
| | PatchTST | 0.455 | 443.88 | 0.48 | 21.56 | 0.037 |
| | TimeMixer++ | 2.357 | 1000.74 | 18.705 | 205.87 | 0.034 |
| | SAITS | 5.297 | 296.32 | 0.509 | 34.59 | 0.034 |
| | ImputeFormer | 1.551 | 1948.48 | 1.936 | 52.97 | 0.042 |
| | CSDI | 0.326 | 1122.19 | 18.238 | 109.6 | 0.084 |

**Computational Overhead of Channel-Head Binding.** We clarify that Channel-Head Binding incurs no additional computational overhead compared to standard Multi-Head Attention (MHA) when the total representation capacity is fixed. Let $M$ denote the number of variables, $C$ the number of channels, and $L$ the latent temporal dimension. In T1, each of the $C$ heads processes a single channel with feature dimension $L$, yielding complexity $O(M^2 \cdot C \cdot L)$. Standard MHA with fewer heads achieves the same total complexity by increasing the per-head dimension proportionally.

To empirically validate this, we measured FLOPs and GPU memory usage across three datasets with varying variable counts (Table 8). We compare T1's 1-to-1 binding ($n_{\text{heads}} = 128$) against grouped-head variants ($n_{\text{heads}} \in \{4, 8, 16\}$) while keeping the total channel count fixed at 128.

The results confirm that FLOPs remain identical across all configurations, as theoretically expected. For memory usage, T1 shows a minor increase on smaller datasets (1–4%) due to maintaining separate head computations. However, on the large-scale Electricity dataset ($M = 321$), T1 consumes approximately 7.5% less memory than grouped-head variants, as the simplified channel-wise operations avoid the overhead of reshaping and managing grouped head dimensions. This suggests that the 1-to-1 binding becomes increasingly efficient as the number of variables grows.

Table 8: Computational overhead comparison across different head configurations. All variants use 128 total channels.

| Dataset | Model | Heads | Channels | FLOPs | Memory (MB) |
|---|---|---|---|---|---|
| ETTh1 ($M = 7$) | T1 (Ours) | 128 | 128 | 155.6 M | 283.2 |
| | Grouped (Base) | 4 | 128 | 155.6 M | 280.8 |
| | Grouped | 8 | 128 | 155.6 M | 280.0 |
| | Grouped | 16 | 128 | 155.6 M | 279.8 |
| Weather ($M = 21$) | T1 (Ours) | 128 | 128 | 466.9 M | 741.1 |
| | Grouped (Base) | 4 | 128 | 466.9 M | 712.0 |
| | Grouped | 8 | 128 | 466.9 M | 709.7 |
| | Grouped | 16 | 128 | 466.9 M | 709.2 |
| Electricity ($M = 321$) | T1 (Ours) | 128 | 128 | 23.8 G | 18,567 |
| | Grouped (Base) | 4 | 128 | 23.8 G | 20,071 |
| | Grouped | 8 | 128 | 23.8 G | 20,010 |
| | Grouped | 16 | 128 | 23.8 G | 20,026 |

## C    HYPERPARAMETER SENSITIVITY

We evaluate the sensitivity of T1 to key hyperparameters: the number of attention heads (corresponding to channel dimension C), convolutional kernel size, and FFN expansion ratio. All models are trained with 40% missing ratio and evaluated on test sets with varying missingness (10%, 30%, 50%, 70%). Results show averaged performance across these test conditions on ETT, Weather, and Electricity datasets in Figure 4. T1 demonstrates robust performance across all tested configurations.

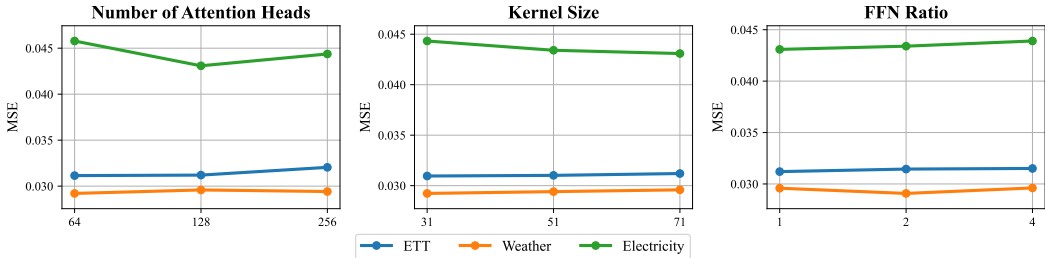

Figure 4: Hyperparameter Sensitivity analysis with respect to the number of heads, FFN ratio, and kernel size.

The model shows minimal sensitivity to variations in the number of heads (64, 128, 256), kernel sizes (31, 51, 71), and FFN expansion ratios (1, 2, 4) across all datasets. Among these, C=128 provides a reasonable balance across diverse datasets and missing ratios, which motivates our default configuration. This stability suggests that T1's Channel-Head Binding mechanism and architectural constraints provide natural regularization, making the model less dependent on precise hyperparameter tuning while maintaining consistent imputation quality across diverse datasets and missing ratios.

Table 9: Detailed numerical results for the hyperparameter sensitivity analysis on the number of attention heads under varying missing ratios (0.1, 0.3, 0.5, 0.7).

| Number of Heads | | 64 | | 128 | | 256 | |
| --- | --- | --- | --- | --- | --- | --- | --- |
| | Metric | MSE | MAE | MSE | MAE | MSE | MAE |
| ETTh1 | 0.1 | **0.023** | **0.102** | 0.025 | 0.104 | 0.027 | 0.110 |
| | 0.3 | **0.032** | **0.116** | 0.033 | 0.118 | 0.036 | 0.124 |
| | 0.5 | **0.047** | **0.139** | 0.048 | 0.140 | 0.050 | 0.144 |
| | 0.7 | 0.095 | 0.196 | **0.091** | **0.192** | 0.093 | 0.194 |
| | Avg | **0.049** | **0.138** | **0.049** | 0.139 | 0.052 | 0.143 |
| ETTh2 | 0.1 | **0.023** | **0.088** | 0.024 | 0.089 | 0.024 | 0.091 |
| | 0.3 | **0.028** | **0.099** | 0.029 | 0.100 | 0.029 | 0.102 |
| | 0.5 | **0.036** | **0.115** | 0.037 | 0.116 | 0.038 | 0.117 |
| | 0.7 | **0.054** | **0.145** | 0.055 | 0.145 | 0.055 | 0.146 |
| | Avg | **0.035** | **0.111** | 0.036 | 0.112 | 0.037 | 0.114 |
| ETTm1 | 0.1 | **0.013** | 0.074 | **0.013** | **0.073** | **0.013** | 0.075 |
| | 0.3 | **0.016** | 0.081 | **0.016** | **0.080** | 0.017 | 0.082 |
| | 0.5 | 0.022 | 0.093 | **0.021** | **0.092** | 0.022 | 0.092 |
| | 0.7 | 0.039 | 0.123 | **0.037** | **0.119** | **0.037** | 0.120 |
| | Avg | **0.022** | 0.093 | **0.022** | **0.091** | **0.022** | 0.092 |
| ETTm2 | 0.1 | 0.012 | 0.057 | **0.011** | **0.055** | 0.012 | 0.056 |
| | 0.3 | **0.014** | 0.064 | **0.014** | **0.062** | **0.014** | 0.063 |
| | 0.5 | **0.018** | 0.074 | **0.018** | **0.073** | **0.018** | 0.073 |
| | 0.7 | 0.027 | 0.092 | **0.026** | **0.091** | **0.026** | 0.092 |
| | Avg | 0.018 | 0.072 | **0.017** | **0.070** | 0.018 | 0.071 |
| Weather | 0.1 | 0.023 | 0.034 | **0.022** | **0.033** | 0.023 | 0.034 |
| | 0.3 | **0.025** | 0.037 | **0.025** | **0.036** | **0.025** | 0.037 |
| | 0.5 | 0.029 | 0.043 | **0.028** | **0.042** | 0.029 | 0.043 |
| | 0.7 | **0.040** | **0.063** | 0.041 | 0.068 | 0.041 | 0.065 |
| | Avg | **0.029** | **0.044** | **0.029** | 0.045 | **0.029** | 0.045 |
| Electricity | 0.1 | 0.033 | 0.117 | 0.032 | 0.114 | **0.031** | **0.111** |
| | 0.3 | 0.037 | 0.123 | 0.037 | 0.121 | **0.036** | **0.119** |
| | 0.5 | 0.045 | 0.136 | 0.045 | 0.134 | **0.043** | **0.132** |
| | 0.7 | 0.069 | 0.174 | 0.070 | 0.174 | **0.068** | **0.172** |
| | Avg | 0.046 | 0.138 | 0.046 | 0.136 | **0.044** | **0.133** |
| Overall Average | | **0.033** | **0.099** | **0.033** | **0.099** | 0.034 | 0.100 |

Table 9 details the sensitivity analysis for the number of attention heads ($n_h$), which determines channel capacity ($n_h = C$) under our binding mechanism. Results indicate that $n_h = 128$ provides a favorable balance that ensures robust generalization across diverse data scales and missing ratios. While a larger capacity ($n_h = 256$) yields marginal benefits on high-dimensional datasets such as Electricity, it compromises stability on smaller ones. Conversely, a reduced capacity ($n_h = 64$) limits the representation of fine-grained series such as the ETTm datasets. This evidence justifies our choice of 128 as a universal default and confirms that T1 achieves robust imputation independent of dataset-specific parameter tuning.

## D   ADDITIONAL EXPERIMENTS AND ANALYSIS

### D.1   IMPACT OF HEAD SCALING

To distinguish the contribution of the Channel-Head Binding mechanism from the effect of simply increasing the attention head count, we performed a scalability analysis using iTransformer as a baseline. We examined whether augmenting the number of heads $n_{heads}$ in standard Multi-Head Attention could reproduce the performance gains observed in T1.

The evaluation encompassed two scaling strategies designed to match the head capacity of T1 at 128 heads. In the first configuration, we increased $n_{heads}$ while maintaining a constant model dimension, which results in a reduced head dimension $d_k$. In the second configuration, we increased $n_{heads}$ while fixing the head dimension, a setup that scales the model dimension $d_{model}$ proportionally.

As detailed in Tables 10 and 11, increasing the head count in standard Multi-Head Attention does not consistently improve performance. For datasets such as ETTh1 and ETTm2, performance tends to plateau or deteriorate under configurations with high head counts. This trend may stem from optimization challenges or overfitting associated with the excessive fragmentation of the feature space or the rapid growth in parameters.

Most importantly, even the optimal iTransformer configuration yields an MSE of 0.072 on Weather, which remains considerably higher than the 0.029 MSE achieved by T1. These results suggest that the performance advantage of T1 derives from the structural efficacy of the Channel-Head Binding mechanism in ensuring semantically aligned information transfer, rather than merely from the increased quantity of attention heads.

Table 10: Impact of increasing the number of attention heads ($n_{heads}$) in iTransformer while keeping the model dimension ($d_{model}$) fixed. Best results for each dataset are highlighted in bold.

| Dataset | $n_{heads}$ | $d_{model}$ | $d_k$ | 0.1 MSE | 0.1 MAE | 0.3 MSE | 0.3 MAE | 0.5 MSE | 0.5 MAE | 0.7 MSE | 0.7 MAE | Avg MSE | Avg MAE |
|---------|-------------|-------------|-------|---------|---------|---------|---------|---------|---------|---------|---------|---------|---------|
| ETTh1 | 8 | 128 | 16 | **0.089** | **0.203** | **0.102** | **0.215** | **0.128** | **0.239** | 0.203 | **0.292** | **0.131** | **0.237** |
| | 16 | 128 | 8 | 0.093 | 0.208 | 0.104 | 0.219 | 0.130 | 0.240 | **0.202** | **0.292** | 0.132 | 0.240 |
| | 32 | 128 | 4 | 0.095 | 0.211 | 0.106 | 0.220 | 0.129 | 0.241 | **0.202** | **0.292** | 0.133 | 0.241 |
| | 64 | 128 | 2 | 0.097 | 0.214 | 0.110 | 0.225 | 0.133 | 0.244 | 0.211 | 0.298 | 0.138 | 0.245 |
| | 128 | 128 | 1 | 0.099 | 0.216 | 0.110 | 0.225 | 0.135 | 0.245 | 0.212 | 0.298 | 0.139 | 0.246 |
| ETTm2 | 8 | 128 | 16 | **0.024** | **0.095** | **0.027** | **0.101** | **0.033** | **0.113** | 0.049 | 0.140 | **0.033** | **0.112** |
| | 16 | 128 | 8 | 0.025 | 0.096 | **0.027** | 0.102 | **0.033** | **0.113** | 0.048 | 0.139 | **0.033** | 0.113 |
| | 32 | 128 | 4 | 0.025 | 0.098 | 0.028 | 0.103 | **0.033** | **0.113** | **0.047** | 0.138 | **0.033** | 0.113 |
| | 64 | 128 | 2 | 0.026 | 0.100 | 0.028 | 0.104 | **0.033** | 0.114 | **0.047** | 0.137 | 0.034 | 0.114 |
| | 128 | 128 | 1 | 0.026 | 0.100 | 0.029 | 0.105 | **0.033** | 0.114 | **0.047** | **0.136** | 0.034 | 0.114 |
| Weather | 8 | 512 | 64 | 0.087 | 0.139 | 0.089 | 0.139 | 0.090 | 0.140 | 0.093 | 0.142 | 0.090 | 0.140 |
| | 16 | 512 | 32 | 0.112 | 0.176 | 0.112 | 0.176 | 0.113 | 0.177 | 0.115 | 0.177 | 0.113 | 0.177 |
| | 32 | 512 | 16 | 0.081 | 0.130 | 0.082 | 0.131 | 0.083 | 0.132 | 0.088 | 0.136 | 0.083 | 0.132 |
| | 64 | 512 | 8 | 0.086 | 0.136 | 0.087 | 0.137 | 0.088 | 0.137 | 0.091 | 0.139 | 0.088 | 0.137 |
| | 128 | 512 | 4 | **0.069** | **0.110** | **0.070** | **0.110** | **0.072** | **0.112** | **0.077** | **0.116** | **0.072** | **0.112** |

Table 11: Impact of increasing the number of attention heads ($n_{\text{heads}}$) in iTransformer while keeping the head dimension ($d_k$) fixed. Best results for each dataset are highlighted in bold.

| Dataset | Configuration | | | 0.1 | | 0.3 | | 0.5 | | 0.7 | | Avg | |
|---|---|---|---|---|---|---|---|---|---|---|---|---|---|
| | $n_{\text{heads}}$ | $d_{\text{model}}$ | $d_k$ | MSE | MAE | MSE | MAE | MSE | MAE | MSE | MAE | MSE | MAE |
| ETTh1 | 8 | 128 | 16 | **0.089** | **0.203** | **0.102** | **0.215** | **0.128** | **0.239** | **0.203** | **0.292** | **0.131** | **0.237** |
| | 16 | 256 | 16 | 0.090 | 0.206 | 0.103 | 0.217 | 0.130 | 0.240 | 0.216 | 0.300 | 0.135 | 0.241 |
| | 32 | 512 | 16 | 0.109 | 0.227 | 0.120 | 0.237 | 0.146 | 0.257 | 0.239 | 0.318 | 0.154 | 0.260 |
| | 64 | 1024 | 16 | 0.129 | 0.248 | 0.144 | 0.259 | 0.171 | 0.279 | 0.256 | 0.329 | 0.175 | 0.279 |
| | 128 | 2048 | 16 | 0.323 | 0.379 | 0.330 | 0.383 | 0.347 | 0.391 | 0.389 | 0.410 | 0.347 | 0.391 |
| ETTm2 | 8 | 128 | 16 | **0.024** | **0.095** | **0.027** | **0.101** | **0.033** | **0.113** | **0.049** | **0.140** | **0.033** | **0.112** |
| | 16 | 256 | 16 | 0.025 | 0.097 | 0.028 | 0.103 | **0.033** | 0.114 | 0.051 | 0.144 | 0.034 | 0.114 |
| | 32 | 512 | 16 | 0.029 | 0.105 | 0.032 | 0.110 | 0.037 | 0.120 | 0.050 | 0.141 | 0.037 | 0.119 |
| | 64 | 1024 | 16 | 0.060 | 0.158 | 0.061 | 0.159 | 0.063 | 0.162 | 0.069 | 0.169 | 0.063 | 0.162 |
| | 128 | 2048 | 16 | 0.067 | 0.171 | 0.069 | 0.172 | 0.071 | 0.174 | 0.075 | 0.179 | 0.071 | 0.174 |
| Weather | 8 | 512 | 64 | **0.087** | **0.139** | 0.089 | **0.139** | **0.090** | **0.140** | 0.093 | 0.142 | **0.090** | **0.140** |
| | 16 | 1024 | 64 | 0.088 | 0.138 | **0.087** | 0.138 | 0.089 | 0.139 | 0.092 | 0.140 | 0.089 | 0.139 |
| | 32 | 2048 | 64 | 0.113 | 0.176 | 0.113 | 0.176 | 0.114 | 0.177 | 0.115 | 0.177 | 0.114 | 0.177 |
| | 64 | 4096 | 64 | 0.113 | 0.176 | 0.113 | 0.176 | 0.114 | 0.177 | 0.115 | 0.177 | 0.113 | 0.177 |
| | 128 | 8192 | 64 | 0.113 | 0.176 | 0.113 | 0.177 | 0.114 | 0.177 | 0.115 | 0.178 | 0.114 | 0.177 |

## D.2 SENSITIVITY ANALYSIS ON TRAINING MASK RATIOS

In practical deployment scenarios, test-time missing ratios are often unknown and dynamic. To validate the robustness of our training strategy, we conducted a comprehensive ablation study by training T1 with various masking ratios (0.1, 0.3, 0.5, 0.7) and evaluating them across a comprehensive range of test ratios.

Table 12: Impact of training mask ratios on generalization performance.Bold indicates the best performance for each test condition.

| | Training Missing Ratio | 0.1 | | 0.3 | | 0.5 | | 0.7 | |
|---|---|---|---|---|---|---|---|---|---|
| | Metric | MSE | MAE | MSE | MAE | MSE | MAE | MSE | MAE |
| ETTh1 | 0.1 | **0.023** | **0.102** | **0.023** | 0.103 | 0.026 | 0.109 | 0.034 | 0.124 |
| | 0.3 | 0.038 | 0.126 | **0.033** | **0.118** | 0.035 | 0.122 | 0.040 | 0.133 |
| | 0.5 | 0.074 | 0.168 | 0.051 | 0.145 | **0.050** | **0.142** | 0.052 | 0.148 |
| | 0.7 | 0.192 | 0.260 | 0.106 | 0.208 | 0.080 | 0.181 | **0.076** | **0.177** |
| | Avg | 0.082 | 0.164 | 0.053 | 0.143 | **0.048** | **0.139** | 0.051 | 0.145 |
| ETTh2 | 0.1 | **0.023** | **0.090** | 0.024 | 0.091 | 0.026 | 0.096 | 0.029 | 0.108 |
| | 0.3 | **0.030** | 0.105 | **0.030** | **0.103** | **0.030** | 0.105 | 0.032 | 0.113 |
| | 0.5 | 0.044 | 0.129 | 0.039 | 0.121 | **0.038** | **0.119** | 0.039 | 0.124 |
| | 0.7 | 0.074 | 0.173 | 0.063 | 0.159 | 0.054 | 0.145 | **0.052** | **0.143** |
| | Avg | 0.043 | 0.124 | 0.039 | 0.118 | **0.037** | **0.116** | 0.038 | 0.122 |
| ETTm1 | 0.1 | **0.013** | 0.074 | **0.013** | **0.073** | **0.013** | 0.076 | 0.016 | 0.083 |
| | 0.3 | 0.018 | 0.086 | **0.016** | **0.080** | **0.016** | 0.082 | 0.018 | 0.087 |
| | 0.5 | 0.033 | 0.112 | 0.022 | 0.093 | **0.021** | **0.092** | 0.022 | 0.094 |
| | 0.7 | 0.096 | 0.181 | 0.038 | 0.122 | 0.033 | 0.113 | **0.031** | **0.111** |
| | Avg | 0.040 | 0.113 | 0.022 | 0.092 | **0.021** | **0.090** | 0.022 | 0.094 |
| ETTm2 | 0.1 | **0.012** | **0.059** | **0.012** | **0.059** | **0.012** | 0.060 | 0.014 | 0.068 |
| | 0.3 | 0.016 | 0.069 | **0.015** | 0.068 | **0.015** | **0.067** | 0.016 | 0.072 |
| | 0.5 | 0.023 | 0.087 | 0.020 | 0.080 | **0.019** | **0.077** | **0.019** | 0.079 |
| | 0.7 | 0.041 | 0.122 | 0.030 | 0.102 | 0.027 | 0.094 | **0.026** | **0.093** |
| | Avg | 0.023 | 0.084 | 0.019 | 0.077 | **0.018** | **0.075** | 0.019 | 0.078 |
| Weather | 0.1 | 0.022 | 0.034 | **0.021** | **0.031** | 0.034 | 0.068 | 0.030 | 0.058 |
| | 0.3 | 0.028 | 0.046 | **0.024** | **0.034** | 0.031 | 0.055 | 0.029 | 0.051 |
| | 0.5 | 0.040 | 0.071 | **0.028** | **0.040** | 0.034 | 0.055 | 0.030 | 0.048 |
| | 0.7 | 0.064 | 0.110 | 0.038 | 0.057 | 0.047 | 0.078 | **0.035** | **0.052** |
| | Avg | 0.039 | 0.065 | **0.028** | **0.041** | 0.036 | 0.064 | 0.031 | 0.052 |
| Electricity | 0.1 | **0.034** | **0.117** | 0.037 | 0.123 | 0.044 | 0.140 | 0.053 | 0.154 |
| | 0.3 | **0.042** | 0.132 | **0.042** | **0.131** | 0.046 | 0.140 | 0.056 | 0.157 |
| | 0.5 | 0.070 | 0.176 | **0.052** | 0.148 | **0.052** | **0.147** | 0.059 | 0.159 |
| | 0.7 | 0.240 | 0.347 | 0.115 | 0.230 | 0.071 | 0.177 | **0.068** | **0.168** |
| | Avg | 0.097 | 0.193 | 0.061 | 0.158 | **0.053** | **0.151** | 0.059 | 0.160 |

Table 12 presents the detailed results across six datasets. As expected, models generally achieve optimal performance when the training missing ratio closely aligns with the test ratio. However, the results demonstrate that T1 maintains high robustness even under distribution shifts; performance degradation is significant only when there is an extreme discrepancy between training and testing conditions (e.g., training at 0.1 and testing at 0.7).

These findings support our choice of a 0.4 training mask ratio as a practical default. As a moderate masking level, it provides reasonable coverage across both sparse and dense test conditions without requiring prior knowledge of test-time missing distributions. This allows a single T1 model to be deployed across diverse missing patterns without instance-specific tuning.

### D.3 COMPARISON WITH A DIFFUSION-BASED MODEL

We compare T1 against SSSD (Alcaraz & Strodthoff, 2023), a diffusion-based imputation model that achieves strong performance through iterative stochastic refinement.

**Imputation Performance.** Table 13 presents the comparison across seven datasets under varying missing ratios. T1 achieves lower MSE on most configurations, with notable improvements on ETTm2 (83.18% average MSE reduction) and Illness (81.70%).

Table 13: Performance comparison between T1 and the diffusion-based model SSSD across varying missing ratios.

| Models | | T1 (Ours) | | SSSD | | Improvement (%) | |
|---|---|---|---|---|---|---|---|
| Metric | | MSE | MAE | MSE | MAE | MSE | MAE |
| ETTh1 | 0.1 | **0.024** | **0.104** | 0.028 | 0.115 | 12.1 | 9.65 |
| | 0.3 | **0.033** | **0.118** | 0.037 | 0.130 | 11.2 | 9.75 |
| | 0.5 | **0.048** | **0.139** | 0.052 | 0.152 | 9.1 | 8.35 |
| | 0.7 | 0.093 | **0.193** | **0.087** | **0.193** | -6.7 | 0.39 |
| | Avg | **0.049** | **0.138** | 0.051 | 0.148 | 6.44 | 7.03 |
| ETTh2 | 0.1 | **0.024** | **0.089** | 0.043 | 0.132 | 45.21 | 32.04 |
| | 0.3 | **0.029** | **0.100** | 0.056 | 0.150 | 48.76 | 33.40 |
| | 0.5 | **0.037** | **0.116** | 0.087 | 0.187 | 57.05 | 37.80 |
| | 0.7 | **0.055** | **0.146** | 0.214 | 0.294 | 74.18 | 50.40 |
| | Avg | **0.036** | **0.113** | 0.100 | 0.191 | 56.30 | 38.41 |
| ETTm1 | 0.1 | **0.013** | **0.073** | 0.018 | 0.092 | 30.16 | 20.52 |
| | 0.3 | **0.016** | **0.080** | 0.025 | 0.103 | 35.96 | 22.28 |
| | 0.5 | **0.021** | **0.091** | 0.038 | 0.122 | 44.21 | 25.20 |
| | 0.7 | **0.037** | **0.120** | 0.080 | 0.169 | 53.89 | 29.29 |
| | Avg | **0.022** | **0.091** | 0.040 | 0.122 | 41.05 | 24.32 |
| ETTm2 | 0.1 | **0.011** | **0.056** | 0.044 | 0.133 | 74.00 | 58.16 |
| | 0.3 | **0.014** | **0.063** | 0.071 | 0.172 | 80.39 | 63.64 |
| | 0.5 | **0.018** | **0.073** | 0.131 | 0.238 | 86.28 | 69.39 |
| | 0.7 | **0.026** | **0.091** | 0.330 | 0.383 | 92.06 | 76.25 |
| | Avg | **0.017** | **0.070** | 0.144 | 0.231 | 83.18 | 66.86 |
| Weather | 0.1 | **0.023** | 0.034 | 0.026 | **0.031** | 12.05 | -9.83 |
| | 0.3 | **0.025** | **0.037** | 0.030 | 0.039 | 15.2 | 13.86 |
| | 0.5 | **0.029** | **0.043** | 0.036 | 0.048 | 20.3 | 39.86 |
| | 0.7 | **0.041** | **0.066** | 0.051 | 0.069 | 19.76 | 3.87 |
| | Avg | **0.029** | **0.045** | 0.036 | 0.047 | 16.84 | 1.94 |
| Exchange | 0.1 | **0.001** | **0.014** | 0.003 | 0.035 | 54.25 | 59.34 |
| | 0.3 | **0.002** | **0.016** | 0.004 | 0.041 | 53.80 | 61.40 |
| | 0.5 | **0.002** | **0.019** | 0.007 | 0.058 | 71.78 | 67.19 |
| | 0.7 | **0.003** | **0.025** | 0.024 | 0.103 | 89.04 | 75.86 |
| | Avg | **0.002** | **0.018** | 0.009 | 0.059 | 67.22 | 65.95 |
| Illness | 0.1 | **0.016** | **0.073** | 0.109 | 0.166 | 85.59 | 55.83 |
| | 0.3 | **0.020** | **0.081** | 0.141 | 0.193 | 86.06 | 58.27 |
| | 0.5 | **0.031** | **0.098** | 0.195 | 0.231 | 84.10 | 57.50 |
| | 0.7 | **0.085** | **0.157** | 0.293 | 0.293 | 71.05 | 46.47 |
| | Avg | **0.038** | **0.102** | 0.184 | 0.221 | 81.70 | 54.52 |

**Computational Efficiency.** Table 14 compares computational requirements. T1's single-pass architecture provides substantial efficiency gains: approximately 1,344× faster inference and 4,295× faster training compared to SSSD's multi-step denoising process. T1 also requires significantly fewer parameters (0.54M vs. 48.11M) and less memory (234MB vs. 5,646MB), making it more practical for resource-constrained deployment scenarios.

Table 14: Computational efficiency comparison between T1 and SSSD.

| Metric | T1 (Ours) | SSSD | Ratio |
|---|---|---|---|
| Parameters (M) | **0.54** | 48.11 | 89× smaller |
| GFLOPs | **0.155** | 1.863 | 12× less |
| Training Memory (MB) | **234** | 5646 | 24× less |
| Inference (ms/sample) | **0.60** | 811 | 1,344× faster |
| Training (ms/iter) | **47** | 201,845 | 4,295× faster |

## D.4 VISUALIZATION OF CROSS-VARIABLE ATTENTION PATTERNS

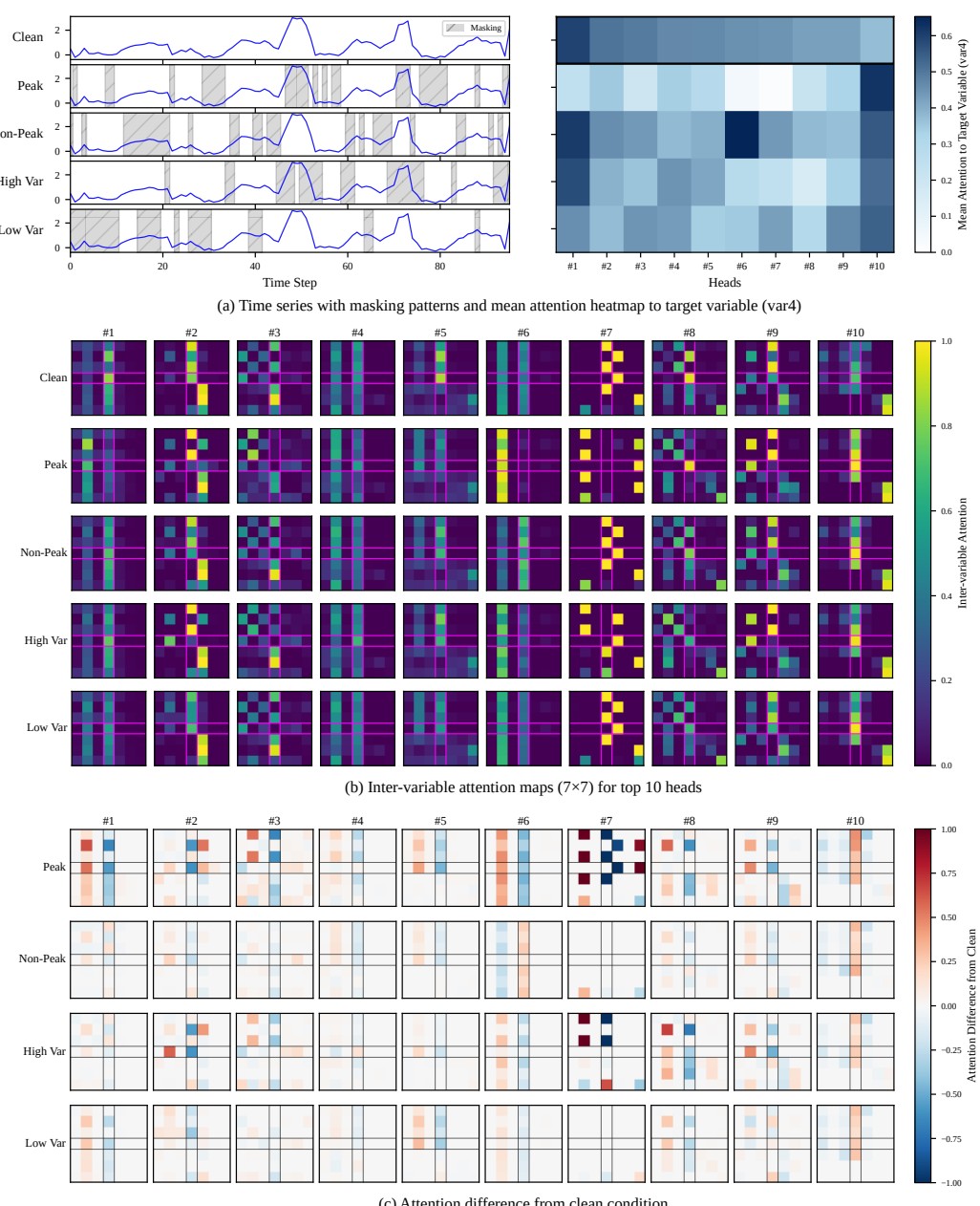

(a) Time series with masking patterns and mean attention heatmap to target variable (var4)

(b) Inter-variable attention maps (7×7) for top 10 heads

(c) Attention difference from clean condition

Figure 5: Extended attention analysis under varying missingness patterns (expansion of Figure 3(b)). (a) Left: Example time series (var4 from ETTh1) with four masking strategies targeting peak, non-peak, high-variance, and low-variance regions. Right: Mean attention weights to the target variable across 10 heads and 5 conditions. (b) Full 7×7 inter-variable attention maps for the top-10 heads (sorted by clean attention weights). Magenta lines indicate the target variable (var4). (c) Attention difference from the clean condition, showing how each head adapts its attention distribution in response to different missing patterns. Red indicates increased attention; blue indicates decreased attention.

To provide quantitative illustration of Channel-Head Binding effects, we extend Figure 3(b) with detailed attention maps. While Figure 3(b) visualizes the top-20 heads, here we focus on the top-10 heads for clearer presentation of attention dynamics under different masking conditions.

We use a single ETTh1 test sample with 7 variables. Following the experimental setup in Section 4.2.4, we mask 30% of the target variable (var4) using four strategies targeting different temporal characteristics: peak regions, non-peak regions, high-variance segments, and low-variance segments.

Figure 5(b) reveals that different heads learn specialized attention patterns. Some heads exhibit strong diagonal patterns indicating self-variable focus, while others develop off-diagonal connections capturing cross-variable dependencies.

Figure 5(c) demonstrates that these attention patterns adapt to the missingness configuration rather than remaining static. When var4 is masked, many heads reduce their attention to var4 and redistribute it to other variables. This indicates that the model recognizes unreliable sources and seeks information from alternative variables, enabled by Channel-Head Binding.

# E CASE STUDY OF PHYSIONET2012

## E.1 DATASET CHARACTERIZATION

PhysioNet Challenge 2012 contains multivariate clinical time series from 4,000 ICU patients with 37 physiological variables recorded over approximately 48 hours. The dataset exhibits substantial variable-level heterogeneity in missing rates, reflecting real-world clinical measurement protocols. Vital signs (13 variables) range from 19% to 94% missing, where continuously monitored signals (e.g., HR at 19%) contrast sharply with intermittently recorded ones (e.g., NISysABP at 94%). Lab measurements (23 variables) range from 51% to 100% missing, as they require explicit sample collection. This heterogeneity—where missing rates vary by an order of magnitude even within the same category—makes PhysioNet2012 an ideal testbed for evaluating T1's imputation robustness under realistic, non-uniform missingness.

Table 15: PhysioNet2012 variable-level missing rate distribution by category.

| Category | # Vars | Missing Rate | Examples |
|---|---|---|---|
| Vital Signs | 13 | $0.19 - 0.94$ | HR (0.19), Urine (0.37), Temp (0.67), NISysABP (0.94) |
| Lab Measurements | 23 | $0.51 - 1.00$ | Mg (0.51), PaO2 (0.90), Cholesterol (1.00) |

## E.2 PER-VARIABLE IMPUTATION PERFORMANCE

To examine per-variable imputation behavior, we report MSE and MAE for six representative variables spanning diverse missing rates (0.19 to 0.91) under the 0.5 additional masking condition (total $\sim$90% missing). Table 16 presents results for both vital signs (HR, Urine, Temp) and lab measurements (Mg, PaO2, HCT). Variables were selected to represent diverse missing rates across both categories.

Table 16: Per-variable imputation performance on PhysioNet2012 under 0.5 additional masking. Variables span diverse missing rates (0.19–0.91) across vital signs and lab measurements. Best results are in **red bold**, and second best are blue underlined.

| Variable | Category | Natural Missing Rate | T1 (Ours) MSE | T1 (Ours) MAE | TimesNet MSE | TimesNet MAE | DLinear MSE | DLinear MAE | ImputeFormer MSE | ImputeFormer MAE | SAITS MSE | SAITS MAE | PatchTST MSE | PatchTST MAE |
|---|---|---|---|---|---|---|---|---|---|---|---|---|---|---|
| HR | Vital | 0.19 | **0.189** | **0.296** | 0.237 | 0.347 | 0.245 | 0.356 | 0.392 | 0.431 | 0.208 | 0.309 | 0.351 | 0.439 |
| Urine | Vital | 0.37 | **0.364** | 0.297 | 0.408 | 0.318 | 0.408 | 0.322 | 0.458 | 0.330 | 0.426 | **0.279** | 0.420 | 0.332 |
| Temp | Vital | 0.67 | **0.239** | **0.179** | 0.306 | 0.222 | 0.300 | 0.214 | 0.339 | 0.186 | 0.327 | 0.186 | 0.317 | 0.227 |
| Mg | Lab | 0.51 | **0.126** | 0.171 | 0.190 | 0.225 | 0.195 | 0.228 | 0.207 | 0.203 | 0.133 | **0.165** | 0.299 | 0.321 |
| PaO2 | Lab | 0.90 | **0.067** | 0.085 | 0.082 | 0.118 | 0.083 | 0.117 | 0.094 | **0.077** | 0.094 | 0.079 | 0.091 | 0.120 |
| HCT | Lab | 0.91 | **0.069** | 0.087 | 0.094 | 0.128 | 0.094 | 0.127 | 0.097 | **0.076** | 0.097 | 0.076 | 0.095 | 0.127 |

T1 achieves the lowest MSE across all six variables, demonstrating consistent improvements regardless of the inherent missing rate. Notably, the model maintains its performance advantage even for variables with extremely high missing rates (e.g., PaO2, HCT), confirming its capability to robustly reconstruct dynamics from sparse, irregular observations where baseline methods often struggle.

### E.3 QUALITATIVE VISUALIZATION

To provide a clear comparison among different models, we present imputation showcases for three representative variables in Figures 6–8, which are produced by the following models: T1, Times-Net (Wu et al., 2023), PatchTST (Nie et al., 2023), DLinear (Zeng et al., 2023), ImputeFormer (Nie et al., 2024), and SAITS (Du et al., 2023). All results are shown under 50% additional masking. Among the compared models, T1 produces the most accurate imputations across various sparsity levels.

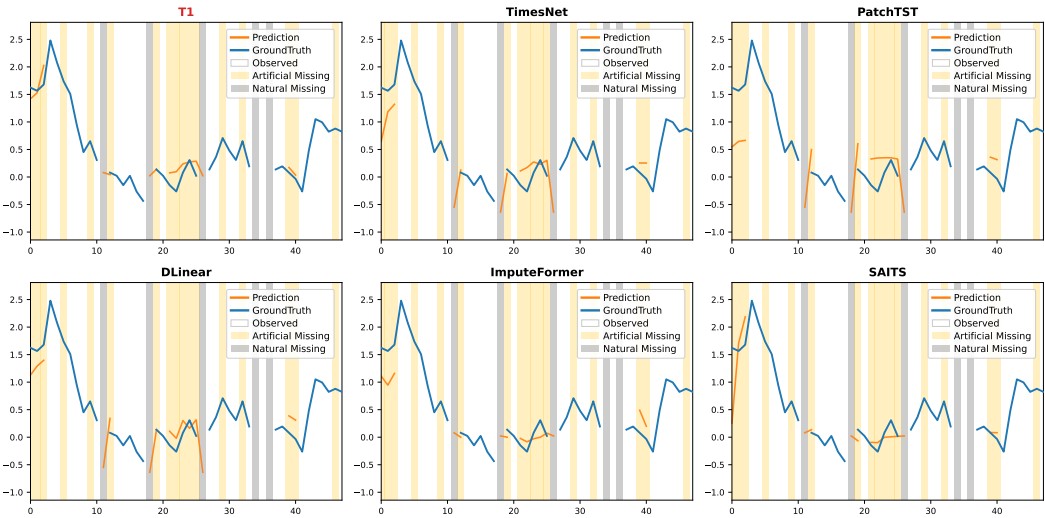

Figure 6: Visualization of imputation results on PhysioNet2012 for HR.

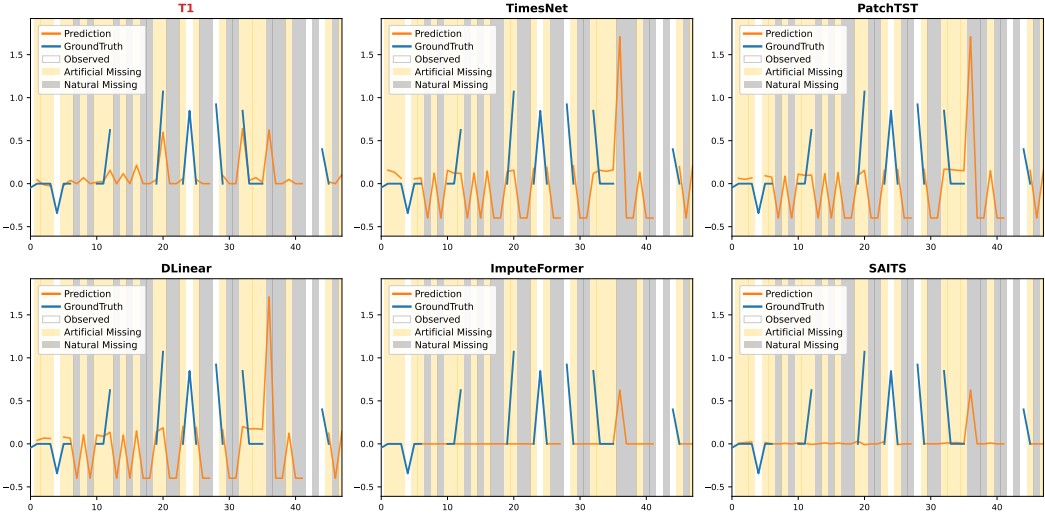

Figure 7: Visualization of imputation results on PhysioNet2012 for Temp.

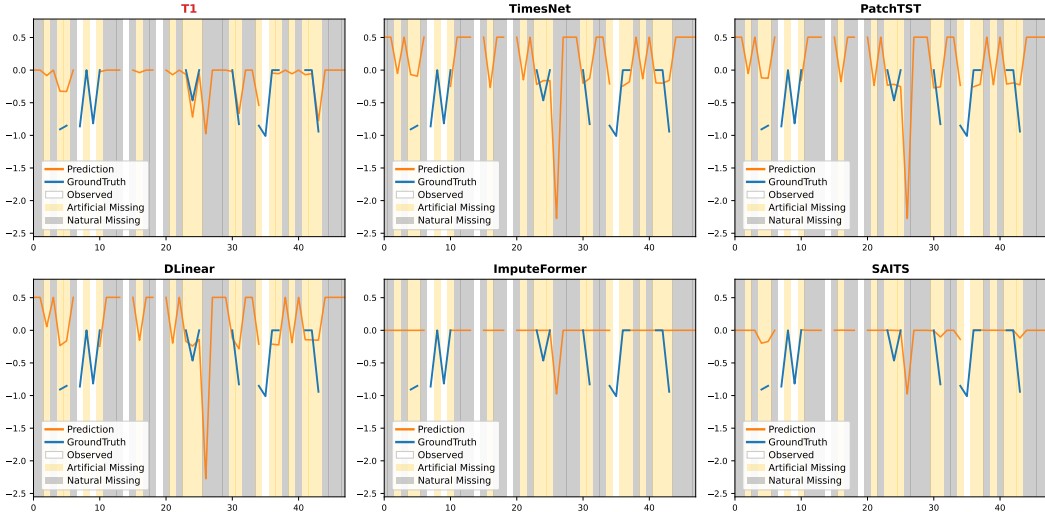

Figure 8: Visualization of imputation results on PhysioNet2012 for PaO2.

# F FULL RESULTS

Table 17: Full results with point missing ratios( 0.1, 0.3, 0.5, 0.7) across datasets.

| Models | | T1 (Ours) | | TimeMixer++ | | ModernTCN | | iTransformer | | Timesnet | | PatchTST | | DLinear | | ImputeFormer | | Saits | | CSDI | | BRITS | | PSW-I | |
|---|---|---|---|---|---|---|---|---|---|---|---|---|---|---|---|---|---|---|---|---|---|---|---|---|---|---|
| Metric | | MSE | MAE | MSE | MAE | MSE | MAE | MSE | MAE | MSE | MAE | MSE | MAE | MSE | MAE | MSE | MAE | MSE | MAE | MSE | MAE | MSE | MAE | MSE | MAE |
| ETTh1 | 0.1 | **0.024** | **0.104** | 0.090 | 0.201 | 0.053 | 0.162 | 0.087 | 0.201 | 0.094 | 0.211 | 0.044 | 0.146 | 0.161 | 0.274 | 0.064 | 0.158 | 0.026 | 0.109 | 0.039 | 0.129 | 0.052 | 0.146 | 0.079 | 0.188 |
| | 0.3 | **0.033** | **0.118** | 0.098 | 0.208 | 0.054 | 0.161 | 0.100 | 0.213 | 0.100 | 0.214 | 0.050 | 0.152 | 0.107 | 0.222 | 0.129 | 0.204 | 0.044 | 0.132 | 0.060 | 0.154 | 0.077 | 0.180 | 0.105 | 0.213 |
| | 0.5 | **0.048** | **0.139** | 0.125 | 0.229 | 0.073 | 0.181 | 0.127 | 0.237 | 0.120 | 0.229 | 0.074 | 0.180 | 0.153 | 0.251 | 0.238 | 0.279 | 0.085 | 0.181 | 0.089 | 0.187 | 0.122 | 0.234 | 0.125 | 0.234 |
| | 0.7 | **0.093** | **0.193** | 0.215 | 0.290 | 0.153 | 0.251 | 0.203 | 0.293 | 0.208 | 0.293 | 0.161 | 0.261 | 0.300 | 0.347 | 0.459 | 0.421 | 0.214 | 0.291 | 0.146 | 0.241 | 0.233 | 0.334 | 0.194 | 0.289 |
| | Avg | **0.049** | **0.138** | 0.132 | 0.232 | 0.083 | 0.189 | 0.129 | 0.236 | 0.130 | 0.237 | 0.082 | 0.185 | 0.180 | 0.273 | 0.223 | 0.266 | 0.092 | 0.178 | 0.083 | 0.178 | 0.121 | 0.223 | 0.126 | 0.231 |
| ETTh2 | 0.1 | **0.024** | **0.089** | 0.057 | 0.148 | 0.041 | 0.131 | 0.051 | 0.148 | 0.052 | 0.153 | 0.036 | 0.122 | 0.066 | 0.172 | 0.132 | 0.212 | 0.132 | 0.254 | 0.058 | 0.110 | 0.126 | 0.247 | 0.035 | 0.124 |
| | 0.3 | **0.029** | **0.100** | 0.060 | 0.151 | 0.041 | 0.130 | 0.055 | 0.154 | 0.054 | 0.156 | 0.039 | 0.127 | 0.055 | 0.156 | 0.183 | 0.251 | 0.155 | 0.276 | 0.054 | 0.127 | 0.162 | 0.281 | 0.041 | 0.133 |
| | 0.5 | **0.037** | **0.116** | 0.067 | 0.160 | 0.048 | 0.141 | 0.064 | 0.166 | 0.063 | 0.167 | 0.048 | 0.141 | 0.067 | 0.171 | 0.356 | 0.342 | 0.230 | 0.331 | 0.075 | 0.150 | 0.232 | 0.337 | 0.047 | 0.145 |
| | 0.7 | **0.055** | **0.146** | 0.087 | 0.186 | 0.075 | 0.178 | 0.085 | 0.193 | 0.092 | 0.201 | 0.075 | 0.178 | 0.104 | 0.215 | 1.047 | 0.611 | 0.583 | 0.507 | 0.116 | 0.188 | 0.385 | 0.443 | 0.059 | 0.165 |
| | Avg | **0.036** | **0.113** | 0.068 | 0.161 | 0.051 | 0.145 | 0.064 | 0.165 | 0.065 | 0.169 | 0.049 | 0.142 | 0.073 | 0.178 | 0.429 | 0.354 | 0.275 | 0.342 | 0.075 | 0.144 | 0.226 | 0.327 | 0.046 | 0.142 |
| ETTm1 | 0.1 | **0.013** | **0.073** | 0.035 | 0.117 | 0.022 | 0.101 | 0.041 | 0.132 | 0.025 | 0.106 | 0.019 | 0.092 | 0.147 | 0.248 | 0.025 | 0.102 | 0.015 | 0.082 | 0.020 | 0.091 | 0.024 | 0.099 | 0.034 | 0.112 |
| | 0.3 | **0.016** | **0.080** | 0.036 | 0.118 | 0.023 | 0.100 | 0.046 | 0.140 | 0.025 | 0.106 | 0.022 | 0.097 | 0.063 | 0.164 | 0.041 | 0.121 | 0.021 | 0.094 | 0.027 | 0.102 | 0.037 | 0.123 | 0.040 | 0.120 |
| | 0.5 | **0.021** | **0.091** | 0.042 | 0.128 | 0.032 | 0.116 | 0.057 | 0.156 | 0.035 | 0.121 | 0.031 | 0.112 | 0.089 | 0.188 | 0.075 | 0.154 | 0.038 | 0.122 | 0.036 | 0.117 | 0.063 | 0.167 | 0.048 | 0.133 |
| | 0.7 | **0.037** | **0.120** | 0.094 | 0.180 | 0.085 | 0.179 | 0.110 | 0.208 | 0.095 | 0.187 | 0.081 | 0.173 | 0.229 | 0.300 | 0.203 | 0.244 | 0.129 | 0.210 | 0.054 | 0.144 | 0.158 | 0.276 | 0.066 | 0.158 |
| | Avg | **0.022** | **0.091** | 0.052 | 0.136 | 0.040 | 0.124 | 0.063 | 0.159 | 0.045 | 0.130 | 0.038 | 0.119 | 0.132 | 0.225 | 0.086 | 0.155 | 0.051 | 0.127 | 0.034 | 0.114 | 0.070 | 0.166 | 0.047 | 0.131 |
| ETTm2 | 0.1 | **0.011** | **0.056** | 0.024 | 0.088 | 0.019 | 0.084 | 0.024 | 0.095 | 0.021 | 0.088 | 0.017 | 0.074 | 0.037 | 0.127 | 0.061 | 0.121 | 0.057 | 0.155 | 0.022 | 0.067 | 0.069 | 0.177 | 0.016 | 0.083 |
| | 0.3 | **0.014** | **0.063** | 0.026 | 0.090 | 0.020 | 0.085 | 0.027 | 0.101 | 0.021 | 0.089 | 0.019 | 0.079 | 0.028 | 0.105 | 0.067 | 0.132 | 0.071 | 0.174 | 0.027 | 0.077 | 0.108 | 0.225 | 0.018 | 0.088 |
| | 0.5 | **0.018** | **0.073** | 0.030 | 0.098 | 0.025 | 0.096 | 0.032 | 0.111 | 0.026 | 0.098 | 0.023 | 0.087 | 0.035 | 0.118 | 0.093 | 0.160 | 0.093 | 0.200 | 0.036 | 0.091 | 0.211 | 0.318 | 0.021 | 0.096 |
| | 0.7 | **0.026** | **0.091** | 0.041 | 0.119 | 0.041 | 0.125 | 0.046 | 0.136 | 0.040 | 0.125 | 0.036 | 0.114 | 0.060 | 0.160 | 0.382 | 0.317 | 0.190 | 0.273 | 0.055 | 0.111 | 0.592 | 0.536 | 0.029 | 0.110 |
| | Avg | **0.017** | **0.070** | 0.030 | 0.099 | 0.026 | 0.098 | 0.032 | 0.111 | 0.027 | 0.100 | 0.024 | 0.089 | 0.040 | 0.128 | 0.151 | 0.183 | 0.103 | 0.201 | 0.035 | 0.087 | 0.245 | 0.314 | 0.021 | 0.094 |
| Weather | 0.1 | **0.023** | **0.034** | 0.028 | 0.048 | 0.035 | 0.076 | 0.087 | 0.137 | 0.036 | 0.079 | 0.032 | 0.063 | 0.043 | 0.093 | 0.030 | 0.039 | 0.024 | 0.028 | 0.045 | 0.035 | 0.026 | 0.039 | 0.092 | 0.062 |
| | 0.3 | **0.025** | **0.037** | 0.030 | 0.047 | 0.031 | 0.059 | 0.088 | 0.137 | 0.033 | 0.065 | 0.032 | 0.058 | 0.033 | 0.063 | 0.033 | 0.042 | 0.026 | 0.031 | 0.092 | 0.038 | 0.037 | 0.062 | 0.098 | 0.066 |
| | 0.5 | **0.029** | **0.043** | 0.034 | 0.052 | 0.034 | 0.062 | 0.090 | 0.138 | 0.037 | 0.069 | 0.036 | 0.064 | 0.038 | 0.070 | 0.039 | 0.049 | 0.033 | 0.041 | 0.098 | 0.043 | 0.076 | 0.113 | 0.107 | 0.072 |
| | 0.7 | **0.041** | 0.066 | 0.045 | 0.071 | 0.051 | 0.093 | 0.093 | 0.140 | 0.055 | 0.102 | 0.049 | 0.089 | 0.060 | 0.110 | 0.065 | 0.084 | 0.055 | 0.078 | 0.099 | **0.051** | 0.307 | 0.256 | 0.131 | 0.088 |
| | Avg | **0.029** | **0.045** | 0.034 | 0.055 | 0.038 | 0.072 | 0.090 | 0.138 | 0.040 | 0.079 | 0.037 | 0.069 | 0.044 | 0.084 | 0.042 | 0.053 | 0.034 | 0.045 | 0.084 | **0.042** | 0.112 | 0.117 | 0.107 | 0.072 |
| PEMS03 | 0.1 | **0.014** | **0.076** | 0.035 | 0.131 | 0.049 | 0.162 | 0.036 | 0.134 | 0.048 | 0.160 | 0.032 | 0.120 | 0.126 | 0.269 | 0.025 | 0.096 | 0.049 | 0.134 | 0.113 | 0.141 | 0.048 | 0.128 | 0.044 | 0.142 |
| | 0.3 | **0.015** | **0.078** | 0.036 | 0.131 | 0.034 | 0.128 | 0.029 | 0.116 | 0.040 | 0.138 | 0.029 | 0.116 | 0.047 | 0.155 | 0.028 | 0.103 | 0.055 | 0.147 | 0.067 | 0.147 | 0.055 | 0.144 | 0.046 | 0.146 |
| | 0.5 | **0.018** | **0.089** | 0.039 | 0.136 | 0.036 | 0.132 | 0.036 | 0.130 | 0.046 | 0.152 | 0.034 | 0.128 | 0.054 | 0.166 | 0.050 | 0.151 | 0.060 | 0.155 | 0.068 | 0.158 | 0.074 | 0.179 | 0.049 | 0.150 |
| | 0.7 | **0.035** | **0.130** | 0.064 | 0.175 | 0.106 | 0.242 | 0.092 | 0.206 | 0.101 | 0.236 | 0.055 | 0.166 | 0.150 | 0.290 | 0.216 | 0.349 | 0.077 | 0.181 | 0.078 | 0.177 | 0.125 | 0.255 | 0.056 | 0.159 |
| | Avg | **0.021** | **0.093** | 0.044 | 0.143 | 0.056 | 0.166 | 0.048 | 0.147 | 0.059 | 0.171 | 0.038 | 0.133 | 0.094 | 0.220 | 0.080 | 0.175 | 0.060 | 0.154 | 0.082 | 0.155 | 0.076 | 0.176 | 0.049 | 0.149 |
| Exchange | 0.1 | **0.001** | **0.014** | 0.002 | 0.019 | 0.005 | 0.047 | 0.003 | 0.029 | 0.003 | 0.029 | 0.002 | 0.023 | 0.006 | 0.047 | 0.018 | 0.042 | 0.099 | 0.275 | 0.008 | 0.053 | 0.024 | 0.113 | 0.026 | 0.022 |
| | 0.3 | **0.002** | **0.016** | 0.002 | 0.020 | 0.007 | 0.057 | 0.003 | 0.031 | 0.003 | 0.028 | 0.002 | 0.023 | 0.003 | 0.033 | 0.016 | 0.044 | 0.127 | 0.304 | 0.006 | 0.051 | 0.050 | 0.175 | 0.028 | 0.023 |
| | 0.5 | **0.002** | **0.019** | 0.002 | 0.022 | 0.010 | 0.067 | 0.004 | 0.035 | 0.003 | 0.030 | 0.002 | 0.026 | 0.004 | 0.037 | 0.019 | 0.057 | 0.184 | 0.351 | 0.006 | 0.053 | 0.113 | 0.272 | 0.032 | 0.026 |
| | 0.7 | **0.003** | **0.025** | 0.003 | 0.029 | 0.013 | 0.078 | 0.005 | 0.042 | 0.005 | 0.041 | 0.004 | 0.037 | 0.008 | 0.057 | 0.071 | 0.135 | 0.311 | 0.447 | 0.008 | 0.060 | 0.272 | 0.434 | 0.039 | 0.033 |
| | Avg | **0.002** | **0.018** | 0.002 | 0.023 | 0.009 | 0.062 | 0.004 | 0.034 | 0.003 | 0.032 | 0.003 | 0.027 | 0.005 | 0.044 | 0.031 | 0.070 | 0.180 | 0.344 | 0.007 | 0.054 | 0.115 | 0.249 | 0.031 | 0.026 |
| Illness | 0.1 | **0.016** | **0.073** | 0.167 | 0.254 | 0.216 | 0.356 | 0.122 | 0.223 | 0.435 | 0.388 | 0.100 | 0.208 | 0.414 | 0.468 | 0.478 | 0.435 | 0.397 | 0.405 | 1118. | 12.22 | 0.234 | 0.293 | 0.029 | 0.086 |
| | 0.3 | **0.020** | **0.081** | 0.170 | 0.252 | 0.134 | 0.256 | 0.151 | 0.247 | 0.480 | 0.404 | 0.080 | 0.176 | 0.173 | 0.282 | 0.545 | 0.468 | 0.484 | 0.447 | 677.8 | 10.65 | 0.317 | 0.343 | 0.056 | 0.109 |
| | 0.5 | **0.031** | **0.098** | 0.220 | 0.282 | 0.189 | 0.292 | 0.208 | 0.291 | 0.585 | 0.459 | 0.107 | 0.200 | 0.227 | 0.304 | 0.643 | 0.515 | 0.629 | 0.508 | 362.9 | 7.886 | 0.451 | 0.419 | 0.070 | 0.124 |
| | 0.7 | **0.085** | **0.157** | 0.396 | 0.377 | 0.500 | 0.495 | 0.340 | 0.370 | 0.834 | 0.583 | 0.234 | 0.308 | 0.565 | 0.516 | 0.880 | 0.603 | 0.944 | 0.621 | 189.5 | 5.476 | 0.701 | 0.543 | 0.114 | 0.170 |
| | Avg | **0.038** | **0.102** | 0.238 | 0.291 | 0.260 | 0.350 | 0.205 | 0.283 | 0.583 | 0.458 | 0.130 | 0.223 | 0.345 | 0.392 | 0.636 | 0.505 | 0.614 | 0.495 | 586.9 | 9.057 | 0.426 | 0.399 | 0.067 | 0.122 |
| Electricity | 0.1 | **0.031** | **0.112** | 0.056 | 0.153 | 0.124 | 0.260 | 0.060 | 0.170 | 0.092 | 0.208 | 0.079 | 0.198 | 0.240 | 0.394 | 0.049 | 0.142 | 0.133 | 0.260 | 0.130 | 0.219 | 0.116 | 0.242 | 0.073 | 0.177 |
| | 0.3 | **0.036** | **0.119** | 0.050 | 0.145 | 0.088 | 0.208 | 0.053 | 0.153 | 0.095 | 0.213 | 0.070 | 0.185 | 0.102 | 0.235 | 0.054 | 0.148 | 0.138 | 0.264 | 0.135 | 0.226 | 0.135 | 0.265 | 0.089 | 0.196 |
| | 0.5 | **0.043** | **0.131** | 0.064 | 0.165 | 0.085 | 0.205 | 0.068 | 0.174 | 0.103 | 0.224 | 0.076 | 0.193 | 0.121 | 0.257 | 0.069 | 0.171 | 0.149 | 0.272 | 0.145 | 0.237 | 0.171 | 0.305 | 0.115 | 0.220 |
| | 0.7 | **0.063** | **0.162** | 0.116 | 0.226 | 0.189 | 0.337 | 0.180 | 0.300 | 0.128 | 0.255 | 0.130 | 0.258 | 0.302 | 0.437 | 0.133 | 0.247 | 0.189 | 0.311 | 0.167 | 0.258 | 0.247 | 0.378 | 0.148 | 0.239 |
| | Avg | **0.043** | **0.131** | 0.071 | 0.172 | 0.121 | 0.253 | 0.090 | 0.199 | 0.105 | 0.225 | 0.089 | 0.208 | 0.191 | 0.331 | 0.076 | 0.177 | 0.152 | 0.277 | 0.144 | 0.235 | 0.168 | 0.298 | 0.106 | 0.208 |

Table 18: The standard deviation of Table 17.

| Models Metric | T1 (Ours) MSE MAE | TimeMixer++ MSE MAE | ModernTCN MSE MAE | iTransformer MSE MAE | Timesnet MSE MAE | PatchTST MSE MAE | DLinear MSE MAE | ImputeFormer MSE MAE | Saits MSE MAE | CSDI MSE MAE | BRITS MSE MAE | PSW-I MSE MAE |
|---|---|---|---|---|---|---|---|---|---|---|---|---|
| **ETTh1** 0.1 | 0.001 0.001 | 0.008 0.011 | 0.004 0.006 | 0.001 0.001 | 0.001 0.001 | 0.001 0.002 | 0.002 0.001 | 0.007 0.010 | 0.002 0.005 | 0.002 0.002 | 0.008 0.004 | 0.008 0.004 |
| 0.3 | 0.000 0.001 | 0.011 0.012 | 0.001 0.002 | 0.001 0.001 | 0.000 0.001 | 0.001 0.001 | 0.001 0.001 | 0.023 0.017 | 0.006 0.009 | 0.004 0.004 | 0.004 0.008 | 0.004 0.008 |
| 0.5 | 0.000 0.000 | 0.016 0.015 | 0.002 0.003 | 0.001 0.001 | 0.001 0.001 | 0.002 0.002 | 0.001 0.001 | 0.046 0.029 | 0.015 0.017 | 0.008 0.007 | 0.003 0.009 | 0.003 0.009 |
| 0.7 | 0.002 0.002 | 0.024 0.016 | 0.008 0.007 | 0.003 0.002 | 0.004 0.003 | 0.004 0.005 | 0.002 0.001 | 0.093 0.058 | 0.033 0.030 | 0.016 0.012 | 0.004 0.006 | 0.004 0.006 |
| Avg | 0.001 0.001 | 0.015 0.013 | 0.004 0.004 | 0.001 0.001 | 0.001 0.002 | 0.002 0.003 | 0.001 0.001 | 0.042 0.029 | 0.014 0.015 | 0.008 0.006 | 0.005 0.007 | 0.005 0.007 |
| **ETTh2** 0.1 | 0.000 0.001 | 0.001 0.001 | 0.001 0.001 | 0.000 0.000 | 0.001 0.002 | 0.000 0.001 | 0.001 0.002 | 0.006 0.004 | 0.017 0.018 | 0.027 0.005 | 0.007 0.002 | 0.007 0.002 |
| 0.3 | 0.000 0.001 | 0.000 0.000 | 0.000 0.001 | 0.000 0.000 | 0.001 0.001 | 0.000 0.000 | 0.000 0.000 | 0.023 0.009 | 0.017 0.016 | 0.003 0.005 | 0.005 0.002 | 0.005 0.002 |
| 0.5 | 0.000 0.001 | 0.000 0.000 | 0.000 0.001 | 0.000 0.000 | 0.001 0.001 | 0.001 0.001 | 0.001 0.001 | 0.075 0.026 | 0.020 0.011 | 0.012 0.005 | 0.006 0.003 | 0.006 0.003 |
| 0.7 | 0.001 0.001 | 0.001 0.001 | 0.001 0.001 | 0.000 0.001 | 0.003 0.003 | 0.003 0.003 | 0.001 0.001 | 0.077 0.024 | 0.139 0.062 | 0.037 0.005 | 0.001 0.003 | 0.001 0.003 |
| Avg | 0.000 0.001 | 0.001 0.001 | 0.001 0.001 | 0.000 0.001 | 0.001 0.002 | 0.001 0.001 | 0.001 0.001 | 0.045 0.016 | 0.048 0.027 | 0.020 0.005 | 0.005 0.003 | 0.005 0.003 |
| **ETTm1** 0.1 | 0.000 0.000 | 0.000 0.001 | 0.001 0.002 | 0.001 0.001 | 0.002 0.004 | 0.000 0.000 | 0.004 0.004 | 0.004 0.005 | 0.002 0.008 | 0.001 0.002 | 0.001 0.002 | 0.001 0.002 |
| 0.3 | 0.000 0.000 | 0.000 0.000 | 0.000 0.001 | 0.001 0.001 | 0.003 0.003 | 0.000 0.000 | 0.001 0.001 | 0.012 0.009 | 0.004 0.011 | 0.001 0.002 | 0.009 0.009 | 0.009 0.009 |
| 0.5 | 0.000 0.000 | 0.000 0.001 | 0.000 0.001 | 0.001 0.002 | 0.004 0.003 | 0.000 0.001 | 0.001 0.001 | 0.032 0.021 | 0.010 0.018 | 0.002 0.003 | 0.000 0.008 | 0.000 0.008 |
| 0.7 | 0.000 0.001 | 0.006 0.004 | 0.001 0.002 | 0.003 0.002 | 0.009 0.006 | 0.018 0.017 | 0.002 0.001 | 0.103 0.056 | 0.029 0.032 | 0.003 0.003 | 0.003 0.009 | 0.003 0.009 |
| Avg | 0.000 0.000 | 0.002 0.002 | 0.001 0.001 | 0.001 0.002 | 0.004 0.004 | 0.005 0.004 | 0.002 0.002 | 0.038 0.023 | 0.011 0.017 | 0.002 0.002 | 0.003 0.007 | 0.003 0.007 |
| **ETTm2** 0.1 | 0.000 0.000 | 0.001 0.002 | 0.000 0.000 | 0.000 0.000 | 0.000 0.000 | 0.000 0.000 | 0.001 0.001 | 0.004 0.006 | 0.008 0.015 | 0.001 0.002 | 0.003 0.003 | 0.003 0.003 |
| 0.3 | 0.000 0.000 | 0.000 0.001 | 0.000 0.000 | 0.000 0.000 | 0.000 0.000 | 0.000 0.000 | 0.000 0.000 | 0.005 0.007 | 0.011 0.016 | 0.001 0.002 | 0.004 0.009 | 0.004 0.009 |
| 0.5 | 0.000 0.000 | 0.001 0.001 | 0.000 0.000 | 0.000 0.000 | 0.000 0.000 | 0.000 0.000 | 0.000 0.000 | 0.007 0.010 | 0.011 0.015 | 0.004 0.002 | 0.008 0.007 | 0.008 0.007 |
| 0.7 | 0.000 0.001 | 0.001 0.001 | 0.001 0.001 | 0.000 0.000 | 0.001 0.001 | 0.000 0.001 | 0.000 0.000 | 0.097 0.053 | 0.021 0.012 | 0.017 0.004 | 0.006 0.002 | 0.006 0.002 |
| Avg | 0.000 0.001 | 0.001 0.002 | 0.000 0.001 | 0.000 0.000 | 0.000 0.001 | 0.000 0.000 | 0.000 0.001 | 0.029 0.019 | 0.013 0.014 | 0.006 0.002 | 0.005 0.005 | 0.005 0.005 |
| **Weather** 0.1 | 0.001 0.001 | 0.000 0.001 | 0.002 0.006 | 0.000 0.000 | 0.001 0.003 | 0.001 0.001 | 0.001 0.002 | 0.001 0.002 | 0.000 0.004 | 0.045 0.035 | 0.005 0.010 | 0.005 0.010 |
| 0.3 | 0.000 0.001 | 0.000 0.001 | 0.000 0.000 | 0.000 0.000 | 0.001 0.003 | 0.001 0.001 | 0.001 0.001 | 0.003 0.004 | 0.000 0.001 | 0.092 0.038 | 0.004 0.003 | 0.004 0.003 |
| 0.5 | 0.000 0.001 | 0.000 0.001 | 0.001 0.002 | 0.000 0.000 | 0.001 0.002 | 0.001 0.001 | 0.001 0.001 | 0.002 0.003 | 0.001 0.002 | 0.098 0.043 | 0.000 0.009 | 0.000 0.009 |
| 0.7 | 0.000 0.001 | 0.000 0.001 | 0.002 0.004 | 0.001 0.000 | 0.001 0.002 | 0.002 0.004 | 0.001 0.001 | 0.004 0.006 | 0.002 0.004 | 0.099 0.051 | 0.002 0.001 | 0.002 0.001 |
| Avg | 0.000 0.001 | 0.000 0.001 | 0.001 0.003 | 0.000 0.000 | 0.001 0.002 | 0.001 0.002 | 0.001 0.001 | 0.003 0.004 | 0.001 0.002 | 0.084 0.042 | 0.003 0.006 | 0.003 0.006 |
| **PEMS03** 0.1 | 0.000 0.001 | 0.001 0.002 | 0.002 0.003 | 0.001 0.003 | 0.002 0.005 | 0.002 0.004 | 0.001 0.001 | 0.003 0.006 | 0.000 0.001 | 0.113 0.141 | 0.008 0.009 | 0.008 0.009 |
| 0.3 | 0.000 0.001 | 0.000 0.002 | 0.001 0.002 | 0.000 0.001 | 0.001 0.001 | 0.000 0.000 | 0.000 0.000 | 0.002 0.005 | 0.001 0.002 | 0.067 0.147 | 0.009 0.000 | 0.009 0.000 |
| 0.5 | 0.000 0.001 | 0.000 0.001 | 0.000 0.001 | 0.001 0.001 | 0.002 0.003 | 0.000 0.000 | 0.000 0.000 | 0.009 0.019 | 0.001 0.003 | 0.068 0.158 | 0.003 0.006 | 0.003 0.006 |
| 0.7 | 0.003 0.007 | 0.001 0.002 | 0.002 0.002 | 0.003 0.004 | 0.016 0.021 | 0.000 0.001 | 0.000 0.000 | 0.073 0.080 | 0.002 0.004 | 0.078 0.177 | 0.009 0.007 | 0.009 0.007 |
| Avg | 0.001 0.002 | 0.001 0.002 | 0.001 0.002 | 0.001 0.002 | 0.005 0.007 | 0.001 0.002 | 0.001 0.001 | 0.022 0.028 | 0.001 0.003 | 0.082 0.155 | 0.007 0.006 | 0.007 0.006 |
| **Exchange** 0.1 | 0.000 0.000 | 0.000 0.000 | 0.000 0.000 | 0.000 0.000 | 0.000 0.000 | 0.000 0.000 | 0.000 0.001 | 0.011 0.010 | 0.008 0.013 | 0.008 0.053 | 0.000 0.002 | 0.000 0.002 |
| 0.3 | 0.000 0.000 | 0.000 0.000 | 0.000 0.000 | 0.000 0.000 | 0.000 0.000 | 0.000 0.000 | 0.000 0.000 | 0.008 0.007 | 0.009 0.012 | 0.006 0.051 | 0.001 0.005 | 0.001 0.005 |
| 0.5 | 0.000 0.000 | 0.000 0.000 | 0.000 0.000 | 0.000 0.000 | 0.000 0.000 | 0.000 0.000 | 0.000 0.000 | 0.004 0.010 | 0.010 0.010 | 0.006 0.053 | 0.005 0.005 | 0.005 0.005 |
| 0.7 | 0.000 0.000 | 0.000 0.000 | 0.000 0.000 | 0.000 0.000 | 0.000 0.000 | 0.000 0.000 | 0.000 0.000 | 0.068 0.084 | 0.005 0.008 | 0.008 0.060 | 0.009 0.004 | 0.009 0.004 |
| Avg | 0.000 0.000 | 0.000 0.000 | 0.000 0.000 | 0.000 0.000 | 0.000 0.000 | 0.000 0.000 | 0.000 0.000 | 0.023 0.028 | 0.008 0.011 | 0.007 0.054 | 0.004 0.004 | 0.004 0.004 |
| **Illness** 0.1 | 0.001 0.003 | 0.050 0.043 | 0.037 0.026 | 0.000 0.000 | 0.023 0.011 | 0.003 0.002 | 0.038 0.023 | 0.030 0.017 | 0.054 0.039 | 596.549 5.643 | 0.005 0.005 | 0.005 0.005 |
| 0.3 | 0.001 0.002 | 0.045 0.042 | 0.019 0.018 | 0.000 0.003 | 0.017 0.010 | 0.003 0.004 | 0.010 0.008 | 0.031 0.015 | 0.073 0.042 | 411.993 5.189 | 0.007 0.004 | 0.007 0.004 |
| 0.5 | 0.001 0.001 | 0.044 0.040 | 0.023 0.015 | 0.000 0.000 | 0.016 0.009 | 0.001 0.000 | 0.014 0.009 | 0.026 0.011 | 0.075 0.038 | 213.524 3.328 | 0.006 0.007 | 0.006 0.007 |
| 0.7 | 0.004 0.003 | 0.024 0.031 | 0.049 0.016 | 0.009 0.004 | 0.015 0.009 | 0.010 0.006 | 0.019 0.012 | 0.039 0.011 | 0.092 0.043 | 119.279 2.150 | 0.007 0.006 | 0.007 0.006 |
| Avg | 0.002 0.002 | 0.041 0.039 | 0.032 0.019 | 0.002 0.003 | 0.018 0.009 | 0.004 0.003 | 0.020 0.013 | 0.031 0.014 | 0.073 0.041 | 335.336 4.078 | 0.006 0.006 | 0.006 0.006 |
| **Electricity** 0.1 | 0.000 0.001 | 0.007 0.008 | 0.004 0.007 | 0.002 0.003 | 0.001 0.000 | 0.001 0.002 | 0.005 0.004 | 0.003 0.004 | 0.003 0.002 | 0.025 0.019 | 0.008 0.005 | 0.008 0.005 |
| 0.3 | 0.000 0.000 | 0.007 0.008 | 0.002 0.003 | 0.000 0.000 | 0.000 0.000 | 0.000 0.000 | 0.001 0.002 | 0.003 0.004 | 0.002 0.002 | 0.021 0.018 | 0.002 0.001 | 0.002 0.001 |
| 0.5 | 0.000 0.000 | 0.008 0.007 | 0.001 0.002 | 0.000 0.000 | 0.000 0.000 | 0.000 0.000 | 0.001 0.001 | 0.004 0.007 | 0.003 0.002 | 0.019 0.015 | 0.003 0.002 | 0.003 0.002 |
| 0.7 | 0.001 0.002 | 0.007 0.007 | 0.012 0.014 | 0.002 0.002 | 0.003 0.003 | 0.001 0.001 | 0.002 0.001 | 0.016 0.020 | 0.007 0.008 | 0.016 0.012 | 0.002 0.009 | 0.002 0.009 |
| Avg | 0.000 0.001 | 0.007 0.007 | 0.005 0.006 | 0.001 0.001 | 0.001 0.001 | 0.000 0.001 | 0.002 0.002 | 0.007 0.009 | 0.004 0.003 | 0.020 0.016 | 0.004 0.004 | 0.004 0.004 |

Table 19: The standard deviation of Table 3.

| Dataset | T1 (Ours) MSE MAE | TimeMixer++ MSE MAE | ModernTCN MSE MAE | iTransformer MSE MAE | TimesNet MSE MAE | PatchTST MSE MAE | DLinear MSE MAE | ImputeFormer MSE MAE | SAITS MSE MAE |
|---|---|---|---|---|---|---|---|---|---|
| ETTh1 | 0.004 0.003 | 0.010 0.010 | 0.011 0.008 | 0.002 0.002 | 0.003 0.002 | 0.003 0.003 | 0.003 0.001 | 0.008 0.010 | 0.002 0.005 |
| ETTh2 | 0.002 0.001 | 0.002 0.001 | 0.004 0.003 | 0.006 0.002 | 0.002 0.002 | 0.000 0.001 | 0.006 0.003 | 0.022 0.008 | 0.014 0.015 |
| ETTm1 | 0.003 0.001 | 0.002 0.002 | 0.003 0.003 | 0.003 0.002 | 0.007 0.005 | 0.000 0.000 | 0.004 0.005 | 0.005 0.006 | 0.004 0.008 |
| ETTm2 | 0.001 0.001 | 0.000 0.002 | 0.001 0.001 | 0.001 0.001 | 0.001 0.001 | 0.001 0.000 | 0.002 0.001 | 0.008 0.004 | 0.010 0.014 |
| Weather | 0.001 0.001 | 0.001 0.001 | 0.003 0.007 | 0.001 0.000 | 0.003 0.003 | 0.002 0.002 | 0.001 0.002 | 0.002 0.004 | 0.001 0.001 |
| PEMS03 | 0.000 0.001 | 0.003 0.004 | 0.002 0.004 | 0.001 0.003 | 0.002 0.005 | 0.003 0.005 | 0.001 0.001 | 0.004 0.008 | 0.001 0.001 |
| Exchange | 0.001 0.001 | 0.000 0.000 | 0.000 0.001 | 0.002 0.001 | 0.000 0.000 | 0.001 0.000 | 0.001 0.000 | 0.006 0.010 | 0.007 0.013 |
| Illness | 0.011 0.007 | 0.037 0.038 | 0.044 0.033 | 0.007 0.005 | 0.017 0.007 | 0.023 0.008 | 0.038 0.021 | 0.016 0.008 | 0.045 0.034 |
| Electricity | 0.002 0.001 | 0.007 0.007 | 0.004 0.007 | 0.002 0.004 | 0.001 0.000 | 0.001 0.002 | 0.006 0.005 | 0.003 0.004 | 0.003 0.002 |
| Avg | 0.003 0.002 | 0.007 0.007 | 0.008 0.007 | 0.003 0.002 | 0.004 0.003 | 0.004 0.002 | 0.007 0.004 | 0.008 0.007 | 0.010 0.010 |

Table 20: The standard deviation of Table 4.

| PhysioNet2012 - Natural ( 80% ) + Additional Missing | | | | | | | | | |
|---|---|---|---|---|---|---|---|---|---|
| Additional Missing Ratio | T1 (Ours) MSE MAE | TimeMixer++ MSE MAE | ModernTCN MSE MAE | iTransformer MSE MAE | TimesNet MSE MAE | PatchTST MSE MAE | DLinear MSE MAE | ImputeFormer MSE MAE | SAITS MSE MAE |
| 0.1 (Total: 82%) | 0.003 0.002 | 0.001 0.001 | 0.031 0.027 | 0.012 0.006 | 0.014 0.006 | 0.026 0.018 | 0.006 0.003 | 0.010 0.009 | 0.019 0.002 |
| 0.3 (Total: 86%) | 0.005 0.002 | 0.376 0.001 | 0.029 0.024 | 0.006 0.005 | 0.010 0.005 | 0.018 0.014 | 0.005 0.003 | 0.010 0.008 | 0.010 0.002 |
| 0.5 (Total: 90%) | 0.004 0.002 | 0.035 0.001 | 0.027 0.019 | 0.011 0.004 | 0.008 0.005 | 0.015 0.011 | 0.006 0.003 | 0.010 0.007 | 0.012 0.002 |
| 0.7 (Total: 94%) | 0.006 0.002 | 0.164 0.002 | 0.018 0.013 | 0.009 0.003 | 0.008 0.003 | 0.012 0.008 | 0.010 0.003 | 0.009 0.006 | 0.010 0.002 |
| Avg | 0.004 0.002 | 0.144 0.001 | 0.026 0.021 | 0.010 0.004 | 0.010 0.005 | 0.018 0.013 | 0.007 0.003 | 0.010 0.007 | 0.013 0.002 |
| **AQI36 - Natural Missing Only (15-30%)** | | | | | | | | | |
| Test Set | 0.007 0.003 | 0.015 0.005 | 0.007 0.004 | 0.008 0.004 | 0.008 0.007 | 0.004 0.004 | 0.005 0.006 | 0.024 0.024 | 0.007 0.007 |

