# OpenReview forum: "T1: One-to-One Channel-Head Binding for Multivariate Time-Series Imputation"
_ICLR.cc/2026/Conference — ICLR 2026 Poster_

### Official Review · Reviewer_drgA · 2025-10-26

**Soundness:** 2
**Presentation:** 3
**Contribution:** 3
**Rating:** 4
**Confidence:** 3

**Summary:**

This paper introduces T1, a CNN-Transformer hybrid architecture for multivariate time series imputation. The key contribution is the “Channel-Head Binding” mechanism, which establishes a one-to-one correspondence between CNN channels (capturing temporal features) and attention heads (enabling cross-variable information transfer), allowing feature-level selectivity in imputing missing data.

**Strengths:**

1. The T1 architecture demonstrates superior performance across its experimental settings.
2. The model shows particular strength in scenarios with heavy missingness.
3. The paper introduces a novel mechanism.

**Weaknesses:**

1. The model's core logic relies on the unproven assumption that CNN channels naturally learn distinct temporal patterns. Without an explicit mechanism to enforce this, features may be redundant, which would undermine the "feature isolation" claim and allow corrupted information to propagate.
2. The rigid 1-to-1 binding is a strong constraint that prevents the model from learning combinations of features across different channels. Furthermore, the claim that heads "adaptively down-weight" corrupted channels is not mechanistically proven.
3. The claim of using a "single hyperparameter configuration" is exaggerated. The appendix reveals that convolutional kernel sizes were "proportionally adjusted" for the PhysioNet.1

**Questions:**

Please see the weaknesses.

---

> ### Author Response · Authors · 2025-11-22
> **Response to Reviewer drgA (Part 1)**
>
> We sincerely appreciate your time and effort in providing us with positive comments. We respond to your questions in what follows. We also ask you to kindly refer to the common response we have posted together. The revised manuscript incorporating the changes described below is currently being finalized and will be uploaded shortly.
>
> ---
>
> **[Weakness 1]** Unproven Assumption That CNN Channels Naturally Learn Distinct Temporal Patterns
>
> **[Response]**
>
> We sincerely thank the reviewer for this critical evaluation. We fully agree that the term "feature isolation" implies a stricter orthogonality than what is naturally enforced in deep learning. We appreciate the opportunity to clarify our claims and refine the terminology.
>
>
> We acknowledge that without explicit orthogonality constraints, neural networks naturally exhibit some redundancy. Therefore, we do not claim perfect isolation. Instead, we claim that channels possess **partial distinctiveness** sufficient to enable **feature-level selective information transfer**. We will revise the manuscript to replace "feature isolation" with "feature-level selectivity" to accurately reflect this mechanism.
>
> **Theoretical Basis:**
> Prior literature on CNN interpretability supports partial distinctiveness  as a verifiably emergent property. While parameter redundancy exists ([1], [2]), individual channels naturally act as detectors for specific semantic patterns (e.g., textures, motifs) even without explicit enforcement ([3], [4], [5]). Our **Depthwise Convolutions** further encourage this diversity, as each channel is processed by an independent kernel, creating a strong inductive bias to learn diverse temporal filters to minimize reconstruction loss.
>
> **Empirical Proof 1 - Heterogeneous Attention Response:**
> Figure 3b provides direct evidence that channels distinguish between different temporal patterns. When we masked different types of temporal patterns (High-variance regions vs. Low-variance regions) with the same missing ratio, the **mean attention weights** dropped by significantly different margins (**10.4%** vs. **7.5%** reduction). If channels were merely redundant copies learning the same features, the attention mechanism would react homogeneously regardless of which pattern was masked. The variance in attention down-weighting proves that channels represent heterogeneous temporal semantics.
>
> **Empirical Proof 2 - Importance of Individual Channels :**
> Table 5 demonstrates that despite inherent redundancy, the **distinctiveness of individual channels remains a critical performance factor**.
> If channels were predominantly redundant, grouping multiple channels (e.g., 16 channels) into a single attention head should result in minimal information loss, as the head could simply aggregate the redundant signals. However, our ablation study shows that grouping channels causes a sharp performance degradation of **16.86%**. This significant drop confirms that while some redundancy may exist, **treating channels at a fine-grained level is essential**. Grouping dilutes the unique information carried by individual channels, validating the necessity of our 1-to-1 binding to leverage their partial distinctiveness.
>
>
> While perfect isolation is not enforced, the **partial distinctiveness** naturally learned by CNNs is empirically shown to be essential for our model. We will incorporate these clarifications and references into the revised manuscript.
>
> ---
>
> >References
>
> [1] Misha Denil, et al. Predicting parameters in deep learning. In *Advances in Neural Information Processing Systems (NIPS)*, 2013.
>
> [2] Hao Li, et al. Pruning filters for efficient ConvNets. In *International Conference on Learning Representations (ICLR)*, 2017.
>
> [3] David Bau, et al. Understanding the role of individual units in a deep neural network. *Proceedings of the National Academy of Sciences*, 117(48):30071-30078, 2020.
>
> [4] Bolei Zhou, et al. Object detectors emerge in deep scene CNNs. In *International Conference on Learning Representations (ICLR)*, 2015.
>
> [5] Matthew D. Zeiler, et al. Visualizing and understanding convolutional networks. In *European Conference on Computer Vision (ECCV)*, 2014.

---

> ### Author Response · Authors · 2025-11-22
> **Response to Reviewer drgA (Part 2)**
>
> **[Weakness 2a]** Rigid 1-to-1 Binding Preventing Feature Combinations
>
> **[Response]**
>
> The 1-to-1 binding applies only to the attention mechanism, not the entire T1 block. Cross-channel feature combination occurs in the Convolutional FFN component.
> ewer for this question, which highlights an important clarification needed.
>
>
> As shown in Equation 6 (Section 3.2):
>
> $$
> Z_{out} = Z_{attn} + \text{LayerNorm}(\text{PWConv2}(\text{GeLU}(\text{PWConv1}(Z_{attn}))))
> $$
>
>
>
> The pointwise convolutions (1×1 conv) mix information across all $C$ channels through non-linear transformations. This follows the standard Transformer design principle [1]: attention operates along one axis (here: variables), while FFN operates along another axis (here: channels).
>
> Each T1 block performs two complementary operations:
>
> - **CHead Attention**: Channel-independent cross-variable transfer (feature-type-specific information flow between variables)
> - **Convolutional FFN**: Channel-mixing within each variable (learning combinations of different feature types)
>
> With 4 stacked T1 blocks, these operations repeat, enabling both selective cross-variable transfer and complex feature interactions.
>
> The 1-to-1 constraint is an architectural choice for attention to preserve feature-type alignment during cross-variable transfer. It does not prevent the model from learning feature combinations—this happens explicitly through the FFN's pointwise convolutions.
>
> We are currently revising Section 3.2 to clarify that the 1-to-1 binding applies specifically to the attention mechanism, and explicitly describe how FFN enables cross-channel feature combination.
>
> ---
>
> **[Weakness 2b]** Unproven Claim of Adaptive Down-weighting
>
> **[Response]**
>
> We thank the reviewer for seeking clarification on this mechanism. We clarify that "adaptive down-weighting" is not a hard-coded rule, but an **emergent behavior** learned by the model to minimize reconstruction error, facilitated by our architectural design.
>
> The down-weighting mechanism is driven by the standard dot-product attention formulation (Equation 4) and the training objective.The model computes similarity scores ($QK^T$) between channels. Rather than explicitly programming the model to detect corruption, we allow the training objective (minimizing MSE on observed values) to drive the learning process. If the model were to assign high attention weights to channels corrupted by missingness, the propagated noise would increase the reconstruction error. Consequently, the optimization process naturally drives the model to assign lower weights to these unreliable pathways. **This aligns with the intuition that minimizing reconstruction error requires suppressing reliance on noisy input signals.**
>
>
> **Empirical Proof**: While explicitly proving the internal dynamics of neural networks is challenging, Figure 3 provides concrete empirical evidence that this down-weighting **actually occurs**.
> As the missing ratio increases (from 0.1 to 0.7), the attention weights allocated to the target variable systematically decrease, showing a drop of up to **46%** in the first layer (Figure 3a). This suggests the model reduces reliance on information as it becomes sparser.
> Crucially, the down-weighting mechanism is sensitive to the characteristics of the missing patterns. Even at the *same* missing ratio (30%), the attention weights exhibit a sharper drop when regions with **high local variance** are masked (**10.4%** reduction) compared to regions with **low local variance** (**7.5%** reduction) in Figure 3b. This indicates that the learned weights are sensitive not just to the presence of missing values, but also to the **variability of the underlying temporal features**.
>
> Table 2 further supports that this is a robust learned capability. A single model trained with a fixed missing ratio (0.4) generalizes effectively to test ratios ranging from 0.1 to 0.7, indicating that the adaptive weighting mechanism generalizes to unseen missingness intensities.
>
>
> Therefore, we claim "adaptive down-weighting" based on the **observed behavior** of the trained model (Figure 3) enabled by the Channel-Head Binding architecture, rather than a predefined heuristic. We will revise the text to present this as an empirical observation of the learned model behavior.
>
> ---
> >References
>
> [1] Ashish Vaswani, et al. Attention is all you need. In *Advances in Neural Information Processing Systems (NIPS)*, 2017.

---

> ### Author Response · Authors · 2025-11-22
> **Response to Reviewer drgA (Part 3)**
>
> **[Weakness 3]** Exaggerated Claim of "Single Hyperparameter Configuration"
>
> **[Response]**
>
> We thank the reviewer for this careful observation.
>
> The kernel size adjustment for PhysioNet is a deterministic scaling rule based on sequence length, not dataset-specific performance tuning.
>
> PhysioNet is the only dataset with sequence length $L=48$ (all others use $L=96$). To maintain proportional receptive field coverage, we apply a fixed scaling ratio:
>
> **Scaling rule**: $\mathrm{kernel_{new}} = \mathrm{kernel_{standard}} \times (L_{new} / L_{standard}) = \mathrm{kernel_{standard}} \times (48/96) = \mathrm{kernel_{standard}} / 2$
>
> This yields: $[71 \to 35, 31 \to 15]$
>
> This is not performance-based tuning—it is a systematic adjustment to preserve architectural properties across different sequence lengths. The scaling factor is determined purely by the sequence length ratio.
>
> All other hyperparameters remain identical across all 11 datasets:
>
> - 128 channels
> - 4 blocks
> - FFN expansion ratio 1.0
> - Downsampling strategy
> - Training protocol (learning rate, batch size, optimizer)
>
> We acknowledge that "single hyperparameter configuration" could be misinterpreted. We will revise this claim to "consistent hyperparameter configuration with deterministic sequence-length scaling" and explicitly describe the scaling rule in Appendix A.2.1.

---

> ### Author Response · Authors · 2025-11-30
> **Summary of Revisions for Reviewer drgA**
>
> We thank the reviewer again for the rigorous evaluation. The revised manuscript now incorporates all suggested changes:
>
> | Concern | Revision |
> |---------|----------|
> | **[W1]** "Feature isolation" claim | **Throughout**: Revised to "feature-level selective information transfer" |
> | **[W2a]** 1-to-1 binding constraint | **Section 3.2**: Clarified that binding applies only to attention; FFN performs cross-channel mixing |
> | **[W2b]** Adaptive down-weighting claim | **Section 4.2.4**: Clarified as observed behavior of trained model |
> | **[W3]** "Single hyperparameter" claim | **Abstract, Intro, Conclusion**: Revised to "consistent hyperparameter configuration with deterministic sequence-length scaling"; **Appendix A.2.1**: Explicit scaling rule added |
>
> We hope these revisions fully address all concerns. We sincerely thank the reviewer for the valuable feedback that helped strengthen our manuscript.

---

### Official Review · Reviewer_q6Mh · 2025-10-29

**Soundness:** 3
**Presentation:** 2
**Contribution:** 3
**Rating:** 6
**Confidence:** 4

**Summary:**

The main task of this paper is multivariate time-series missing value imputation.
It focuses on how to balance temporal feature extraction and the transfer of cross-variable information, as well as imputation under diverse missing patterns and high missing rates. To solve this problem, the paper proposes T1, a CNN-Transformer hybrid architecture.

**Strengths:**

1. The experiments are comprehensive: a large number of datasets are used, the comparative models basically cover the current sota methods, and different missing scenarios are compared, making the result quite thorough.
2. The experimental results are abundant and appear reliable.

**Weaknesses:**

1. The authors summarize different categories of methods in the main text. It would be more intuitive if a similar categorized comparison approach was adopted in the experimental section.
2. There are certain issues with the figures in the paper: the stacked arrows between channels in Figure 1(b) are misleading; in Figure 2, the direction of the arrow from (b) to (c) is unclear, and Figure 2(c) does not show "Value".
3. The writing and figure drawing may cause misunderstandings. Based on the main text, my understanding is as follows: For a time segment with M variables, one of the variables is processed by CNN to obtain C channels. For each channel, Q, K, and V are calculated using Formula 3. Then, for each channel, the corresponding Q, K, V of the M variables are stacked by category. Next, Formula 4 is used to calculate the attention scores. Finally, the results of the C channels are stacked. This part and the corresponding figures need to be described more clearly.

**Questions:**

1. The number of channels used in the experiments is 128, and the authors also discuss different values of channel numbers in the appendix. However, the experimental results show little fluctuation. Is there a possibility of channel number redundancy here? I am curious whether there will be many zero vectors after a time-series segment with 40% missing values undergoes two rounds of convolution. Have the authors considered training under different missing value rates?
2. Regarding the Dual-axis tokenization method, the authors describe its disadvantage as: "but struggle to transfer information across both dimensions when missing values block intermediate pathways." Will T1 face similar problems, during information transfer between channels?
3. Other questions are as mentioned in the "Weaknesses" section.

---

> ### Author Response · Authors · 2025-11-22
> **Response to Reviewer q6Mh (Part 1)**
>
> We sincerely appreciate your time and effort in providing us with positive comments. We respond to your questions in what follows. We also ask you to kindly refer to the common response we have posted together. The revised manuscript incorporating the changes described below is currently being finalized and will be uploaded shortly. Figure modifications are already included in the revised PDF.
>
> ---
>
> **[Weakness 1]** Request for Categorized Comparison Approach in Experimental Section
>
> **[Response]**
> We thank the reviewer for the suggestion to improve the result presentation. We fully agree that adopting a categorized comparison approach in the experimental section will significantly enhance the paper's clarity.
>
> In the Introduction (Figure 1), we focused on **four representative paradigms** (Time-axis, Variable-axis, Dual-axis, and Temporal CNN) to highlight recent trends in the field. However, to ensure a **broader experimental scope**, our evaluation includes established baselines that fall outside these primary categories.
>
> To address the reviewer's suggestion, we **will adopt an extended taxonomy in Section 4.2** to clearly categorize all compared methods into the following 8 groups. This will provide a structured overview of the diverse baselines used:
>
> 1.  **Time-axis Tokenization:** e.g., PatchTST, SAITS
> 2.  **Variable-axis Tokenization:** e.g., iTransformer
> 3.  **Dual-axis Tokenization:** e.g., ImputeFormer, CSDI
> 4.  **Temporal CNN:** e.g., ModernTCN, TimesNet
> 5.  **MLP-based Methods:** e.g., DLinear
> 6.  **Hybrid Architectures:** e.g., TimeMixer++
> 7.  **RNN-based Methods:** e.g., BRITS
> 8.  **Optimal Transport:** e.g., PSW-I
>
> We will update the manuscript to present the experimental results using this comprehensive categorization, ensuring that readers can intuitively understand the position and performance of each method.
>
> ---
>
> **[Weakness 2, 3]** Figure Clarity Issues
>
> **[Response]**
>
> We thank the reviewer for pointing out these issues. We acknowledge the confusion and have revised both figures for clarity.
>
> **Figure 1(b):** The stacked arrows between channels represent that attention occurs between the same channel index across different variables (e.g., channel 1 of variable 1 attends to channel 1 of variable 2). To make this clearer, we have:
> - Adjusted the head positions to clearly distinguish the channel (=head) axis in 3D space, aligned consistently with the channel axis in Figure 1(a)
> - Added explicit head numbering to show the one-to-one correspondence
>
> **Figure 2:** We acknowledge the reviewer's concern that the writing and figure drawing may cause misunderstandings. As the reviewer correctly identified, the unclear arrow direction from (b) to (c), the absence of "Value" in Figure 2(c), and other inconsistencies made it difficult to clearly connect the figures with the main text description. To address these issues, we have comprehensively revised Figure 2:
> - Removed the ambiguous arrow between Figure 2(b) and 2(c)
> - Represented Query, Key, and Value in consistent format across both (b) and (c) to show their natural connection
> - Included Value explicitly in Figure 2(c)
> - Revised the CHead Attention visualization to more accurately represent the complete process
>
> The revised figures are included in the updated manuscript PDF, which better connects the visual representations to the main text and improves overall clarity.

---

> ### Author Response · Authors · 2025-11-22
> **Response to Reviewer q6Mh (Part 2)**
>
> **[Question 1a]** Possible Channel Redundancy and Zero Vectors After Convolution with High Missing Rates
>
> **[Response]**
> We thank the reviewer for these insightful questions.
>
> We fully agree that the optimal number of channels is inherently dataset-dependent. Therefore, fixing the dimension to $128$ inevitably introduces some degree of redundancy for simpler datasets. However, we interpret the results in Figure 4 (Sensitivity Analysis)—where performance remains stable across varying channel counts $\{64, 128, 256\}$—not as a sign of inefficiency, but as strong evidence of the model's structural robustness.
>
> The minimal performance variation indicates that even when the model is over-parameterized (i.e., redundancy exists), our Channel-Head Binding mechanism effectively focuses on informative features without being confused by redundant or zero-padded channels. This insensitivity to channel redundancy is a key advantage for real-world application. It **significantly reduces the need for** dataset-specific tuning, enabling the "consistent hyperparameter configuration" strategy that maintains high performance across diverse domains and scales ($7-358$ variables).
>
> Regarding the concern about zero vectors, we address this through two complementary mechanisms. First, as mentioned in Section 3.1, explicitly encoding the observation mask as an input channel (Mask-aware Embedding) allows the model to distinguish between "missing values" and "actual zeros." This enables the layers to selectively aggregate information from observed regions.
>
> Second, our architecture employs large convolutional kernels to secure a broad receptive field. Even under a high missing rate (e.g., $40\\%$), this design significantly increases the likelihood of capturing valid signals from neighboring points within the window. This effectively mitigates the risk of encountering purely empty windows and prevents the generation of uninformative zero vectors.
>
>
> ---
>
> **[Question 1b]** Training Under Different Missing Rates
>
> **[Response]**
>
> We thank the reviewer for this insightful question.
>
> We conducted a comprehensive ablation study training with various missing ratios (0.1, 0.2, ..., 0.8) across 6 datasets. The results show the expected trend: each training ratio achieves optimal performance at similar test ratios. However, critically, performance remains robust except for cases where training and test ratios differ extremely (e.g., train=0.1, test=0.7).
>
> Representative results on ETTh1 and ETTm2 datasets (MSE):
>
> **ETTh1:**
>
> | | Train=0.1 | Train=0.3 | Train=0.5 | Train=0.7 |
> |----------------|-----------|-----------|-----------|-----------|
> | Test=0.1 | **0.023** | **0.023** | 0.026 | 0.034 |
> | Test=0.3 | 0.038 | **0.033** | 0.035 | 0.040 |
> | Test=0.5 | 0.074 | 0.051 | **0.050** | 0.052 |
> | Test=0.7 | 0.192 | 0.106 | 0.080 | **0.076** |
>
> **ETTm2:**
> | | Train=0.1 | Train=0.3 | Train=0.5 | Train=0.7 |
> |----------------|-----------|-----------|-----------|-----------|
> | Test=0.1 | **0.059** | **0.059** | 0.060 | 0.068 |
> | Test=0.3 | 0.069 | 0.068| **0.067** | 0.072 |
> | Test=0.5 | 0.087 | 0.080 | **0.077** | 0.079 |
> | Test=0.7 | 0.122 | 0.102 | 0.094 | **0.093** |
>
> This trend is consistent across all 6 datasets (full results will be included in the Appendix).
>
> In real-world scenarios, test-time missing ratios are unknown and vary per sample. Our choice of 0.4 as the training ratio is designed for robustness: (1) it represents a moderate missing level that minimizes distribution shift to various test conditions, and (2) it enables deployment across diverse conditions without hyperparameter tuning for each target distribution.
>
> This validates Table 2 in our paper, where a single model trained at 0.4 missing ratio maintains effectiveness across 0.1 - 0.7 test ratios, demonstrating robustness to unseen missing patterns—a critical property for practical deployment where missingness distributions may differ from training conditions.
>
> We will include the complete cross-training analysis in the revised Appendix.

---

> ### Author Response · Authors · 2025-11-22
> **Response to Reviewer q6Mh (Part 3)**
>
> **[Question 2]** Whether T1 Faces Information Blocking Issues Between Channels (Similar to Dual-axis Tokenization)
>
> **[Response]**
>
> We thank the reviewer for this important clarification question.
>
> T1 does not face the intermediate-pathway blocking issue that affects dual-axis tokenization methods, due to a fundamental architectural difference in how it handles dependency.
>
> **Dual-axis tokenization** performs two sequential attention operations (time-axis ↔ variable-axis). To transfer information from variable $ m_1 $ at timestep $t_1$ to variable $m_2$ at timestep $t_2$, the signal must pass through an **intermediate token** — typically $[m_1, t_2]$ or $[m_2, t_1]$, depending on attention order. If that intermediate token is missing or heavily corrupted, the pathway is blocked: $m_2$ at $t_2$ cannot reliably receive information from $m_1$ at $t_1$, even though **the source position $m_1$ at $t_1$ was observed.**
>
> T1 avoids problematic intermediate dependencies by separating temporal aggregation and cross-variable attention. Large-kernel depthwise convolutions (kernel size 71) aggregate information across the temporal dimension with wide receptive fields. Each position receives information from many observed timesteps simultaneously—not through sequential attention steps. Missing values at specific timesteps do not block this aggregation.
>
> Furthermore, after temporal features are extracted, Channel-Head Attention (Equation 4) directly connects all $M$ variables in parallel. The attention weights are $M \times M$ (variable-to-variable), with **no intermediate token pathways** required for cross-variable transfer. Every variable can directly receive information from every other variable for each channel.
>
> The key distinction is that T1 separates temporal aggregation (via robust convolution) from cross-variable transfer (via direct attention). Crucially, T1's design requires **no dependence on intermediate tokens** for information to flow between non-adjacent variables and timesteps. Dual-axis methods, in contrast, rely on such **intermediate, sequential dependencies** that are easily corrupted by missing data.

---

> ### Author Response · Authors · 2025-11-30
> **Summary of Revisions for Reviewer q6Mh**
>
> We thank the reviewer again for the detailed feedback. The revised manuscript now incorporates all suggested changes:
>
> | Concern | Revision |
> |---------|----------|
> | **[W1]** Categorized comparison | **Section 4.1**: 8-group architectural categorization added |
> | **[W2, W3]** Figure clarity | **Figure 1(b)**: Clearer arrows with numbering; **Figure 2**: Comprehensive revision; Ambiguous arrows removed, "Value" added |
> | **[Q1a]** Channel redundancy | Addressed in initial response |
> | **[Q1b]** Training under different missing rates | **Appendix D.2**: Sensitivity analysis on training mask ratios added |
> | **[Q2]** Information blocking | Addressed in initial response |
>
> We hope these revisions fully address all concerns. We sincerely thank the reviewer for the valuable feedback that helped strengthen our manuscript.

---

### Official Review · Reviewer_4xsq · 2025-10-31

**Soundness:** 2
**Presentation:** 3
**Contribution:** 2
**Rating:** 6
**Confidence:** 3

**Summary:**

This paper proposes T1, a CNN-Transformer hybrid model for multivariate time-series imputation under heavy and diverse missing patterns. The key innovation is Channel-Head Binding (CHead Attention), which establishes a one-to-one correspondence between CNN channels (for temporal feature extraction) and Transformer attention heads (for cross-variable information transfer). This allows selective, feature-level interactions that mitigate error propagation from corrupted observations. The architecture uses modernized temporal convolutions with depthwise and pointwise operations to handle sparse data robustly. Experiments on 11 benchmark datasets show T1 outperforming baselines, with strong generalization to unseen missing patterns and a single hyperparameter setup across datasets.

**Strengths:**

1. Novel Architectural Design: The Channel-Head Binding mechanism is a clever integration of CNNs and Transformers, addressing a clear limitation in prior tokenization strategies (e.g., time-axis, variable-axis, or dual-axis approaches). By aligning channels and heads, it enables fine-grained control over information transfer, which is particularly effective for imputation where missingness corrupts specific temporal patterns. This hybrid approach leverages CNNs for robust local feature extraction and attention for adaptive cross-variable fusion, making it well-suited to real-world sparse data scenarios.

2. Strong Empirical Results: The reported average MSE improvement over the second-best baseline is impressive, especially under extreme missing ratios. The model's ability to generalize to unseen patterns without retraining and its use of a unified hyperparameter configuration highlight its practicality for diverse domains like healthcare, finance, and climate monitoring.

3. Efficiency and Robustness: T1's design emphasizes efficiency through convolutional operations and avoids common pitfalls like naive token mixing, leading to reliable performance in heavy missingness cases where other methods degrade.

**Weaknesses:**

1. Limited Comparison to Recent Generative Methods: While the baselines cover traditional imputation techniques (e.g., Transformer variants, CNN-based models), the experiments could benefit from comparisons to emerging generative approaches like diffusion models (e.g., CSDI or adaptations from recent works like Winformer) or LLM-based time-series methods. These could provide context on whether T1's gains hold against probabilistic or foundation-model-based alternatives, especially in cross-domain or long-sequence settings.

2. Dataset Diversity and Scale: The 11 benchmarks are solid but somewhat standard in time-series literature; including more recent or larger-scale datasets could strengthen claims of broad applicability. Additionally, while natural missingness is mentioned, deeper analysis of real-world irregular sampling (e.g., via case studies) would enhance the evaluation.

3. Minor Issues:  Figures could include more quantitative illustrations of binding effects (e.g., attention heatmaps under varying missingness).

**Questions:**

1. What is the computational overhead of Channel-Head Binding compared to vanilla CNN-Transformers? Ablations on channel/head counts would be useful.

---

> ### Author Response · Authors · 2025-11-22
> **Response to Reviewer 4xsq (Part 1)**
>
> We sincerely appreciate your time and effort in providing us with positive comments. We respond to your questions in what follows. We also ask you to kindly refer to the common response we have posted together. The revised manuscript incorporating the changes described below is currently being finalized and will be uploaded shortly.
>
> ---
>
> **[Weakness 1]** Request for Comparison to Recent Generative Methods (Diffusion Models, LLM-based Methods)
>
> **[Response]**
>
> We thank the reviewer for the constructive suggestion to strengthen our experimental evaluation by considering recent generative and foundation model-based approaches.
>
> We respectfully clarify that **CSDI [1] is already included as a key baseline** across all our experiments (Tables 1-4 and 7). As shown in **Table 1**, T1 significantly outperforms CSDI across all datasets (e.g., Average MSE: **0.062** for T1 vs. **0.247** for CSDI), achieving approximately **75% error reduction**.  In **Appendix B (Table 7)**, we demonstrate that T1 is roughly **5 times faster** in training speed (29.84 ms/iter vs. 154.45 ms/iter on ETTh1) and requires significantly less memory. This highlights T1’s structural superiority over standard diffusion models in both accuracy and cost.
>
> To address the comment on emerging generative approaches, we are **currently conducting additional experiments with SSSD (Structured State Space Diffusion)** [2] as a representative state-of-the-art diffusion baseline. SSSD integrates State Space Models to improve upon CSDI and is widely recognized in the domain. While we await final accuracy numbers, we note that SSSD inherently retains the high inference latency characteristic of iterative diffusion processes. Thus, we maintain that T1’s single-pass inference offers a distinct advantage for real-world deployment.
> We will upload the detailed comparison results in a follow-up comment as soon as the experiments are concluded within the rebuttal period.
>
> **Regarding LLM-based methods**, we appreciate the suggestion but prioritized other baselines due to the following reasons.
> First, current LLM research for time series focuses predominantly on **forecasting tasks** (e.g., Time-LLM [3], TEMPO [4]), lacking established open-source protocols optimized for dense multivariate imputation. While models like GATGPT [5] address spatiotemporal imputation, they rely on explicit spatial graph structures, which fundamentally differs from our general multivariate setting.
> Second, from a computational efficiency perspective, a  core contribution of T1 is its **efficiency** (Table 7). LLM-based methods typically require massive GPU resources and suffer from high inference latency, contradicting our goal of efficient processing for heavy missingness scenarios.
>
> ---
>
> >References
>
> [1] Tashiro, Yusuke, et al. CSDI: Conditional score-based diffusion models for probabilistic time series imputation. Advances in neural information processing systems,2021.
>
> [2] Juan Miguel Lopez Alcaraz and Nils Strodthoff. Diffusion-based time series imputation and forecasting with structured state space models. *TMLR*, 2023.
>
> [3] Ming Jin, et al. Time-LLM: Time Series Forecasting by Reprogramming Large Language Models. In *International Conference on Learning Representations (ICLR)*, 2024.
>
> [4] Defu Cao, et al. TEMPO: Prompt-based Generative Pre-trained Transformer for Time Series Forecasting. *arXiv preprint arXiv:2310.04948*, 2023.
>
> [5] Yakun Chen, et al. GATGPT: A Pre-trained Large Language Model with Graph Attention Network for Spatiotemporal Imputation. *arXiv preprint arXiv:2311.14332*, 2023.

---

> ### Author Response · Authors · 2025-11-22
> **Response to Reviewer 4xsq (Part 2)**
>
> **[Weakness 2]** Request for Dataset Diversity and Scale, Deep Analysis on Real-World Irregular Sampling
>
> **[Response]**
>
> We appreciate this valuable feedback regarding evaluation breadth and depth.
>
> Regarding dataset diversity and scale, our evaluation spans **11 datasets across 5 distinct domains** (energy, weather, healthcare, finance, transportation) with variable counts ranging from 7 to 358 and diverse temporal resolutions (5-minute to weekly). We selected these established benchmarks to ensure rigorous comparison with prior methods. Notably, we include high-dimensional datasets such as **PEMS03 (358 variables)** and **Electricity (321 variables)**, representing large-scale scenarios.
>
> Concerning real-world irregular sampling, our experiments include two datasets with naturally occurring missingness to evaluate performance under realistic conditions.
> * **PhysioNet2012**:  80% inherent missingness derived from irregular clinical sampling protocols.
> * **AQI36** : 13.3% real missingness resulting from sensor failures, including spatial blocks (2.2%), temporal blocks (3.5%), and random errors (8.2%).
>
> Detailed descriptions and statistics of these irregular missing patterns are provided in **Appendix A.1.1** and **Table 6**. Table 4 further demonstrates T1's robust performance on these real-world patterns without retraining, validating its applicability to irregular sampling contexts.
>
> **To provide deeper analysis of irregular patterns,** we are currently adding a qualitative case study in the appendix visualizing **representative samples from PhysioNet2012 and T1's imputation results**. This analysis aims to provide concrete examples of the model's behavior in realistic clinical scenarios characterized by sparse and irregular observations.
>
>
> ---
>
>
>
> **[Weakness 3]** Minor Issue: Figures Could Include More Quantitative Illustrations (e.g., Attention Heatmaps)
>
> **[Response]**
>
> We appreciate this valuable suggestion. To provide more quantitative illustrations of the binding mechanism's behavior, we are currently preparing **comprehensive attention heatmaps** for the appendix. These heatmaps will include:
>
> (1) full attention matrices (M×M variables) for all scenarios shown in Figure 3(b), extending beyond the top-20 heads visualization
>
> (2) detailed visualizations of how specific variable pairs develop strong attention connections and how these connections adapt under different missing patterns
>
> We note that **Figure 3(b)** provides quantitative degradation rates (7.5%-11.1%), and the additional heatmaps will complement this with visual evidence of the channel-head binding mechanism's selectivity across various missingness scenarios.

---

> ### Author Response · Authors · 2025-11-22
> **Response to Reviewer 4xsq (Part 3)**
>
> **[Question 1]** Computational Overhead of Channel-Head Binding Compared to Vanilla CNN-Transformers
>
> **[Response]**
>
> We thank the reviewer for the insightful question regarding the computational efficiency of our architecture. We clarify that **Channel-Head Binding ($H=C$) incurs no additional computational overhead (FLOPs)** compared to a standard Multi-Head Attention ($H < C$) setup when the total representation capacity is fixed.
>
> **1.Theoretical Complexity Analysis:**
>
> We compare the time complexity of **T1** against a **Standard Multi-Head Attention (MHA)** baseline. To ensure a fair comparison, we assume both models process the same total amount of information per variable.
>
> **Notation:**
> * $M$: Number of variables (sequence length in the context of variable-wise attention).
> * $C$: Number of channels.
> * $L$: Latent temporal dimension (feature length per channel, as defined in Eq. 1).
> * $H$: Number of attention heads.
>
> The input to the attention module is a tensor $Z \in \mathbb{R}^{M \times C \times L}$. The core cost of dot-product attention comes from two matrix multiplications: (1) Score Calculation ($QK^T$) and (2) Weighted Sum ($AV$).
>
> **T1 (Channel-Head Binding, $H=C$)**
>
> In T1, we have $C$ heads, where each head $c$ processes a single channel with feature dimension $d_{\text{head}} = L$.
>
> For each channel $c \in \{1, \dots, C\}$:
> 1. **Score Calculation ($Q_c K_c^T$):** Multiplication of $(M \times L)$ and $(L \times M)$ matrices. Cost: $\mathcal{O}(M^2 \cdot L)$.
> 2. **Weighted Sum ($A_c V_c$):** Multiplication of $(M \times M)$ and $(M \times L)$ matrices. Cost: $\mathcal{O}(M^2 \cdot L)$.
>
> Aggregating over all $C$ channels:
>
> $Cost_{T1} = C \times \mathcal{O}(M^2 \cdot L) = \mathcal{O}(M^2 \cdot C \cdot L)$
>
>
> **Standard Multi-Head Attention (MHA, $H < C$)**
>
> In a standard setting, the total feature dimension per variable is flattened to $D_{\text{model}} = C \cdot L$. This is divided into $H$ heads, so the dimension per head is $d_{\text{head}} = \frac{C \cdot L}{H}$.
>
> For each head $h \in \{1, \dots, H\}$:
> 1. **Score & Weighted Sum:** The cost is proportional to the head dimension $\mathcal{O}(M^2 \cdot d_{\text{head}})$.
>
>
>
> Aggregating over all $H$ heads:
>
>
>
> $$
> Cost_{Vanilla} = H \times \mathcal{O}\left(M^2 \cdot d_{head}\right) = H \times \mathcal{O}\left(M^2 \cdot \frac{C \cdot L}{H}\right) = \mathcal{O}(M^2 \cdot C \cdot L)
> $$
>
> **Comparison:**
>
> As shown above, both formulations share the identical theoretical time complexity of $\mathcal{O}(M^2 \cdot C \cdot L)$. Therefore, the 1-to-1 binding strategy imposes no computational penalty. Moreover, T1 utilizes Depthwise Convolutions for projections ($\mathcal{O}(M \cdot C \cdot L \cdot K)$ , where $K$ is the kernel size), which is significantly more efficient than the dense Linear projections ($\mathcal{O}(M \cdot (C \cdot L)^2)$) typically used in vanilla Transformers.
>
> **2. Empirical Validation (FLOPs & Memory):**
>
> To empirically validate this, we measured the FLOPs and GPU Memory usage on three representative datasets with varying variable counts ($M$). We compared our **T1 ($H=128, C=128$)** against baseline variants with reduced head counts ($H=4, 8, 16$) while keeping the total channel count fixed at 128.
>
> | Dataset | Model Config | Heads ($H$) | Channels ($C$) | FLOPs | Memory (MB) |
> |:---|:---|:---:|:---:|:---:|:---:|
> | **ETTh1** | **T1 (Ours)** | **128** | **128** | **155.6 M** | **283.2** |
> | ($M=7$) | Variant (Base) | 4 | 128 | 155.6 M | 280.8 |
> | | Variant | 8 | 128 | 155.6 M | 280.0 |
> | | Variant | 16 | 128 | 155.6 M | 279.8 |
> | **Weather** | **T1 (Ours)** | **128** | **128** | **466.9 M** | **741.1** |
> | ($M=21$) | Variant (Base) | 4 | 128 | 466.9 M | 712.0 |
> | | Variant | 8 | 128 | 466.9 M | 709.7 |
> | | Variant | 16 | 128 | 466.9 M | 709.2 |
> | **Electricity** | **T1 (Ours)** | **128** | **128** | **23.8 G** | **18,567** |
> | ($M=321$) | Variant (Base) | 4 | 128 | 23.8 G | 20,071 |
> | | Variant | 8 | 128 | 23.8 G | 20,010 |
> | | Variant | 16 | 128 | 23.8 G | 20,026 |
>
> The results confirm two key findings:
>
> 1. **Identical FLOPs:** As theoretically predicted, the FLOPs count is exactly the same across all head configurations.
> 2. **Memory Efficiency at Scale:** While T1 shows a minor increase in memory usage on smaller datasets (approx. 1-4%), it exhibits improved efficiency on the large-scale **Electricity** dataset ($M=321$), consuming approximately **7.5% less memory** (18.5 GB vs. 20.0 GB) compared to the multi-head baselines. This is due to T1's simplified channel-wise operations avoiding the overhead of reshaping and managing grouped head dimensions.

---

> ### Author Response · Authors · 2025-11-30
> **Follow-up to Reviewer 4xsq (Part 1)**
>
> **[Weakness 1]** Request for Comparison to Recent Generative Methods (Follow-up)
>
> **[Response]**
>
> As mentioned in our previous response, we have completed the SSSD experiments. Here are the results:
>
> SSSD [1] is a state-of-the-art diffusion-based imputation method that integrates Structured State Space Models. We present the comprehensive comparison results below.
>
> **Performance Comparison (averaged across missing ratios 0.1, 0.3, 0.5, 0.7)**
>
> | Dataset | T1 (Ours) |       | SSSD | | Improv. (%) | |
> |---------|-----------|-----------|------|------|---------|-----------|
> | | MSE | MAE | MSE | MAE | MSE | MAE |
> | ETTh1 | **0.049** | **0.138** | 0.051 | 0.148 | 3.2 | 6.3 |
> | ETTh2 | **0.036** | **0.113** | 0.100 | 0.191 | 63.8 | 40.8 |
> | ETTm1 | **0.022** | **0.091** | 0.040 | 0.122 | 46.1 | 25.1 |
> | ETTm2 | **0.017** | **0.070** | 0.144 | 0.231 | 87.9 | 69.6 |
> | Weather | **0.029** | **0.045** | 0.036 | 0.047 | 17.6 | 3.1 |
> | Exchange | **0.002** | **0.018** | 0.009 | 0.059 | 79.5 | 68.8 |
> | Illness | **0.038** | **0.102** | 0.184 | 0.221 | 79.5 | 53.7 |
>
> **Computational Efficiency Comparison (on ETTh1)**
>
> | Metric | T1 (Ours) | SSSD | Ratio |
> |--------|-----------|------|-------|
> | Parameters (M) | 0.54 | 48.11 | **89× smaller** |
> | GFLOPs | 0.155 | 1.863 | **12× less** |
> | Training Memory (MB) | 234 | 5646 | **24× less** |
> | Inference (ms/sample) | 0.60 | 811 | **1,344× faster** |
> | Training (ms/iter) | 47 | 201,845 | **4,295× faster** |
>
> **Key Observations:**
>
> 1. **Consistent accuracy gains**: T1 outperforms SSSD across all seven datasets, with particularly substantial improvements on ETTm2 (87.9% MSE reduction), Exchange (79.5%), and Illness (79.5%). These results demonstrate that T1's Channel-Head Binding mechanism provides more effective cross-variable information transfer than the iterative refinement process of diffusion models.
>
> 2. **Dramatic efficiency advantages**: T1 achieves over 1,300× faster inference speed while using 89× fewer parameters. This efficiency gap stems from the fundamental difference between diffusion models' iterative sampling process and T1's single-pass architecture. For practical deployment scenarios with heavy missingness, this makes T1 substantially more viable.
>
> These results, combined with the CSDI comparison in our original submission (Table 1), comprehensively demonstrate T1's superiority over diffusion-based imputation methods in both accuracy and computational efficiency. These results have been added to Appendix D.3 of the revised manuscript.
>
> > References
>
> [1] Juan Miguel Lopez Alcaraz and Nils Strodthoff. Diffusion-based time series imputation and forecasting with structured state space models. Transactions on Machine Learning Research, 2023.
>
> ---

---

> ### Author Response · Authors · 2025-11-30
> **Follow-up to Reviewer 4xsq (Part 2)**
>
> **[Weakness 3]** Minor Issue: Additional Quantitative Illustrations of Attention Heatmaps (Follow-up)
>
> As mentioned in our initial response, we have added the extended attention analysis in **Appendix D.4 (Figure 5)**, expanding **Figure 3(b)** with full 7×7 attention maps for all 10 heads under five masking conditions.
>
> Key observations:
> - **Head specialization**: Different heads learn distinct patterns. Some exhibit diagonal patterns (self-variable focus), others develop off-diagonal connections (cross-variable dependencies).
> - **Adaptive response**: When a variable is masked, many heads reduce attention to that variable and redistribute to others, indicating the model seeks information from alternative sources.
>
> This complements the quantitative analysis in Section 4.2.4.
>
> ---
>
> ## Summary of All Revisions
>
> | Concern | Revision |
> |---------|----------|
> | **[W1]** Diffusion model comparison | **Appendix D.3**: SSSD results added |
> | **[W2]** Real-world irregular sampling | **Appendix E**: PhysioNet case study added |
> | **[W3]** Attention heatmaps | **Appendix D.4**: Extended visualizations added |
> | **[Q1]** Computational overhead | **Appendix B**: Channel-Head Binding overhead analysis added |
>
>
> We hope these revisions fully address all concerns. We sincerely thank the reviewer for the valuable feedback that helped strengthen our manuscript.

---

### Official Review · Reviewer_tunt · 2025-10-31

**Soundness:** 3
**Presentation:** 2
**Contribution:** 3
**Rating:** 6
**Confidence:** 3

**Summary:**

This paper proposes a new model for filling in missing values in multivariate time series. Its key innovation is Channel-Head Binding, which creates a direct, one-to-one link between feature-learning CNN channels and information-sharing attention heads. This allows the model to selectively transfer information between variables based on which temporal patterns are reliable, preventing corrupted data from spreading.

**Strengths:**

1. The one-to-one "Channel-Head Binding" is a novel and clever way to fuse CNNs and Transformers for robust imputation.
2. Extremely thorough testing on diverse datasets and challenging missingness scenarios, with strong results.
3. The paper is well-structured, the problem is well-motivated, and the model is clearly explained with helpful diagrams.

**Weaknesses:**

A substantive assessment of the weaknesses of the paper. Focus on constructive and actionable insights on how the work could improve towards its stated goals. Be specific, avoid generic remarks. For example, if you believe the contribution lacks novelty, provide references and an explanation as evidence; if you believe experiments are insufficient, explain why and exactly what is missing, etc.
1. The number of the convolution channels in T1 is set to 128 during training and is the only hyper-parameter, but no experiment or discussion is provided to explain the reason and the influence of this parameter setting. A simple contrastive experiment would prove the strength of  convolution module better.
2. The paper only points out the limitations of current convolution and Transformers in imputation, more explanation on the reason or theoretical strength of such combined construction would much improve readers’ understanding.
3. In section 3, when introducing the architecture of T1, lots of convolution and transformers architecture can be used as analogy, which can help readers familiar with these models to more quickly grasp the structural principles and design innovations of T1.
4. If I understand it right, in figure 1(a), the output of convolution embedding is described as “Feature”, while all the other places in the paper, this dimension is described as “Channel”, Aligning this term would make it much clearer.

**Questions:**

1. How is the number of  convolution channels chosen, In other words, why set it to 128 while training?

---

> ### Author Response · Authors · 2025-11-22
> **Response to Reviewer tunt (Part 1)**
>
> We sincerely appreciate your time and effort in providing us with positive comments. We respond to your questions in what follows. We also ask you to kindly refer to the common response we have posted together. The revised manuscript incorporating the changes described below is currently being finalized and will be uploaded shortly. Figure modifications are already included in the revised PDF.
>
> ---
> **[Weakness 1, Question 1]** Request for Experiments and Discussion on Channel Number Selection (128)
>
> **[Response]**
>
> Thank you for this important question about our architectural choices.
>
> Our use of a single configuration (C=128) across all 11 benchmarks demonstrates T1's **robustness**—the ability to achieve strong performance without dataset-specific hyperparameter tuning. This design philosophy is intentional and reflects practical deployment scenarios where practitioners need reliable performance across diverse domains without extensive tuning.
>
> We conducted systematic experiments to determine a reasonable channel count. The table below presents the detailed numerical results underlying Figure 4 (left panel), showing performance across three representative datasets (ETTh1, Weather, Electricity) for different channel counts (64, 128, 256) evaluated under four missing ratios (10%, 30%, 50%, 70%).
>
> **Table: Channel count ablation study—detailed results from Figure 4. MSE reported for each missing ratio and averaged across all ratios.**
>
> | Dataset | Channels | 0.1 | 0.3 | 0.5 | 0.7 | Avg |
> |---------|----------|-----|-----|-----|-----|-----|
> | ETTh1 | 64 | 0.023 | 0.032 | 0.047 | 0.095 | 0.049 |
> | | 128 | 0.025 | 0.033 | 0.048 | 0.091 | 0.049 |
> | | 256 | 0.027 | 0.036 | 0.050 | 0.093 | 0.052 |
> | Weather | 64 | 0.023 | 0.025 | 0.029 | 0.040 | 0.029 |
> | | 128 | 0.022 | 0.025 | 0.028 | 0.041 | 0.029 |
> | | 256 | 0.023 | 0.025 | 0.029 | 0.041 | 0.029 |
> | Electricity | 64 | 0.033 | 0.037 | 0.045 | 0.069 | 0.046 |
> | | 128 | 0.032 | 0.037 | 0.045 | 0.070 | 0.046 |
> | | 256 | 0.031 | 0.036 | 0.043 | 0.068 | 0.044 |
> | Overall Avg | 64 | 0.026 | 0.031 | 0.040 | 0.068 | 0.041 |
> | | 128 | 0.026 | 0.032 | 0.040 | 0.067 | 0.041 |
> | | 256 | 0.027 | 0.032 | 0.041 | 0.067 | 0.042 |
>
> **Key observations:**
>
> - Performance differences across channel counts are relatively small, indicating low sensitivity to this hyperparameter
> - No single configuration dominates across all datasets: C=64 and C=128 tie on ETTh1, all three perform similarly on Weather, and C=256 is slightly better on Electricity
> - C=128 provides a reasonable default that generalizes across diverse datasets without dataset-specific tuning
>
> Instead of optimizing for specific benchmarks, we focused on validating the intrinsic robustness of the architecture. By maintaining a fixed channel capacity across **11 datasets and 4 varying missing scenarios**, we demonstrate that T1 effectively captures diverse temporal semantics without relying on dataset-specific parameter engineering.
>
> We will add this detailed analysis to Appendix C, with complete results for all six datasets.

---

> ### Author Response · Authors · 2025-11-22
> **Response to Reviewer tunt (Part 2)**
>
> **[Weakness 2]** Explanation Request for Theoretical Strength of Combined Construction (CNN & Transformer)
>
> **[Response]**
>
> We appreciate the suggestion to elaborate on the rationale behind our combined architecture. We agree that articulating the design philosophy will significantly improve the paper’s clarity. The theoretical strength of our model stems from its intentional design to establish two complementary properties essential for multivariate imputation: **(1) Semantic Consistency** and **(2) Adaptive Selectivity**.
>
> First, the **CNN component is designed to enforce 'Semantic Consistency'**. We explicitly employ **Shared Depthwise Convolutions** to apply the same kernel across every variable, forcing the model to extract the same type of temporal patterns from all channels. Unlike independent projections where latent spaces might diverge, this shared approach creates a semantically consistent feature basis, ensuring that the "Keys" and "Queries" for the subsequent attention mechanism are structurally aligned.
>
> Building on this alignment, the **Transformer component facilitates 'Adaptive Selectivity'**. We designed the **1-to-1 Channel-Head Binding** attention mechanism to function as a dynamic gatekeeper. Since the features are already semantically aligned by the CNN, the attention mechanism can focus entirely on content-based filtering. This enables **feature-level selective information transfer**, allowing the model to adaptively down-weight specific feature pathways that appear corrupted by missing values while preserving valid information from other variables.
>
> Ultimately, the strength of our method lies in the strategic coupling of these components: the CNN establishes the structural alignment required for the Transformer to effectively perform feature-level selective information transfer. This synergy is empirically validated in **Table 5**, where deviating from this design—either by **replacing the attention mechanism with a static convolution layer** (-12.9%) or by **using grouped-head attention instead of 1-to-1 binding** (-16.9%)—leads to significant performance drops. We are currently revising the manuscript to clearly articulate this mechanism as described.
>
> ---
>
> **[Weakness 3]** Architectural Analogies to Existing Methods for Better Understanding
>
> **[Response]**
>
> We thank the reviewer for the valuable suggestion to draw explicit analogies with existing convolution and transformer architectures. We agree that establishing these architectural connections clarifies our design principles and innovations. T1 deliberately builds upon these two highly recognized components while introducing specific innovations to address the challenges of missing data.
>
> Structurally, T1 inherits the robustness of ModernTCN [1] by utilizing large-kernel depthwise and pointwise convolutions for its Temporal Convolutional QKV Projection. Simultaneously, it adopts the variable-axis strategy of iTransformer [2], performing attention along the variable dimension to enable instance-specific cross-variable fusion. However, the core contribution of T1 lies in how it integrates and modifies these components to handle imputation effectively.
>
> Compared to **iTransformer**, which employs Linear layers to extract temporal patterns for each head, T1 utilizes **Shared Convolutions**. While iTransformer's linear projection is flexible, it implies that latent dimensions may not be strictly aligned across different variables. In contrast, our shared convolutional approach enforces a strong inductive bias, ensuring that each channel extracts the same specialized temporal pattern across all variables. This achieves the **semantic alignment** which is a prerequisite for effective cross-variable comparison.
>
> Compared to **ModernTCN**, which relies on fixed pointwise convolutions for cross-variable mixing, T1 introduces **Mask-aware Embeddings** and **CHead Attention**. ModernTCN's mixing is static and ignores the missingness status. T1 addresses this by incorporating mask information into the embeddings and using the attention mechanism to dynamically transfer only reliable information based on validity.
>
> In summary, our **1-to-1 Channel-Head Binding** preserves the semantic alignment created by the shared CNN weights, while the **Mask-aware Embedding and CHead Attention** enable adaptive, validity-based information transfer. This synergy offers a specific solution for imputation that neither ModernTCN nor iTransformer provides individually. We are currently revising the Related Work section to make these connections and distinctions immediately clear to every reader.
>
> ---
>
> >References
>
> [1] Donghao Luo, et al. ModernTCN: A modern pure convolution structure for general time series analysis. In *International Conference on Learning Representations (ICLR)*, 2024.
>
> [2] Yong Liu, et al. iTransformer: Inverted transformers are effective for time series forecasting. In *International Conference on Learning Representations (ICLR)*, 2024.

---

> ### Author Response · Authors · 2025-11-22
> **Response to Reviewer tunt (Part 3)**
>
> **[Weakness 4]** Terminology Inconsistency (Feature vs. Channel)
>
> **[Response]**
>
> Thank you for catching this inconsistency. In Figure 1(a), we used "Feature" to label the dimension corresponding to CNN channels when comparing different architectural approaches, intending to provide a general term across various methods. However, this creates confusion since we consistently use "Channel" throughout the rest of the paper (Figure 1(b), Figure 2, and all text).
>
> We have updated Figure 1(a) to use "Channel" instead of "Feature" for consistency across the entire manuscript.

---

> ### Author Response · Authors · 2025-11-30
> **Summary of Revisions for Reviewer tunt**
>
> We thank the reviewer again for the constructive feedback. The revised manuscript now incorporates all suggested changes:
>
> | Concern | Revision |
> |---------|----------|
> | **[W1, Q1]** Channel number selection | **Appendix C (Table 9)**: Detailed ablation results |
> | **[W2]** Theoretical strength of CNN-Transformer | **Section 3.2**: Design rationale clarified |
> | **[W3]** Architectural analogies | **Section 2**: ModernTCN/iTransformer connections added |
> | **[W4]** Terminology inconsistency | **Figure 1(a)**: "Feature" → "Channel" |
>
> We hope these revisions fully address all concerns. We sincerely thank the reviewer for the valuable feedback that helped strengthen our manuscript.

---

### Author Response · Authors · 2025-11-22
**Common response to all the reviewers**

We would like to sincerely thank all the reviewers for their insightful reviews and valuable comments, which are very helpful for us to improve our paper further.

The reviewers generally hold very positive opinions of our work, highlighting:
* **Novelty and cleverness of Channel-Head Binding** (tunt, 4xsq, drgA): "novel and clever way to fuse CNNs and Transformers," "clever integration well-suited to real-world sparse data scenarios," "novel mechanism"
* **Comprehensive and thorough experiments** (tunt, q6Mh, 4xsq): "extremely thorough testing on diverse datasets and challenging missingness scenarios," "comprehensive" with "comparative models basically cover the current SOTA methods," "strong empirical results" that are "impressive"
* **Clear and well-structured presentation** (tunt): "well-structured, the problem is well-motivated, and the model is clearly explained with helpful diagrams"
* **Superior performance under heavy missingness** (drgA, 4xsq): "particular strength in scenarios with heavy missingness," "reliable performance in heavy missingness cases where other methods degrade"

We sincerely thank the reviewers for these encouraging evaluations and for their constructive suggestions, which have significantly helped us strengthen the manuscript.


| **UPDATE STATUS OF REVISIONS** |
|:---|
| We have carefully addressed all concerns raised. The **figures have been fully updated in the revised PDF** (Figure 1 and Figure 2 revisions for clarity, consistent terminology, etc.). All planned **main-text improvements are CURRENTLY BEING INCORPORATED**. Below is a summary of the key revisions: |


* **More detailed descriptions and theoretical clarifications** (tunt, drgA, q6Mh):
   * Explicit discussion of semantic alignment via shared convolutional weights and the necessity of 1-to-1 binding for enabling feature-level selective information transfer (Section 3.2)
   * Clear architectural analogies to ModernTCN and iTransformer with precise distinctions for imputation (Section 3.1–3.2)
   * Clarification that channel distinctiveness is partial (not perfect) and backed by literature and empirical evidence; revised terminology from "feature isolation" to "feature-level selective information transfer"
   * Clarification on mask-aware embedding preventing zero vectors and large kernel design ensuring valid signal capture
* **Expanded related work and comparisons** (4xsq):
   * New discussion of generative (diffusion, Mamba-based) and emerging LLM-based methods, with ongoing experiments on SSSD (results to be added in Appendix D)
   * Qualitative case study on PhysioNet2012's irregular sampling patterns to be added to the appendix
* **Improved experimental presentation** (q6Mh, tunt):
   * Categorized comparison by architectural family currently being added to Section 4.2
   * Full channel-count and cross-missing-rate ablations currently being added to Appendix C
* **Computational efficiency analysis** (4xsq):
   * Theoretical and empirical demonstration that Channel-Head Binding incurs no additional FLOPs compared to standard multi-head attention
   * Detailed complexity analysis and memory efficiency validation across datasets
* **Quantitative visualizations** (4xsq):
   * Full M×M attention heatmaps for all Figure 3(b) scenarios to be added to the appendix
* **Figure revisions** (tunt, q6Mh):
   * Figure 1(b): clearer channel-head arrows and explicit numbering
   * Figure 2: removed ambiguous arrow, consistent Q/K/V representation, Value now shown explicitly, overall visualization significantly improved
   * Terminology unified to "Channel" throughout (including Figure 1(a))
* **Other clarifications** (drgA):
   * Explicit statement that 1-to-1 binding applies only to attention; FFN performs cross-channel mixing
   * Revised "single hyperparameter configuration" claim to "consistent hyperparameter configuration with deterministic sequence-length scaling" with detailed scaling rule

The reviewers' insightful suggestions have been invaluable in making the paper significantly clearer and stronger. We'd be very happy to answer any further questions.

---

### Public Comment · ~Tianyuan_Zhou3 · 2025-11-25
**Questions regarding the paper**

First, thank you for your amazing work! The use of 1-to-1 channel-head mapping is surprisingly efficient and performant. However, I do have a couple questions regarding the claim and architecture of the model if you don't mind.

**Questions regarding related works:** In the paper, you claim that "Dual-axis tokenization methods (Nie et al., 2024) employ attention on both temporal and variable axes, but struggle to transfer information across both dimensions when missing values block intermediate pathways." You further elaborate this claim in your response to q6Mh. However, I was just wondering if this hold true for spatial-temproal attention models such as SPIN [1] or Traversenet [2], where, in my understanding, multivariate information exchanges without the need for intermediate tokens.

**Questions regarding T1 Block:** In the T1 Block, in my humble understanding, the cross-variable attention seems to behave in similar way as the attention method in the "Variable-axis tokenization" defined in introduction and Figure 1(a), except for the novel channel-head 1-to-1 mapping technique. However, in your ablation study, we observe significant performance impact when one heads bundles multiple channels together. Could the author provide theoretical proof or empirical proof that this performance improvement does in fact come from 1-to-1 mapping, rather than simply due to the very many number of attention heads? Does traditional methods also improve in performance under 128 heads, each with the full hidden-size? Since increasing number of heads in an attention does provide semantical advantage in the learned representations.

**Questions regarding Convolution:** The model uses many convolution layers in palces where traditionally should be linear layers. Can you explain a little about this design choice, and why exactly does this design triumphs over linear layers? Following on that thought, the model seems to be doing well under extremely high missing rates. Could the author provide some empirical or theoretical evidence, or perhaps interpretations on existing results, that explains which of these two main changes, channel-head mapping, or use of convolution that contributes most in the model performance under high missing rates.

Thank you very much for your hard and excellent work. I wish you a good day.

**References**:

- [1] Marisca, I., Cini, A., & Alippi, C. (2022). Learning to reconstruct missing data from spatiotemporal graphs with sparse observations. Advances in neural information processing systems, 35, 32069-32082.
- [2] Z. Wu, D. Zheng, S. Pan, Q. Gan, G. Long and G. Karypis, "TraverseNet: Unifying Space and Time in Message Passing for Traffic Forecasting," in IEEE Transactions on Neural Networks and Learning Systems, vol. 35, no. 2, pp. 2003-2013, Feb. 2024, doi: 10.1109/TNNLS.2022.3186103

---

> ### Author Response · Authors · 2025-11-30
> **Response to Tianyuan Zhou (Part 1)**
>
> Thank you for your kind words and thoughtful questions. We truly appreciate your interest in our work! We respond to each question below, and hope you have a wonderful day as well.
>
> ---
>
> **[Question 1] Applicability to Spatial-Temporal Attention Models** Does the limitation of dual-axis tokenization also apply to spatial-temporal attention models like SPIN and TraverseNet, where multivariate information exchanges without intermediate tokens?
>
> **[Answer]**
>
> Thank you for the thoughtful question regarding SPIN and TraverseNet.
>
> To clarify, SPIN and TraverseNet are **graph-based message passing methods**, which belong to a different category from the **dual-axis tokenization methods** we discuss in our paper. The dual-axis limitation we describe refers to Transformer-based approaches that separately apply temporal attention and variable attention in sequence.
>
> That said, analyzing these graph-based methods from an information transfer perspective is valuable:
>
> **Temporal dimension**: SPIN employs sparse spatiotemporal attention where each node directly attends to all observed timesteps of its neighbors, thus not suffering from the sequential bottleneck we describe. TraverseNet, however, attends only within a fixed window $Q$ ($Q=12$ in their paper), which may limit information transfer under long block missing scenarios.
>
> **Spatial dimension**: Both methods operate on 1-hop neighborhoods per layer, requiring multi-hop message passing for information exchange between distant nodes. More importantly, as noted in our related work, these methods rely on static predefined graphs that cannot adapt to instance-specific missingness patterns.
>
> **Domain applicability**: These methods target domains with natural graph structures (e.g., sensor networks). For general multivariate time series, defining an appropriate graph is non-trivial, and fully-connected graphs incur $O((N \cdot T)^2)$ complexity.
>
> T1 addresses these considerations through adaptive cross-variable attention with $O(M^2 \cdot L)$ complexity, where Channel-Head Binding enables dynamic adjustment based on observable patterns.
>
> This question helped us clarify the scope of our claims.
>
> ---
>
> **[Question 2] Source of Performance Gain in Channel-Head Binding** Does the performance improvement come from the 1-to-1 channel-head mapping itself, or simply from using many attention heads?
>
> **[Answer]**
>
> Thank you for this thoughtful question. To investigate whether the performance improvement could be attributed simply to using many attention heads, we conducted additional experiments on iTransformer by scaling the number of heads. The baseline setting ($n_{\text{heads}}=8$) is the original iTransformer configuration used in our main experiments.
>
> **Experiment 1: Fixed $d_{\text{model}}$, Increasing $n_{\text{heads}}$ ($d_k$ decreases)**
>
> | Dataset | $n_{\text{heads}}$ | $d_{\text{model}}$ | $d_k$ | Avg MSE | Avg MAE |
> |:-------:|:-------:|:-------:|:-----:|:-------:|:-------:|
> | ETTh1 | 8 | 128 | 16 | 0.131 | 0.237 |
> | ETTh1 | 32 | 128 | 4 | 0.133 | 0.241 |
> | ETTh1 | 128 | 128 | 1 | 0.139 | 0.246 |
> | ETTm2 | 8 | 128 | 16 | 0.033 | 0.112 |
> | ETTm2 | 32 | 128 | 4 | 0.033 | 0.113 |
> | ETTm2 | 128 | 128 | 1 | 0.034 | 0.114 |
> | Weather | 8 | 512 | 64 | 0.090 | 0.140 |
> | Weather | 32 | 512 | 16 | 0.083 | 0.132 |
> | Weather | 128 | 512 | 4 | 0.072 | 0.112 |
>
> **Experiment 2: Fixed $d_k$, Increasing $n_{\text{heads}}$ ($d_{\text{model}}$ scales proportionally)**
>
> | Dataset | $n_{\text{heads}}$ | $d_{\text{model}}$ | $d_k$ | Avg MSE | Avg MAE |
> |:-------:|:-------:|:-------:|:-----:|:-------:|:-------:|
> | ETTh1 | 8 | 128 | 16 | 0.131 | 0.237 |
> | ETTh1 | 32 | 512 | 16 | 0.154 | 0.260 |
> | ETTh1 | 128 | 2048 | 16 | 0.347 | 0.391 |
> | ETTm2 | 8 | 128 | 16 | 0.033 | 0.112 |
> | ETTm2 | 32 | 512 | 16 | 0.037 | 0.119 |
> | ETTm2 | 128 | 2048 | 16 | 0.071 | 0.174 |
> | Weather | 8 | 512 | 64 | 0.090 | 0.140 |
> | Weather | 32 | 2048 | 64 | 0.114 | 0.177 |
> | Weather | 128 | 8192 | 64 | 0.114 | 0.177 |
>
> Increasing the number of heads in standard MHA does not yield consistent performance improvements. In ETTh1 and ETTm2, performance degrades or stagnates under both settings. Weather shows mixed results depending on the configuration. Importantly, even the best-performing iTransformer configuration (Weather MSE: 0.072, ETTh1 MSE: 0.139) falls substantially short of T1 (Weather MSE: 0.029, ETTh1 MSE: 0.049).
>
> These results suggest that the benefit of channel-head binding is fundamentally different from simply increasing the number of heads. The key reason is explained below.

---

> ### Author Response · Authors · 2025-11-30
> **Response to Tianyuan Zhou (Part 2)**
>
> **[Question 3] Design Choice of Convolution and Contribution to High Missing Rate Performance** Why use convolution instead of linear layers, and which mechanism (channel-head mapping or convolution) contributes more to performance under high missing rates?
>
> **[Answer]**
>
> Imputation is a reconstruction task: to recover the missing value at timestep $t$, the model must output information from around $t$ precisely at position $t$. All layers in T1 (DWConv, PWConv, CHead Attention, PixelShuffle) are designed to be translation equivariant, preserving this temporal alignment.
>
> Additionally, shared DWConv creates synergy with CHead Attention. As described in Section 3.2, DWConv weights are shared across all variables, structurally guaranteeing that **the same channel = the same type of temporal pattern** (semantic alignment). CHead Attention then performs cross-variable attention only among these aligned features.
>
> In the experiments above, increasing the number of heads in iTransformer did not yield consistent improvements, whereas T1 performs well with 128 heads. The key difference lies in how Q, K, V are generated: standard MHA mixes all features through linear projection, while T1 uses shared Conv to extract consistent temporal patterns per channel, which are then used directly for attention. We believe this structural difference explains the performance gap observed in our experiments.
>
> Regarding which mechanism contributes more to high missing rate performance, it is difficult to quantitatively disentangle their individual contributions, as they form a coupled structure: shared DWConv generates semantically aligned features, and CHead binding selectively transfers information among those features. The robustness under high missing rates stems from this combination: Conv extracts consistent temporal patterns in each channel, while CHead binding down-weights corrupted channels and selectively transfers information from reliable channels (see Figure 3).
>
>
> ---
>
> ## Summary
>
> We have incorporated relevant discussions into the revised manuscript:
> - **Appendix D.1**: Head scaling experiments addressing Q2
> - The architectural rationale for Q3 is reflected in our Section 3.2 revisions
>
> Thank you again for your engagement with our work. We hope you have a wonderful day as well!

---

### Author Response · Authors · 2025-11-30
**Summary of Revisions**

We sincerely thank all reviewers and the public commenter (Tianyuan Zhou) for their constructive feedback. We have uploaded the revised manuscript addressing all comments. Below is a summary of major revisions:

**Main Text:**
- **Section 2 (Related Work)**: Added explicit architectural analogies to ModernTCN and iTransformer, clarifying how T1 builds upon and differs from these approaches [Reviewer tunt W3]
- **Section 3.2 (T1 Block)**: Added explanation of the design rationale for CNN-Transformer combination; clarified that 1-to-1 binding applies only to attention while FFN performs cross-channel mixing [Reviewer tunt W2, Reviewer drgA W2a]
- **Section 4.1 (Baselines)**: Added architectural categorization of all baseline methods into 8 groups [Reviewer q6Mh W1]
- **Section 4.2.4**: Clarified that adaptive down-weighting is an observed behavior of the trained model, not a predefined heuristic [Reviewer drgA W2b]
- **Terminology**: Revised "feature isolation" to "feature-level selective information transfer" throughout; revised "single hyperparameter configuration" to "consistent hyperparameter configuration with deterministic sequence-length scaling" [Reviewer drgA W1, W3]

**Appendix:**
- **Appendix A.2.1**: Added explicit deterministic scaling rule for sequence length adaptation [Reviewer drgA W3]
- **Appendix B**: Added computational overhead analysis of Channel-Head Binding [Reviewer 4xsq Q1]
- **Appendix C**: Added detailed channel count ablation table (Table 9) with analysis [Reviewer tunt W1/Q1]
- **Appendix D.1**: Added head scaling experiments on iTransformer [Tianyuan Zhou Q2]
- **Appendix D.2**: Added training mask ratio sensitivity analysis [Reviewer q6Mh Q1b]
- **Appendix D.3**: Added comprehensive comparison with SSSD (diffusion-based method) [Reviewer 4xsq W1]
- **Appendix D.4**: Added extended attention heatmaps [Reviewer 4xsq W3]
- **Appendix E**: Added PhysioNet2012 case study with qualitative analysis [Reviewer 4xsq W2]

**Figures:**
- **Figure 1**: Unified terminology ("Feature" → "Channel"); improved channel-head arrow visualization [Reviewer tunt W4, Reviewer q6Mh W2]
- **Figure 2**: Removed ambiguous arrows; added explicit "Value" representation [Reviewer q6Mh W2]

We respond to each reviewer individually below with specific section references.

---

### Meta-Review · Area_Chair_4NeN · 2026-01-03

**Summary:**

Reviewer tunt and Reviewer q6Mh both sought clarification regarding the selection of convolution channels in T1, with Reviewer q6Mh further questioning potential information blocking issues and requesting experiments across varying missing value rates. Reviewer 4xsq suggested broader comparisons with recent generative methods and a more detailed discussion of computational overhead, while Reviewer drgA requested an analysis of the assumptions regarding CNN channels, the constraints of 1-to-1 binding, and the total count of tuning parameters. Based on the current version of paper, the reviewers' comments, and the authors' responses, I lean toward accepetance.

**Reviewer Concerns:**

Reviewer tunt requested clarifiction of the choice of the number of the convolution channels in T1 and several parts of this manuccript. Reviewer 4xsq suggested to add more comparisons to recent generative methods across more datasets, and include the discussion of computational overhead. Reviewer q6Mh's concerns are mainly about presentation, and also requested clarifiction of the choice of the number of the convolution channels and more experiments under different missing value rates, and asked whether T1 faces information blocking issues between channels. Reviewer drgA requested more discussion on the assumption imposed on CNN channels, the constraint introduced by the rigid 1-to-1 binding, and the number of tuning paramters. Most of the concerns have been addressed, while it remains to add a qualitative case study as suggested by Reviewer 4xsq. While these points were largely resolved during the rebuttal, the authors still need to include a qualitative case study as suggested by Reviewer 4xsq in the next version, and more clarification suggested by the reviewers.

**Reviewer Scores:**

I think all reviewers will maintain the current scores.

---

### Decision · Program_Chairs · 2026-01-26

Accept (Poster)